# One Rank at a Time:
# Cascading Error Dynamics in Sequential Learning

**Mahtab Alizadeh Vandchali**[*]  *ma202@rice.edu*
*Department of Computer Science*
*Rice University*

**Fangshuo Liao**[*]  *Fangshuo.Liao@rice.edu*
*Department of Computer Science*
*Rice University*

**Anastasios Kyrillidis**  *anastasios@rice.edu*
*Department of Computer Science*
*Rice University*

**Reviewed on OpenReview:** *https://openreview.net/forum?id=EG7XJANxhX*

## Abstract

Sequential learning –where complex tasks are broken down into simpler, hierarchical components– has emerged as a paradigm in AI. This paper views sequential learning through the lens of low-rank linear regression, focusing specifically on how errors propagate when learning rank-1 subspaces sequentially. We present an analysis framework that decomposes the learning process into a series of rank-1 estimation problems, where each subsequent estimation depends on the accuracy of previous steps. Our aim is explanatory rather than comparative: we analyze error propagation and derive compute allocation guidance without claiming superiority over joint or one-shot training. Our contribution is a characterization of the error propagation in this sequential process, establishing bounds on how errors –e.g., due to limited computational budgets and finite precision– affect the overall model accuracy. We prove that these errors compound in predictable ways, with implications for both algorithmic design and stability guarantees. Code is available at: `https://github.com/MahiAV/ORAT`.

## 1 Introduction

Sequential learning (Barto & Mahadevan, 2003; Pateria et al., 2021; Sahni et al., 2017; Ostapenko et al., 2024; Sordoni et al., 2024; Page-Caccia et al., 2024) is a concept found in cognitive science that posits that learning could be structured as a series of stages or levels (Shapere, 1964; Richardson, 2019; Okano et al., 2000; Carey & Markman, 1999; Ericsson et al., 1993; Council et al., 2000). Sequential learning is relevant when the notion of "skills" is orthogonal or correlated to each other, or even layered hierarchically (Farajtabar et al., 2020; Chaudhry et al., 2020). For instance, in a multitask learning environment (Caruana, 1997; Ruder, 2017; Andrychowicz et al., 2016; Crawshaw, 2020), basic skills might serve for common tasks, while specialized skills might be required for specific tasks.

Fully understanding sequential learning is an open question, even for simple models: researchers study not just how AI systems learn, but why they fail to learn when they do, and under what conditions they can learn better (Zhang et al., 2017; Bjorck et al., 2018; Baldi & Sadowski, 2013; Sun, 2020; Liu et al., 2020; Zhou et al., 2022; Wang et al., 2021; Yang et al., 2020). Just to provide a non-exhaustive list of recent efforts: Zhao et al. (2024a) presents a sequential learning strategy on videos and text for sentiment analysis, where learning features sequentially –from simpler to more complex– led to better performance. McAlister et al.

---

[*]Equal contribution.

(2024) combines deep learning with symbolic AI to tackle sequential learning tasks. Bian et al. (2023) deals with recommendation systems, such as those used by Netflix or Amazon and propose a method for these systems to learn from sequences of user interactions with multiple types of data (e.g., text, images, videos) for better recommendations.

Such efforts in understanding the empirical performance of sequential strategy necessitate a theoretical foundation of the training scheme. A line of prior works tackle the problem of sequential learning from a rank-incremental perspective. In particular, (Jin et al., 2023) proves the behavior of rank-incremental recovery in the setting of matrix sensing. A later work (Kwon et al., 2024) studies a similar behavior for the task of deep linear networks, and shows that the sequential learning of the principal components of the model can provably lead to an efficient compression algorithm. In this paper, we focus on the theoretical aspects underpinning the explicitly enforced sequential learning algorithms.

**Our setting.** We focus on low-rank subspaces as feature representations (Sanyal et al., 2018). Low-rank models are mostly compelling due to their interpretable solutions that capture influential factors in the data (Jolliffe, 1995; Koren et al., 2009; Vidal & Favaro, 2014). Assuming the simple low-rank linear regression (Candes & Recht, 2012; Recht et al., 2010; Rohde & Tsybakov, 2011; Ha & Foygel Barber, 2015) and given input $\mathbf{X} = [\mathbf{x}_1, \ldots, \mathbf{x}_n] \in \mathbb{R}^{d \times n}$, the goal is to approximate the relationship between a dependent variable $\mathbf{Y} = [\mathbf{y}_1, \ldots, \mathbf{y}_n] \in \mathbb{R}^{m \times n}$ and independent unknown variables $\mathbf{B} \in \mathbb{R}^{m \times r}$ and $\mathbf{A} \in \mathbb{R}^{r \times d}$ that result in a lower rank $r \ll \min(m, d)$ matrix $\mathbf{W} = \mathbf{BA} \in \mathbb{R}^{m \times d}$ such that:

$$\mathbf{Y} \approx \mathbf{BAX} = \mathbf{WX}.$$

Here, $(\mathbf{x}_i, \mathbf{y}_i)$ represents a data sample of a dataset $\mathcal{D} := \{(\mathbf{x}_i, \mathbf{y}_i)\}_{i=1}^n$. Conceptually, the matrix $\mathbf{W}$ projects the original $d$ features onto an $m$-dimensional space, given that first $\mathbf{x}_i$ is passed through an $r$-dimensional "funnel", thus reducing the dimensionality and complexity of the model.

In view of this, we utilize linear low-rank regression as a framework to study theoretically sequential learning processes, and how errors propagate through sequential rank-1 subspace learning. Algorithmically, the approach we consider relates to deflation in PCA (Hotelling, 1933; Mackey, 2008; Zhang, 2006; Sriperumbudur et al., 2007; Saad, 1988; Danisman et al., 2014). That is, find a rank-1 estimate based on the deflated target $\mathbf{Y}_k$, as in:

$$\mathbf{a}_k, \mathbf{b}_k = \operatorname*{argmin}_{\mathbf{a} \in \mathbb{R}^d, \mathbf{b} \in \mathbb{R}^m} \tfrac{1}{2} \left\| \mathbf{Y}_k - \mathbf{ba}^\top \mathbf{X} \right\|_F^2. \tag{1}$$

Our approach starts off with $\mathbf{Y}_1 = \mathbf{Y}$ and we obtain $\mathbf{a}_1, \mathbf{b}_1$. The matrix $\mathbf{Y}_1$ is further processed to lie in a "subspace" where the contributions of $(\mathbf{a}_1, \mathbf{b}_1)$ are removed: $\mathbf{Y}_2 := \mathbf{Y}_1 - \mathbf{b}_1 \mathbf{a}_1^\top \mathbf{X}$. This process is repeated by applying sequentially rank-1 updates on the deflated matrix, which leads to an approximation of the second pair $(\mathbf{a}_2, \mathbf{b}_2)$, and so on. Overall:

$$\mathbf{Y}_1 = \mathbf{Y}; \quad (\mathbf{a}_k, \mathbf{b}_k) = \texttt{rank-1}(\mathbf{Y}_k, \mathbf{X}, t); \quad \mathbf{Y}_{k+1} := \mathbf{Y}_k - \mathbf{b}_k \mathbf{a}_k^\top \mathbf{X}, \tag{2}$$

where $\texttt{rank-1}(\mathbf{Y}_k, \mathbf{X}, t)$ returns an approximation of a rank-1 estimate in equation 1 that minimizes the mean-squared error $\tfrac{1}{2} \left\| \mathbf{Y}_k - \mathbf{ba}^\top \mathbf{X} \right\|_F^2$, using $t$ iterations. We then estimate the subsequent subspaces by running the same rank-1 algorithm repetitively.

**Motivation.** While subspace tracking has a long history (see (Yang, 1995; Vaswani et al., 2018; Balzano et al., 2018; Peng et al., 2023) and references to these), low-rank subspaces have recently gained attention due to emerging applications in AI. Parameter-efficient fine-tuning (PEFT) methods, such as LoRA (Hu et al., 2021), have demonstrated that representing weight updates as low-rank matrices can effectively adapt large language models, while maintaining performance. We note here that many practical pipelines use sequential rank-1 updates because the procedure: *i*) aligns with deflation-style estimators long used in PCA and subspace tracking; *ii*) mirrors how low-rank adapters are deployed incrementally across tasks or capacity budgets, often without knowing the optimal rank a priori; and *iii*) fits streaming or compute-constrained settings where one adds rank one component at a time and can stop early. In these regimes, the key question is often how errors introduced at each rank-1 step propagate and when the process remains stable.

**Contributions.** This work presents a mathematical formulation of linear low-rank regression that emphasizes its decomposition into rank-1 problems. We focus on the scenario where the sub-routine, `rank-1`$(\mathbf{Y}_k, \mathbf{X}, t)$, incurs numerical errors: even solving equation 2 for a single pair, our estimate is only an approximation to the true pair. This view offers a pathway to examine how errors from each rank-1 estimation affect subsequent estimations. The following contributions are made:

- *Error Propagation Study.* We provide an examination of how errors propagate through the deflation process in linear low-rank regression, highlighting implications in the stability and accuracy.

- *Parameter recovery under fixed design.* We analyze, in the noiseless and noisy-label settings, how sequentially discovered rank-1 components recover the target parameter $\mathbf{W}^\star$ and its data-dependent components for a fixed design matrix $\mathbf{X}$. These are parameter-recovery guarantees under fixed design, not out-of-sample risk guarantees.

- *Empirical validation and exploratory extension.* We validate the theory on synthetic linear low-rank matrix regression problems, and we additionally examine whether analogous qualitative behavior appears in simple nonlinear PEFT settings. In the linear synthetic setting, front-loading consistently helps, while the nonlinear PEFT experiments exhibit a related but more nuanced scheduling pattern rather than a simple monotone front-loading rule.

## 2 Background

We use $\|\boldsymbol{a}\|_2$ to denote the $\ell_2$-norm of vector $\mathbf{a}$; $\|\mathbf{A}\|_2$ denotes the spectral norm, $\|\mathbf{A}\|_F$ the Frobenius norm of matrix $\mathbf{A}$; $\mathrm{sv}_{\mathrm{L}}(\mathbf{A})$ and $\mathrm{sv}_{\mathrm{R}}(\mathbf{A})$ denote the normalized top left and right singular vectors of $\mathbf{A}$.

**Problem setup.** Let $\mathbf{X} \in \mathbb{R}^{d \times n}$ be the input matrix of $n$ data points, each with $d$ features. For the theoretical analysis in Sections 3–4, $\mathbf{X}$ can be any fixed design matrix. The bounds below depend on $\mathbf{X}$ only through spectral quantities such as $\sigma_{\max}(\mathbf{X})$ and, where needed, $\sigma_{\min}(\mathbf{X}) > 0$. Let $\mathbf{Y} \in \mathbb{R}^{m \times n}$ be the output matrix based on the noiseless generative model, as in: $\mathbf{Y} = \mathbf{W}^\star \mathbf{X}$, that simulates the process of "inserting" data samples (columns of $\mathbf{X}$) through a low-rank linear channel $\mathbf{W}^\star \in \mathbb{R}^{m \times d}$ to obtain the corresponding column in $\mathbf{Y}$. The goal is, then, to estimate the best low rank parameter $\mathbf{W}$ given data $(\mathbf{Y}, \mathbf{X})$ as a low-rank linear regression problem:

$$\min_{\mathbf{W} \in \mathbb{R}^{m \times d}} \quad f(\mathbf{W}) := \tfrac{1}{2}\|\mathbf{Y} - \mathbf{W}\mathbf{X}\|_F^2 \quad \text{s.t.} \quad \mathrm{rank}(\mathbf{W}) \le r. \tag{3}$$

**Solutions.** This problem has a long history with various approaches, including convex (Recht et al., 2010; Lee & Bresler, 2009; Liu & Vandenberghe, 2009), non-convex projected-gradient descent (Jain et al., 2010; Lee & Bresler, 2010; Kyrillidis & Cevher, 2014; 2011; Khanna & Kyrillidis, 2017; Xu et al., 2018), as well as matrix factorization ones (Burer & Monteiro, 2003; Jain & Dhillon, 2013; Chen & Wainwright, 2015; Zhao et al., 2015; Zheng & Lafferty, 2015; Tu et al., 2016; Kyrillidis et al., 2018b; Park et al., 2016b; Sun & Luo, 2016; Bhojanapalli et al., 2016a;b; Park et al., 2016a; Ge et al., 2017; Hsieh et al., 2017; Kyrillidis et al., 2018a; Kim et al., 2023). In the latter, the problem turns into:

$$\min_{\mathbf{A} \in \mathbb{R}^{r \times d}, \ \mathbf{B} \in \mathbb{R}^{m \times r}} f(\mathbf{A}, \mathbf{B}) := \tfrac{1}{2}\|\mathbf{Y} - \mathbf{B}\mathbf{A}\mathbf{X}\|_F^2, \tag{4}$$

which is related to modernized task adaptation in neural network training, like LoRA (Hu et al., 2021; Ostapenko et al., 2024). *A key difference in our analysis is that we study the sequential nature of learning*; in contrast, in the above scenarios, one often utilizes factorized gradient descent, a low-rank solver that *updates all $r$ rank-1 components simultaneously*, as follows:

$$\mathbf{A}_{t+1} = \mathbf{A}_t - \eta_{\mathbf{A}} \nabla_{\mathbf{A}} f(\mathbf{A}_t, \mathbf{B}_t), \quad \mathbf{B}_{t+1} = \mathbf{B}_t - \eta_{\mathbf{B}} \nabla_{\mathbf{B}} f(\mathbf{A}_t, \mathbf{B}_t), \quad \text{with } \eta_{\mathbf{A}}, \eta_{\mathbf{B}} \text{ learning rates.}$$

We acknowledge solving (3)-(4) directly with these methods when $r$ is known is more efficient and could be preferable in terms of accuracy, yet does not fall into the sequential scenario we focus on, where $r$ could increase over time or is considered unknown.

**Learning rank-1 subspaces sequentially.** Our aim is to study routines like the ones described in (1) and (2). To do so, we will need to understand the behavior of both the *exact sequential low-rank recovery*, as well as the *inexact sequential low-rank recovery*. We describe some simple algorithms to motivate our work.

---

**Algorithm 1** Exact Sequential Low-Rank

**Require:** Input data $\mathbf{X} \in \mathbb{R}^{d \times n}$, output data $\mathbf{Y} \in \mathbb{R}^{m \times n}$, target rank $r$
**Ensure:** Rank-1 components $\{(\mathbf{a}_k^\star, \mathbf{b}_k^\star)\}_{k=1}^r$
1: $\mathbf{Y}_1^\star \leftarrow \mathbf{Y}$
2: **for** $k = 1$ **to** $r$ **do**
3: $\quad (\mathbf{a}_k^\star, \mathbf{b}_k^\star) \leftarrow \underset{\mathbf{a} \in \mathbb{R}^d, \mathbf{b} \in \mathbb{R}^m}{\operatorname{argmin}} \frac{1}{2} \left\| \mathbf{Y}_k^\star - \mathbf{b}\mathbf{a}^\top \mathbf{X} \right\|_F^2$
4: $\quad \mathbf{Y}_{k+1}^\star \leftarrow \mathbf{Y}_k^\star - \mathbf{b}_k^\star \mathbf{a}_k^{\star\top} \mathbf{X}$
5: **end for**
6: **return** $\{(\mathbf{a}_k^\star, \mathbf{b}_k^\star)\}_{k=1}^r$

---

**Algorithm 2** Inexact Sequential Low-Rank

**Require:** Input data $\mathbf{X} \in \mathbb{R}^{d \times n}$, output data $\mathbf{Y} \in \mathbb{R}^{m \times n}$, target rank $r$, sub-routine steps $t$
**Ensure:** Approx. rank-1 components $\{(\mathbf{a}_k, \mathbf{b}_k)\}_{k=1}^r$
1: $\mathbf{Y}_1 \leftarrow \mathbf{Y}$
2: **for** $k = 1$ **to** $r$ **do**
3: $\quad (\mathbf{a}_k, \mathbf{b}_k) \leftarrow \texttt{rank-1}(\mathbf{Y}_k, \mathbf{X}, t)$
4: $\quad \mathbf{Y}_{k+1} \leftarrow \mathbf{Y}_k - \mathbf{b}_k \mathbf{a}_k^\top \mathbf{X}$
5: **end for**
6: **return** $\{(\mathbf{a}_k, \mathbf{b}_k)\}_{k=1}^r$

---

Algorithm 1 aims to find exact low-rank subspaces by iteratively computing pairs of vectors $(\mathbf{a}_k^\star, \mathbf{b}_k^\star)$. It starts with the original output data $\mathbf{Y}$ as $\mathbf{Y}_1^\star$ (Line 1). In each iteration $k$, an optimization problem is solved to find the **best** pair $(\mathbf{a}_k^\star, \mathbf{b}_k^\star)$ that minimizes the Frobenius norm of the difference between the current matrix $\mathbf{Y}_k^\star$ and the rank-1 estimate $\mathbf{b}\mathbf{a}^\top \mathbf{X}$ (see Line 3)[1]. By applying the singular value decomposition (SVD) to $\mathbf{Y}$, we decompose it as

$$\mathbf{Y} = \sum_{k=1}^p \sigma_k^\star \mathbf{u}_k^\star \mathbf{v}_k^{\star\top}, \qquad \sigma_1^\star \geq \sigma_2^\star \geq \cdots \geq \sigma_p^\star > 0,$$

where $\sigma_k^\star$ are the singular values, and $\mathbf{u}_k^\star$ and $\mathbf{v}_k^\star$ are the left and right singular vectors, respectively. Notice that we denote $p = \operatorname{rank}(\mathbf{Y})$, but when executing Algorithm 1 and Algorithm 2 we may choose a target rank $r \neq p$.

**Lemma 1.** *Under our defined settings and deflation method, for each $k$, we have that $\mathbf{Y}_k^\star = \sum_{k'=k}^p \sigma_{k'}^\star \mathbf{u}_{k'}^\star \mathbf{v}_{k'}^{\star\top}$ and $\mathbf{b}_k^\star \mathbf{a}_k^{\star\top} \mathbf{X} = \sigma_k^\star \mathbf{u}_k^\star \mathbf{v}_k^{\star\top}$.*

The proof of Lemma 1 is provided in Appendix B.1. Namely, when $\mathbf{X}$ has full rank with $n \geq d$, $\mathbf{b}_k^\star$ and $\mathbf{a}_k^\star$ can be uniquely identified up to scalar multiplication.

After determining the pair $(\mathbf{a}_k^\star, \mathbf{b}_k^\star)$, the matrix $\mathbf{Y}_{k+1}^\star$ is updated by subtracting the rank-1 component $\mathbf{b}_k^\star \mathbf{a}_k^{\star\top} \mathbf{X}$ from $\mathbf{Y}_k^\star$ (see Line 4). This iterative process continues for $r$ iterations, generating $r$ pairs of vectors, which collectively represent the exact low-rank subspaces.

Algorithm 2 differs from Algorithm 1 in Lines 3 and 4. In Algorithm 2, Line 3 executes sub-routine for $t$ iterations, denoted by $\texttt{rank-1}(\mathbf{Y}_k, \mathbf{X}, t)$, to approximate the solution of equation 1 and return estimates $(\mathbf{a}_k, \mathbf{b}_k)$. The $t$ parameter represents the number of iterations for this approximate computation. An example of the $\texttt{rank-1}$ subroutine can be the gradient descent algorithm, which executes:

$$\mathbf{a}^{(t+1)} = \mathbf{a}^{(t)} - \eta_\mathbf{a} \mathbf{X} \left( \mathbf{b}^{(t)} \mathbf{a}^{(t)\top} \mathbf{X} - \mathbf{Y_k} \right)^\top \mathbf{b}^{(t)},$$

$$\mathbf{b}^{(t+1)} = \mathbf{b}^{(t)} - \eta_\mathbf{b} \left( \mathbf{b}^{(t)} \mathbf{a}^{(t)\top} \mathbf{X} - \mathbf{Y_k} \right) \mathbf{X}^\top \mathbf{a}^{(t)}. \tag{5}$$

Iterative algorithms such as (5) often produce numerical errors, leading to $\mathbf{b}_k \mathbf{a}_k^\top \neq \mathbf{b}_k^\star \mathbf{a}_k^{\star\top}$. Subsequently, this affects the quality of $\mathbf{Y}_{k+1}$ (Line 4), since the "deflation" step $\mathbf{Y}_k - \mathbf{b}_k \mathbf{a}_k^\top \mathbf{X}$ is based on the approximate matrix $\mathbf{Y}_k$ and the current estimates $\mathbf{b}_k \mathbf{a}_k^\top \mathbf{X}$, rather than the exact versions $\mathbf{Y}_k^\star$ and $\mathbf{b}_k^\star \mathbf{a}_k^{\star\top} \mathbf{X}$ used in Algorithm 1. To study the influence of the numerical errors produced by (5), we define:

---

[1]Note that, while $\sum_{k=1}^r \mathbf{b}_k^\star \mathbf{a}_k^\star = \mathbf{W}^\star$, $(\mathbf{b}_k^\star, \mathbf{a}_k^\star)$ not necessarily align with the $k$th left- and right- singular vectors of $\mathbf{W}^\star$.

Table 1: Rank-1 objects used at step $k$.

| Symbol | Meaning |
|---|---|
| $(\mathbf{a}_k^\star, \mathbf{b}_k^\star)$ | Exact (ideal) rank-1 pair |
| $(\overline{\mathbf{a}}_k, \overline{\mathbf{b}}_k)$ | Subroutine-optimal pair (minimizer of step-$k$ objective) |
| $(\mathbf{a}_k, \mathbf{b}_k)$ | Inexact pair returned by the finite-iteration routine |
| $\boldsymbol{\Psi}_k := \mathbf{b}_k \mathbf{a}_k^\top - \overline{\mathbf{b}}_k \overline{\mathbf{a}}_k^\top$ | Numerical error of the rank-1 step |
| $\mathbf{Y}_k$ | Residual output after removing first $(k-1)$ rank-1 components |

**Definition 1** (Numerical Error). *Let $(\overline{\mathbf{a}}_k, \overline{\mathbf{b}}_k)$ represents the exact rank-1 solution that approximates the processed label matrix $\mathbf{Y}_k$ using data $\mathbf{X}$:*

$$\overline{\mathbf{a}}_k, \overline{\mathbf{b}}_k = \operatorname*{argmin}_{\mathbf{a} \in \mathbb{R}^d, \mathbf{b} \in \mathbb{R}^m} \frac{1}{2} \left\| \mathbf{Y}_k - \mathbf{b}\mathbf{a}^\top \mathbf{X} \right\|_F^2 . \tag{6}$$

*and recall that $\mathbf{a}_k, \mathbf{b}_k$ are outputs of $\texttt{rank-1}\,(\mathbf{Y}_k, \mathbf{X}, T)$. We define the numerical errors incurred at iteration $k$ from the $\texttt{rank-1}$ sub-routine as:*

$$\boldsymbol{\Psi}_k := \mathbf{b}_k \mathbf{a}_k^\top - \overline{\mathbf{b}}_k \overline{\mathbf{a}}_k^\top . \tag{7}$$

For notational convenience, we set $\boldsymbol{\Psi}_0 := 0$ whenever later bounds use sums indexed from $k' = 0$. Notice that the definition of $\boldsymbol{\Psi}_k$ is based on $\mathbf{Y}_k$, not $\mathbf{Y}_k^\star$; recall that $\mathbf{Y}_k$ is constructed recursively using $\mathbf{b}_k \mathbf{a}_k^\top$. When $\mathbf{b}_k \mathbf{a}_k^\top$ is solved inexactly, we cannot guarantee that $\mathbf{Y}_k = \mathbf{Y}_k^\star$. Consequently, it is almost always the case that $(\overline{\mathbf{a}}_k, \overline{\mathbf{b}}_k) \neq (\mathbf{a}_k^\star, \mathbf{b}_k^\star)$, implying $\|\mathbf{b}_k^\star \mathbf{a}_k^{\star\top} - \overline{\mathbf{b}}_k \overline{\mathbf{a}}_k^\top\|_F > 0$. However, since the $\texttt{rank-1}$ subroutine only has access to $\mathbf{Y}_k$, its output $(\mathbf{a}_k, \mathbf{b}_k)$ converges to $(\overline{\mathbf{a}}_k, \overline{\mathbf{b}}_k)$ instead of $(\mathbf{a}_k^\star, \mathbf{b}_k^\star)$ as the number of iterations in the $\texttt{rank-1}$ subroutine increases.

**Related works.** Sequential low-rank subspace identification has roots in Principal Component Analysis (PCA) (Jolliffe, 1995; Golub & Van Loan, 2013) and has been explored via hierarchical game-theoretic models. In these models, rank-1 "players" learn distinct skills by optimizing a deflated objective (Gemp et al., 2021a;b; Hotelling, 1933; Mackey, 2008).

*Incremental learning of Eigenspaces.* (Arora et al., 2019) has shown that deep matrix factorization models implicitly discover low-rank components sequentially, an observation later extended to symmetric (Jin et al., 2023) and asymmetric (Soltanolkotabi et al., 2023) matrix sensing. This implicit recovery can be used for model compression (Kwon et al., 2024), but typically requires a deep model or specific initialization. In contrast, our work considers the simple model of low-rank linear regression by explicitly enforcing the sequential learning. While the work of (Wang et al., 2023b) is also similar, their design is more specific to matrix completion.

*Low-Rank Adapter (LoRA).* Our work connects to parameter-efficient fine-tuning (PEFT) methods like LoRA (Hu et al., 2021), particularly in continual learning where adapters are learned sequentially for new tasks, sometimes with orthogonality constraints to prevent catastrophic forgetting (Wistuba et al., 2023; Wang et al., 2023a). While theory shows LoRA can succeed with a sufficiently high rank (Zhang et al., 2023), the error propagation dynamics at lower ranks remain unexplored. Other recent work has focused on merging multiple LoRA adapters (Zhao et al., 2024b; Dimitriadis et al., 2024; Wu et al., 2024; Ostapenko et al.; Huh et al., 2024).

## 3 Error propagation during training

Recall that Lemma 1 guarantees that the rank-1 components given by the exact Algorithm 1 recovers the top-$r$ singular vector/values of $\mathbf{Y}^\star$. In this section, we study the recovery error under the inexact Algorithm 2. To effectively compare its output with Algorithm 1, we express these outputs in terms of the singular values and singular vectors of the deflated matrices $\mathbf{Y}_k^\star$ and $\mathbf{Y}_k$. We apply similar reasoning as in Lemma 1 for the term $\overline{\mathbf{b}}_k \overline{\mathbf{a}}_k^\top \mathbf{X}$ based on equation 6. To do so, we define $\sigma_{i_k}, \mathbf{u}_{i_k}$, and $\mathbf{v}_{i_k}$ as the $i$-th top singular value and vectors pairs of the matrix $\mathbf{Y}_k$. Note that $\sigma_{i_k} \neq \sigma_i^\star$ and $(\mathbf{u}_{i_k}, \mathbf{v}_{i_k}) \neq (\mathbf{u}_i^\star, \mathbf{v}_i^\star)$, for all $i$. Then, for each $k$, the

SVD on $\mathbf{Y}_k$ gives us: $\mathbf{Y}_k = \sum_{i=1}^{p-k+1} \sigma_{i_k} \mathbf{u}_{i_k} \mathbf{v}_{i_k}^\top$. Since $\bar{\mathbf{b}}_k \bar{\mathbf{a}}_k^\top \mathbf{X}$ is also rank-1, Eckart-Young-Mirsky theorem implies that it is the optimal rank-1 approximation of $\mathbf{Y}_k$ based on equation 6. Thus $\bar{\mathbf{b}}_k \bar{\mathbf{a}}_k^\top \mathbf{X} = \sigma_{1_k} \mathbf{u}_{1_k} \mathbf{v}_{1_k}^\top$, where $\sigma_{1_k}$, $\mathbf{u}_{1_k}$, and $\mathbf{v}_{1_k}$ correspond to the top singular value and singular vectors of $\mathbf{Y}_k$.

Recall that $\mathbf{u}_k^\star$ and $\mathbf{v}_k^\star$ are the top left and right singular vectors, respectively, of $\mathbf{Y}_k^\star$. Since singular vectors are unique only up to a sign, both $\mathrm{sv}_\mathrm{L}(\mathbf{Y}_k)$ and $-\mathrm{sv}_\mathrm{L}(\mathbf{Y}_k)$ are valid left singular vectors, and similarly, both $\mathrm{sv}_\mathrm{R}(\mathbf{Y}_k)$ and $-\mathrm{sv}_\mathrm{R}(\mathbf{Y}_k)$ are valid right singular vectors of $\mathbf{Y}_k$. We will choose $\mathbf{u}_{1_k}$ and $\mathbf{v}_{1_k}$ to be the ones such that $0 \leq \mathbf{v}_k^{\star\top} \mathbf{v}_{1_k}$ and $0 \leq \mathbf{u}_k^{\star\top} \mathbf{u}_{1_k}$:

$$\mathbf{u}_{1_k} := \mathrm{sv}_\mathrm{L}(\mathbf{Y}_k) \cdot \underset{s \in \{\pm 1\}}{\arg\min} \|s \cdot \mathrm{sv}_\mathrm{L}(\mathbf{Y}_k) - \mathbf{u}_k^\star\|_2; \quad \mathbf{v}_{1_k} := \mathrm{sv}_\mathrm{R}(\mathbf{Y}_k) \cdot \underset{s \in \{\pm 1\}}{\arg\min} \|s \cdot \mathrm{sv}_\mathrm{R}(\mathbf{Y}_k) - \mathbf{v}_k^\star\|_2.$$

We provide a characterization of the error propagation in the deflation methods in Algorithm 2 that is agnostic to the details of the sub-routine `rank-1`, i.e., when one only has knowledge about $\|\mathbf{\Psi}_k\|_F$. See Appendix A for the proof.

**Theorem 1.** *Let $\{(\mathbf{a}_k, \mathbf{b}_k)\}_{k=1}^r$ be the output of Algorithm 2. Let $\mathbf{\Psi}_k$ be given as in Definition 1 with $\|\mathbf{\Psi}_k\|_F > 0$. Let $\sigma_1^\star \geq \sigma_2^\star \geq \cdots \geq \sigma_p^\star > 0$ denote the singular values of $\mathbf{Y}$, where $p = \mathrm{rank}(\mathbf{Y})$, and adopt the convention $\sigma_{p+1}^\star := 0$. Define the singular value gap as $\mathcal{T}_k^\star := \sigma_k^\star - \sigma_{k+1}^\star$. Assume $\mathcal{T}_k^\star > 0$ for all $k \in [r]$. Define an error bound $E(k)$ as:*

$$E(k) := \sigma_{\max}(\mathbf{X}) \sum_{k'=0}^{k-1} \|\mathbf{\Psi}_{k'}\|_F \prod_{j=k'+1}^{k-1} \left(2 + \frac{6\sigma_j^\star}{\mathcal{T}_j^\star}\right).$$

*If $E(k) < \frac{1}{2}\mathcal{T}_k^\star$ for all $k \in [r]$, then the output of Algorithm 2 satisfies:*

$$\left\|\mathbf{Y} - \sum_{k=1}^r \mathbf{b}_k \mathbf{a}_k^\top \mathbf{X}\right\|_F \leq \left(\sum_{k=r+1}^p (\sigma_k^\star)^2\right)^{1/2} + \sigma_{\max}(\mathbf{X}) \sum_{k=1}^r \sum_{k'=0}^k \|\mathbf{\Psi}_{k'}\|_F \prod_{j=k'+1}^k \left(2 + \frac{6\sigma_j^\star}{\mathcal{T}_j^\star}\right) \tag{8}$$

Theorem 1 characterizes how errors from approximately solving the rank-1 subroutine propagate through the deflation procedure in sequential low-rank approximations. The theorem asserts that, as long as each error $\mathbf{\Psi}_k$ is sufficiently small, the compounded effect of errors across the sequence remains bounded, thereby preserving the accuracy of the final low-rank approximation. The first term is the exact Frobenius residual of the rank-$r$ truncated SVD of $\mathbf{Y}$. The smallness condition on $E(k)$ is exactly what ensures that the perturbation argument used in the proof is valid; see Appendix B.4, especially Eqs. (32)–(33).

**Remark 2.** *The error bound in Theorem 1 reflects sensitivity to the singular spectrum of the target matrix $\mathbf{Y}$. The second term in (8) is the numerical-error contribution, whereas the truncation term $\left(\sum_{k=r+1}^p (\sigma_k^\star)^2\right)^{1/2}$ is independent of $\mathbf{\Psi}_k$. For $1 \leq \ell \leq k \leq r$, define*

$$P_{\ell,k} := \prod_{j=\ell+1}^k \left(2 + \frac{6\sigma_j^\star}{\mathcal{T}_j^\star}\right), \qquad P_{\ell,\ell} := 1,$$

*The product is stage-wise over the intermediate indices $j$: each intervening step contributes its own local gap $\mathcal{T}_j^\star$. Let $A_\ell := \sum_{k=\ell}^r P_{\ell,k}$. A slightly stronger but cleaner sufficient condition for the numerical-error term in (8) to remain $O(1)$ is*

$$\|\mathbf{\Psi}_\ell\|_F \leq \frac{c}{r\,\sigma_{\max}(\mathbf{X})} A_\ell^{-1}, \qquad \forall \ell \in [r],$$

*for an absolute constant $c > 0$. Thus, each stage error must decay like the inverse of its total downstream amplification. In particular, earlier components must be solved more accurately than later ones, especially when the spectral gaps are small. While we do not claim this condition to be tight, it is a simple benign regime extracted from the upper bound. The same interpretation applies to the corresponding product terms in Theorems 5, 6, and 8; in Theorems 5 and 6 the front factor is $\kappa(\mathbf{X})$ rather than $\sigma_{\max}(\mathbf{X})$.*

**Remark 3.** *Our proof relies on a recursive bound on the errors of $\|\mathbf{b}_k \mathbf{a}_k^\top \mathbf{X} - \mathbf{b}_k^\star \mathbf{a}_k^{\star\top} \mathbf{X}\|_F$, which requires a strict ordering of the relevant singular values of $\mathbf{Y}$. In the case where $\mathcal{T}_k^\star = 0$, the singular value $\sigma_k^\star$ is not separated from the remaining spectrum, so the corresponding rank-1 component is not uniquely identifiable. In that case, the error $\|\mathbf{b}_k \mathbf{a}_k^\top \mathbf{X} - \mathbf{b}_k^\star \mathbf{a}_k^{\star\top} \mathbf{X}\|_F$ is not well defined as a component-wise target quantity. Thus, we require $\mathcal{T}_k^\star > 0$ in our analysis. The case $\mathcal{T}_k^\star = 0$ could be handled by tracking errors at the level of invariant subspaces corresponding to repeated singular values rather than individual singular vectors. We leave this extension for future work.*

**Remark 4.** *We do not claim our upper bound in Theorem 1 to be tight. Rather, Theorem 1 serves as an upper bound that explicitly relates the loss $\left\|\mathbf{Y} - \sum_{k=1}^r \mathbf{b}_k \mathbf{a}_k^\top \mathbf{X}\right\|_F$ to the numerical errors $\mathbf{\Psi}_k$ in each step. It is possible that, under additional assumptions of the rank-1 algorithms and the structural information of the data, we are able to obtain a tighter bound. However, this direction is beyond the scope of our work, as this work is considered the first such attempt, to our knowledge.*

## 4 Parameter recovery under fixed design

Thus far, we have been focusing on constructing components $\{(\mathbf{a}_k, \mathbf{b}_k)\}_{k=1}^r$ such that $\sum_{k=1}^r \mathbf{b}_k \mathbf{a}_k^\top \mathbf{X}$ estimates $\mathbf{Y}$. Given that $\mathbf{X}$ and $\mathbf{Y}$ are considered as training data, the previous section characterized the training error of Algorithm 2. Here, we study the parameter-recovery behavior of Algorithm 2, assuming that the data is generated based on some optimal parameter $\mathbf{W}^\star$ with $\text{rank}(\mathbf{W}^\star) = r^\star$:

$$\mathbf{Y} = \mathbf{Y}^\star + \mathcal{E}; \quad \mathbf{Y}^\star = \mathbf{W}^\star \mathbf{X}, \tag{9}$$

where $\mathcal{E} \in \mathbb{R}^{m \times n}$ denotes an additive label noise generated from a certain distribution, and $\mathbf{Y}^\star$ denotes the noiseless label. Throughout this section, $\mathbf{X}$ is treated as a fixed design matrix. Accordingly, Theorems 5–6 are parameter-recovery results, rather than out-of-sample prediction risk on a separate test distribution. In the noiseless case where $\mathcal{E} = \mathbf{0}$, we have that $p = \text{rank}(\mathbf{Y}) = r^\star$ and, Algorithm 1 can recover $\{(\mathbf{a}_k^\star, \mathbf{b}_k^\star)\}_{k=1}^r$ such that $\mathbf{W}^\star = \sum_{k=1}^r \mathbf{b}_k^\star \mathbf{a}_k^{\star\top}$ when $r = r^\star$. However, the exact recovered pair $(\mathbf{a}_k^\star, \mathbf{b}_k^\star)$ need not correspond to the $k$th singular component of $\mathbf{W}^\star$, because it is induced by the fixed design matrix $\mathbf{X}$. In other words, each pair $(\mathbf{a}_k^\star, \mathbf{b}_k^\star)$ contains a component of $\mathbf{W}^\star$ that *extracts a certain information from the input data* $\mathbf{X}$. When $\mathcal{E} \neq \mathbf{0}$, it is possible that $p = \text{rank}(\mathbf{Y}) > r^\star$.

**Parameter recovery under noiseless labels.** As a warm up, we consider the case where the noise $\mathcal{E} = \mathbf{0}$. Intuitively, when $\mathbf{X}$ is full rank, a zero training loss would imply a perfect recovery of the optimal parameter $\mathbf{W}^\star$. We state a more general result below covering the case of non-zero training loss with component-wise parameter-recovery error.

**Theorem 5.** *Let $\{(\mathbf{a}_k, \mathbf{b}_k)\}_{k=1}^r$ be the output of Algorithm 2. Let $\mathbf{\Psi}_k$ be given as in Definition 1 with $\|\mathbf{\Psi}_k\|_F > 0$. Let $\sigma_1^\star \geq \sigma_2^\star \geq \cdots \geq \sigma_p^\star > 0$ denote the singular values of $\mathbf{Y}$, where $p = \text{rank}(\mathbf{Y})$, and adopt the convention $\sigma_{p+1}^\star := 0$. Define the singular value gap as $\mathcal{T}_k^\star := \sigma_k^\star - \sigma_{k+1}^\star$. Assume $\mathcal{T}_k^\star > 0$ for all $k \in [r]$. Also, define an error bound $E(k)$ as:*

$$E(k) := \sigma_{\max}(\mathbf{X}) \sum_{k'=0}^{k-1} \|\mathbf{\Psi}_{k'}\|_F \prod_{j=k'+1}^{k-1} \left(2 + \frac{6\sigma_j^\star}{\mathcal{T}_j^\star}\right).$$

*If $E(k) < \frac{1}{2}\mathcal{T}_k^\star$ for all $k \in [r]$, and $\sigma_{\min}(\mathbf{X}) > 0$, then the output of Algorithm 2 satisfies:*

$$\left\|\mathbf{b}_k^\star \mathbf{a}_k^{\star\top} - \mathbf{b}_k \mathbf{a}_k^\top\right\|_F \leq \kappa(\mathbf{X}) \sum_{k'=0}^k \|\mathbf{\Psi}_{k'}\|_F \prod_{j=k'+1}^k \left(2 + \frac{6\sigma_j^\star}{\mathcal{T}_j^\star}\right); \quad \forall k \in [r]. \tag{10}$$

*Moreover, the aggregation of the components $(\mathbf{a}_k, \mathbf{b}_k)$'s approximates $\mathbf{W}^\star$ as*

$$\left\|\mathbf{W}^\star - \sum_{k=1}^r \mathbf{b}_k \mathbf{a}_k^\top\right\|_F \leq \frac{\left(\sum_{k=r+1}^{r^\star} (\sigma_k^\star)^2\right)^{1/2}}{\sigma_{\min}(\mathbf{X})} + \kappa(\mathbf{X}) \sum_{k=1}^r \sum_{k'=1}^k \|\mathbf{\Psi}_{k'}\|_F \prod_{j=k'+1}^k \left(2 + \frac{6\sigma_j^\star}{\mathcal{T}_j^\star}\right). \tag{11}$$

*Here, $\kappa(\mathbf{X}) = \frac{\sigma_{\max}(\mathbf{X})}{\sigma_{\min}(\mathbf{X})}$ denotes the condition number of $\mathbf{X}$.*

The proof of Theorem 5 is provided in Appendix C. In particular, Theorem 5 states two results. First, (10) measures how the errors of individual components are influenced even by the numerical errors that appear when solving previous components. This bound illustrates key factors contributing to the error at each iteration. As we discussed previously, $(\mathbf{a}_k^\star, \mathbf{b}_k^\star)$ can be considered as the components of $\mathbf{W}^\star$ extracted based on the importance defined by the input data $\mathbf{X}$. From this perspective, (10) shows how well these data-dependent components are approximated by Algorithm 2. Moreover, (11) measures how well the inexact method approximates $\mathbf{W}^\star$, including errors due to inexact computations and limitations of representing $\mathbf{W}^\star$ with rank $r$. This bound sheds light on how the components $(\mathbf{a}_k, \mathbf{b}_k)$'s collaboratively contribute to the overall parameter recovery of $\mathbf{W}^\star$ under fixed design.

**Parameter recovery under noisy labels.** In the previous section, we studied the parameter-recovery behavior of Algorithm 2 under a noiseless scenario with $\mathbf{Y} = \mathbf{W}^\star \mathbf{X}$, where the algorithmic choice of choosing $r = \mathrm{rank}(\mathbf{Y})$ can be shown to be optimal. However, this argument may not hold when the labels are generated with a non-zero additive noise. In this section, we consider the noise matrix $\boldsymbol{\mathcal{E}}$ to consist of I.I.D. entries $\boldsymbol{\mathcal{E}}_{ij} \sim \mathcal{N}(0, \varepsilon^2)$, where $\varepsilon$ controls the magnitude of the noise. In this case, with a high probability, we have $p = \mathrm{rank}(\mathbf{Y}) = m$. Let $\{(\mathbf{a}_k, \mathbf{b}_k)\}_{k=1}^r$ be the recovery result according to Algorithm 2. Then we are interested in an upper bound on $\|\mathbf{W}^\star - \sum_{k=1}^r \mathbf{b}_k \mathbf{a}_k\|_F$. In particular, we have the following guarantee on the parameter-recovery error.

**Theorem 6.** *Consider the scenario of finding the top-r rank-1 subspaces that minimize the loss in equation 1. Let $\mathcal{T}_k^\star := \sigma_k^\star - \sigma_{k+1}^\star$, and $\mathcal{T}_{\min}^\star := \min_{k \in [1,r]} \mathcal{T}_k^\star$. Assume $\mathcal{T}_k^\star > 0$ for all $k \in [r]$. Assume further that the noise scale satisfies the ordering condition*

$$\varepsilon \le c_{\mathrm{ord}} \frac{\mathcal{T}_{\min}^\star}{\sqrt{n} + \sqrt{\log(1/\gamma)}}$$

*for a sufficiently small absolute constant $c_{\mathrm{ord}} > 0$. Then, with probability at least $1 - \gamma$, the output of Algorithm 2 satisfies:*

$$\left\| \mathbf{W}^\star - \sum_{k=1}^r \mathbf{b}_k \mathbf{a}_k^\top \right\|_F \le \kappa(\mathbf{X}) \left( \left( \sum_{k=r+1}^{r^\star} \sigma_k(\mathbf{W}^\star)^2 \right)^{1/2} + \sum_{k=1}^r \sum_{k'=0}^k \|\boldsymbol{\Psi}_{k'}\|_F \prod_{j=k'+1}^k \left( 2 + \frac{6\sigma_j^\star}{\mathcal{T}_j^\star} \right) \right) \\ + O\left( \frac{\varepsilon \sqrt{n \log 1/\gamma}}{\sigma_{\min}(\mathbf{X})} \left( r + \sqrt{\frac{\min\{r^\star, r\}}{\mathcal{T}_{\min}^\star}} \right) \right) \tag{12}$$

**Remark 7.** *The small-noise condition in Theorem 6 is an ordering condition. It ensures that the perturbation induced by the additive noise is small relative to the minimum singular-value gap $\mathcal{T}_{\min}^\star$, so that the ordering of the relevant rank-1 components is preserved after noise is added. Equivalently, the relevant noisy components can still be matched component-wise with the corresponding noiseless components. This is the regime required by the perturbation argument used in the proof of Theorem 6.*

*The second term in equation 12 is the contribution of the additive noise and scales as*

$$O\left( \frac{\varepsilon \sqrt{n \log(1/\gamma)}}{\sigma_{\min}(\mathbf{X})} \left( r + \sqrt{\frac{\min\{r^\star, r\}}{\mathcal{T}_{\min}^\star}} \right) \right).$$

*Hence, if one additionally wants this noise contribution to remain $O(1)$, it suffices to require*

$$\varepsilon \le \frac{c_{\mathrm{noise}}\, \sigma_{\min}(\mathbf{X})}{\sqrt{n \log(1/\gamma)} \left( r + \sqrt{\frac{\min\{r^\star, r\}}{\mathcal{T}_{\min}^\star}} \right)},$$

*for a sufficiently small absolute constant $c_{\mathrm{noise}} > 0$. This is stronger than the ordering condition stated in Theorem 6. Without this stronger condition, the second term in equation 12 should be interpreted as an explicit scaling law in $\varepsilon$, $r$, $\mathcal{T}_{\min}^\star$, and $\sigma_{\min}(\mathbf{X})$, rather than as a constant-order bound. In particular, increasing $r$ can reduce the truncation bias in the first term while increasing the noise contribution in the second term, which is exactly the bias-variance trade-off highlighted by Theorem 6.*

The proof of Theorem 6 is given in Appendix C.3. In particular, Theorem 6 characterizes how Algorithm 2 recovers the target parameter $\mathbf{W}^\star$ under label noise in the fixed-design setting. The first term in the upper bound of (12) contains both the truncation error from restricting the recovery to rank $r$ and the propagated numerical errors from the inexact `rank-1` subroutine. The second term demonstrates the influence of the additive noise $\mathcal{E}$ on the parameter-recovery error. To start, a larger noise scale $\epsilon$ implies a worse parameter-recovery error. Moreover, it should be noticed that a good choice of $r$ can greatly impact the parameter-recovery error as well: choosing $r < r^\star$ can result in a larger error in the first term due to the incomplete estimation of the components in $\mathbf{W}^\star$. On the other hand, since the second term scales with $r$, choosing a larger $r$ can result in a larger error caused by the noise. This is the scenario where the noise is overfitted by increasing the complexity of the model. From this perspective, Theorem 6 characterizes a bias-variance-type trade-off in fixed-design parameter recovery. See Remark 7 for the role of the small-noise assumption in Theorem 6 and for a stronger sufficient condition under which the additive-noise contribution in (12) remains $O(1)$.

## 5 Experiments

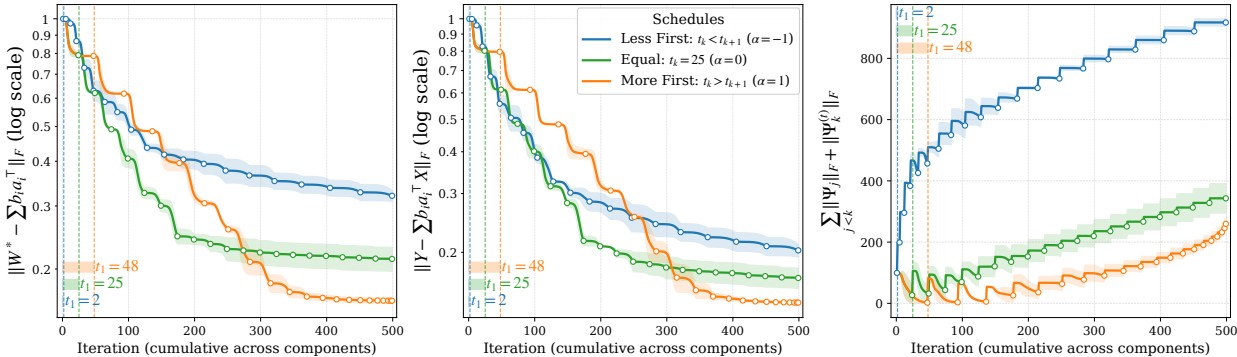

Figure 1: **Impact of per-component iteration-allocation strategies in the synthetic linear setting under a fixed total budget $T$.** Open-circle markers denote component-switch points, and longer horizontal segments on the x-axis indicate more iterations allocated to that component. We compare three representative schedules from the family $t(\alpha)$: Less First, Equal, and More First. Left: reconstruction error. Middle: training objective error. Right: cumulative numerical-error proxy built from the $\|\mathbf{\Psi}_k\|_F$ terms in Theorems 1 and 5, where $k$ is the currently optimized component at cumulative iteration $t$. Vertical dashed lines mark the first-component budget $t_1$. More-first schedules yield the smallest accumulated numerical error and the best final reconstruction and training performance. Exact budget allocation details are given in Appendix G.

We present two types of experiments. Section 5.1 (together with Appendix G) validates the theoretical predictions in the linear low-rank regression setting studied in Sections 3–4. Sections 5.2–5.3 then extend the study to simple nonlinear LoRA-based classification settings and examine whether related qualitative scheduling patterns—especially the importance of accurately learning early components—also appear in practical PEFT settings. Because these latter settings lie outside the scope of the regression theorems, we present them as exploratory qualitative evidence rather than as direct theorem validations or superiority claims over one-shot training; the linear theory should therefore not be read as directly predicting nonlinear LoRA behavior. To compare schedule allocation consistently across all experiments, we next introduce a common one-parameter schedule family.

**Common schedule family.** To study schedule effects in a consistent way, we parameterize per-component budgets by a single trend variable $\alpha$. Positive $\alpha$ shifts a fixed total budget toward earlier components, negative $\alpha$ shifts it toward later components, and $\alpha = 0$ yields equal allocation. Given target rank $r$, total budget $T$,

minimum floor $m \geq 1$, and trend parameter $\alpha \in \mathbb{R}$, define

$$x_k = 1 - \frac{2(k-1)}{r-1}, \qquad w_k(\alpha) = \frac{\max(1 + \alpha x_k, 10^{-9})}{\sum_{j=1}^r \max(1 + \alpha x_j, 10^{-9})}, \qquad k = 1, \ldots, r.$$

We then allocate

$$e_k = (T - rm)w_k(\alpha), \qquad t_k = m + \lfloor e_k \rfloor + \mathbf{1}[k \in \mathcal{R}],$$

where $\mathcal{R}$ contains the indices with the largest fractional parts $e_k - \lfloor e_k \rfloor$, chosen so that $\sum_{k=1}^r t_k = T$. Thus, $\alpha = 0$ gives the equal schedule, $\alpha > 0$ gives a more-first schedule, and $\alpha < 0$ gives a less-first schedule. Larger positive values of $\alpha$ yield a stronger more-first trend in the realized schedule, typically assigning more budget to earlier components, while larger negative values yield the symmetric less-first trend. The schedules for $\pm\alpha$ are time reversals of one another. We use this same family in Sections 5.1–5.3; each experiment fixes its own $(r, T, m)$ and then varies $\alpha$. Appendix G gives a concrete Figure 1 instantiation and the resulting realized schedules.

### 5.1 Synthetic validation of theoretical setting.

Our synthetic experiments study how the distribution of a fixed computational budget across sequential rank-1 components affects the final approximation quality, especially through error propagation from early components to later ones. Following the general set-up in (9), we use the common schedule family above to vary this allocation through the trend parameter $\alpha$. The three named schedules in Figure 1 are representative members of the common family above: Less First corresponds to a back-loaded schedule ($\alpha < 0$), Equal to $\alpha = 0$, and More First to a front-loaded schedule ($\alpha > 0$). Beyond these three illustrative cases, in Figure 2, we also sweep $\alpha$ over a finer grid and report the resulting final errors, so the reader can see where more-first schedule helps, and where it saturates. For Figure 1, we instantiate this family with $(r, T, m) = (20, 500, 2)$ and $\alpha \in \{-1, 0, 1\}$; Figure 2 uses the same $(r, T, m)$ and sweeps $\alpha$ over a finer grid.

Our analysis dictates that errors in early components propagate to later components, suggesting that allocating more iterations to earlier components leads to better overall performance. Figure 1 shows the reconstruction error, training objective error, and cumulative numerical error for the three allocation strategies under a fixed computational budget. The right panel does not plot the full theorem bound; rather, it isolates a cumulative proxy based on the $\|\mathbf{\Psi}_k\|_F$ terms, which are the schedule-dependent numerical errors driving the bounds in Theorems 1 and 5.

The results confirm our theoretical predictions. Allocating more iterations to earlier components leads to better final reconstruction and training errors compared to allocating fewer iterations initially. The equal iteration strategy performs better than the "less first iterations" approach, but worse than the "more first iterations" strategy. This validates our theoretical finding that errors in early components propagate and compound through the sequential process. This cascading effect means that if early components are poorly approximated, their errors get magnified in subsequent components. The right panel makes this connection explicit by showing that the more-first schedule keeps the accumulated numerical error smallest throughout the sequence, followed by equal and then less-first. Appendix G records the exact realized schedules used in Figure 1 and provides additional synthetic-setting details. To move beyond these three anchor schedules, we next perform a finer-grained sweep over the one-parameter schedule family defined above, while keeping the same synthetic setting and the same total iteration budget $T = 500$.

Figure 2 refines the three-way comparison in Figure 1 by sweeping continuously over the schedule family $t(\alpha)$. In both panels, $\Delta$ denotes the difference between the final value under schedule $t(\alpha)$ and the corresponding final value under the equal schedule, so $\Delta < 0$ indicates an improvement over equal allocation. The main pattern is asymmetric: as $\alpha$ increases from 0, allocating more iterations to earlier components reduces both the final reconstruction error and the final accumulated numerical-error proxy relative to the equal schedule. These gains then flatten once the first component already receives a sufficiently large budget, indicating saturation rather than indefinite monotone improvement. In contrast, negative $\alpha$ values—which back-load the budget onto later components—lead to a sharp degradation in both quantities. Thus, in the linear synthetic setting, the finer sweep shows where front-loading helps and where the benefit saturates. Appendix G further evaluates the theorem-aligned dependencies directly by measuring the cumulative nu-

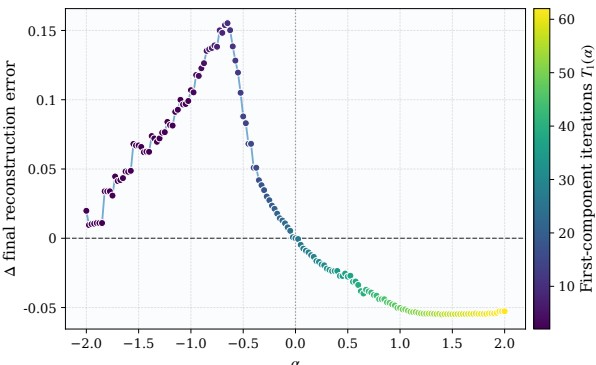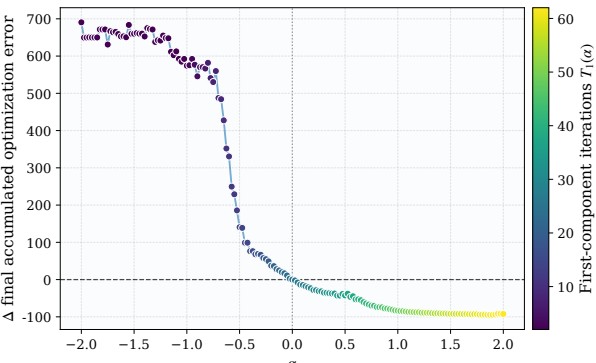

Figure 2: **Finer-grained schedule sweep in the synthetic linear setting of Figure 1.** Each point is a realized schedule $t(\alpha)$ under the same $(r, T, m) = (20, 500, 2)$ setting. The horizontal axis runs from less-first ($\alpha < 0$) through equal ($\alpha = 0$) to more-first ($\alpha > 0$) schedules; the color bar demonstrates the first component budget allocation $T_1(\alpha)$. Both vertical axes plot differences relative to the equal schedule, so $\Delta < 0$ means that the corresponding schedule achieved a smaller final value of the plotted quantity than the equal schedule. **Left:** $\Delta$ final reconstruction error. **Right:** $\Delta$ final accumulated numerical error. More-first ($\alpha > 0$) schedules consistently achieve better performance, with gains plateauing around $\alpha = 1.5$, while less-first schedules ($\alpha < 0$ reduce performance.

merical error $\sum_k \|\boldsymbol{\Psi}_k\|_F$, the stage-wise gaps $\mathcal{T}_k^\star$, and the resulting evaluated right-hand side of the bound across schedules and spectral profiles (Figs. 16–17).

## 5.2 Experimental analysis using LoRA on simple vision tasks.

We evaluate our sequential rank-1 approach in LoRA adaptation on three standard image classification datasets: MNIST, CIFAR10, and CIFAR100. These experiments use classification loss and nonlinear networks, so they are not direct instantiations of the regression theory in Sections 3–4; rather, they test whether a related qualitative scheduling intuition appears in a simple PEFT setting. We design these experiments to provide simple PEFT testbeds with different baseline strengths. *The goal is **not** to claim state-of-the-art performance or realistic deployment scenarios; rather, it is to examine how the qualitative scheduling picture changes in classification settings that lie outside the scope of our theory. The use of a feedforward network is therefore intentional.*

**Mathematical formulation.** To make the connection to the sequential framework above explicit, we first state the non-linear LoRA objective used in these experiments. This is the non-linear analogue of the sequential rank-1 updates studied in the linear setting. In standard LoRA, we parameterize the weight change during fine-tuning as a low-rank decomposition: $\Delta\mathbf{W} = \mathbf{B}\mathbf{A}^\top \in \mathbb{R}^{m \times n}$, where $\mathbf{A} \in \mathbb{R}^{n \times r}$ and $\mathbf{B} \in \mathbb{R}^{m \times r}$ with $r \ll \min(m, n)$. In these experiments and w.l.o.g., $r = 3$. In our approach, instead of optimizing all $r$ components simultaneously, we optimize one rank-1 component at a time, using the residual error from previous components to guide each subsequent step.

In particular, let $\mathbf{W}_0 \in \mathbb{R}^{m \times n}$ be a pre-trained weight matrix (in our case, we have $\mathbf{W}_0$ for every layer of the pretrained fully-connected network). Let $\mathcal{L}$ be the task-specific loss function, $f$ is the network function, and $(\mathbf{x}, \mathbf{y})$ represents the task data. Then,

$$\Delta\mathbf{W} = \underset{\Delta\hat{\mathbf{W}} \in \mathbb{R}^{m \times n}, \text{rank}(\Delta\hat{\mathbf{W}}) = r}{\arg\min} \mathcal{L}(f(\mathbf{x}; \mathbf{W}_0 + \Delta\hat{\mathbf{W}}), \mathbf{y}). \tag{13}$$

We define our sequential rank-1 adaptation procedure as follows: For $k = 1, 2, \ldots, r$, find the rank-1 update that minimizes the task loss given the previously learned components:

$$\mathbf{a}_k, \mathbf{b}_k = \underset{\mathbf{a} \in \mathbb{R}^n, \mathbf{b} \in \mathbb{R}^m}{\arg\min} \mathcal{L}\left(f\left(\mathbf{x}; \mathbf{W}_0 + \sum_{j=1}^{k-1} \mathbf{b}_j \mathbf{a}_j^\top + \mathbf{b}\mathbf{a}^\top\right), \mathbf{y}\right) \tag{14}$$

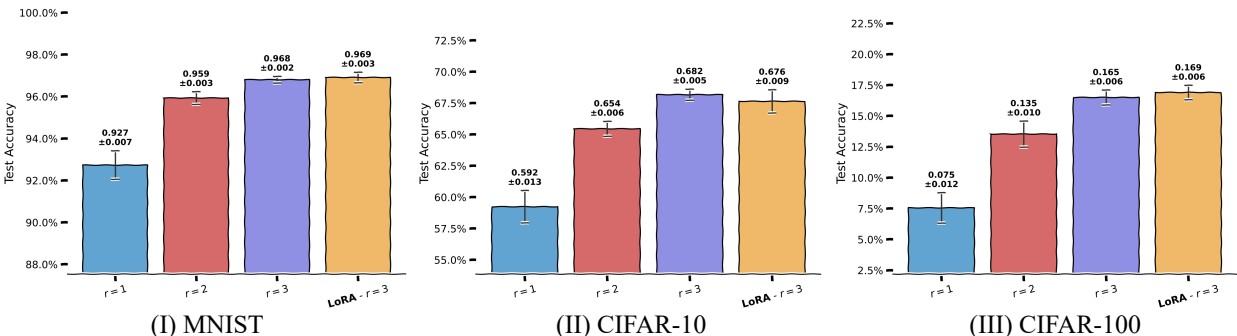

Figure 3: **Test accuracy on the new classes after sequential rank-1 LoRA adaptation across MNIST, CIFAR10, and CIFAR100.** Bars $r = 1, 2, 3$ show the sequential model after adding one, two, and three rank-1 components, and LoRA-$r = 3$ denotes the jointly trained rank-3 baseline. Accuracy improves as more sequential components are added, with larger gains when the starting model is weaker. The final sequential rank-3 model lies in a similar accuracy range to jointly trained rank-3 LoRA while using similar adaptation compute (within 5% of total FLOPs). We report mean performance and standard deviation over 5 runs. Broader compute-matched comparisons across representative schedule choices are shown in Fig. 8 and the per-sample FLOP accounting used for these comparisons is shown in Fig. 9.

The final adapted model uses the weight matrix on the new data domain: $\mathbf{W} = \mathbf{W}_0 + \sum_{k=1}^{r} \mathbf{b}_k \mathbf{a}_k^\top$. We approximate the optimal rank-1 updates using (stochastic) gradient descent on (14).

**Experimental setting.** We employ a simple feedforward network as our base architecture across all experiments. For each dataset, we first train the baseline model on a subset of classes (in particular, the first half of available classes). We then apply our sequential rank-1 LoRA adaptation approach to handle the remaining classes for 3 sequential rank-1 trainings, i.e., $r = 3$.

Our architecture consists of three fully-connected layers that map flattened input images to class logits. As usual, for MNIST, inputs are 784-dimensional (28×28 grayscale images), while for CIFAR10 and CIFAR100, inputs are 3072-dimensional (32×32×3 RGB images). Hidden layers have 512 units with ReLU activations, and the output layer dimension matches the number of classes in each dataset. We analyze three distinct scenarios:

1. **MNIST (Strong Baseline)**: The baseline network achieves high accuracy (∼98%) on classes 0-4, providing a strong foundation for adaptation on the remaining 5-9 classes.

2. **CIFAR10 (Moderate Baseline)**: The baseline network reaches moderate accuracy (∼40%) on classes 0-4, representing a partially optimized model (reminder that the model is not a CNN-based model but just a FF connected network).

3. **CIFAR100 (Weak Baseline)**: The baseline network attains lower accuracy (∼20%) on classes 0-49, exemplifying a relatively poor initial representation, where LoRA models adapt over the remaining 50-99 classes.

All experiments were conducted using Google Colab Pro+ with NVIDIA A100 GPU (40GB memory).

**Adaptation performance across datasets.** We report these results only as exploratory qualitative evidence in this simple PEFT setting, not as a validation of the regression theorems. Figure 3 presents the test accuracy of sequential rank-1 LoRA components when adapting to new classes across the three datasets. The accuracy is measured solely on the new classes (classes "5-9" for MNIST and CIFAR10, classes "50-99" for CIFAR100), highlighting the adaptation capabilities rather than overall performance. Figure 3 reports mean test accuracy over 5 runs; error bars and the values above the bars denote standard deviation across runs. Across the 5 runs, the qualitative conclusions remained consistent: adding sequential components improved accuracy, and the final sequential rank-3 model remained in a similar accuracy range to jointly trained rank-3 LoRA.

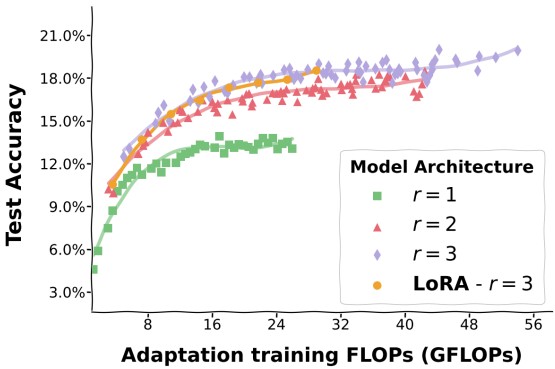

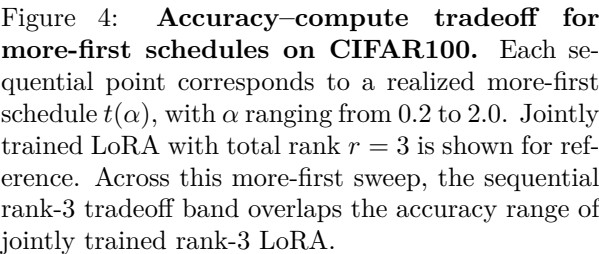

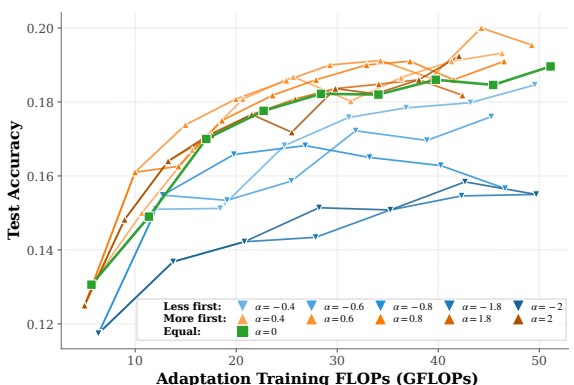

Figure 4: **Accuracy–compute tradeoff for more-first schedules on CIFAR100.** Each sequential point corresponds to a realized more-first schedule $t(\alpha)$, with $\alpha$ ranging from 0.2 to 2.0. Jointly trained LoRA with total rank $r = 3$ is shown for reference. Across this more-first sweep, the sequential rank-3 tradeoff band overlaps the accuracy range of jointly trained rank-3 LoRA.

Figure 5: **Accuracy–compute tradeoff under the $\alpha$-schedule family for CIFAR100.** Each curve traces a realized schedule $t(\alpha)$ with the indicated value of $\alpha$: $\alpha > 0$ denotes more-first schedules, $\alpha = 0$ the equal schedule, and $\alpha < 0$ less-first schedules. Over much of the plotted compute range, more-first schedules occupy a higher accuracy range than less-first schedules, with the equal schedule typically in between.

Our results show that sequential LoRA adaptation transfers knowledge across all three scenarios, with the gains depending on the quality of the baseline model. In all cases, the final sequential rank-3 model lies in a similar accuracy range to jointly trained rank-3 LoRA while building adaptation capacity one component at a time. To make the fairness criterion explicit, Appendix F reports both the per-sample FLOP accounting used in this comparison (Figure 9) and the resulting compute-matched comparison between sequential and jointly trained LoRA across representative schedule choices (Figure 8). In that compute-matched view, sequential LoRA remains in a similar accuracy range to joint LoRA for the equal and moderately more-first schedules, while less-first schedules are less favorable; at some matched-compute points, the moderately more-first curves in Appendix Fig. 8 also lie slightly above the joint-LoRA curve. We view these LoRA comparisons as exploratory qualitative evidence in this simple PEFT setting.

**Accuracy–compute tradeoff under more-first schedules.** Figure 4 complements Figure 3 by giving a compute-aware view of sequential LoRA in the CIFAR100 setting within the more-first branch of the common schedule family. Here, $r = 1, 2, 3$ denote sequential models obtained after adding one, two, and three rank-1 components, respectively. Each sequential point corresponds to a realized more-first schedule $t(\alpha)$ with $\alpha \in [0.2, 2.0]$, plotted at its corresponding adaptation-training FLOPs, while jointly trained rank-3 LoRA with total rank $r = 3$ is shown for reference. Across this sweep, the sequential rank-3 models occupy a similar accuracy range to jointly trained rank-3 LoRA, while ranks 1 and 2 provide lower-compute operating points at lower accuracy. Because only more-first schedules are shown here, Figure 4 should be read as characterizing the attainable accuracy–compute tradeoff within this branch; Figure 5 then broadens the comparison to equal and less-first schedules. Practically, this highlights the incremental-rank nature of the sequential procedure: one can stop at a lower rank if the resulting accuracy is already sufficient, rather than fixing the final rank in advance.

To study schedule effects more directly, we next fix the dataset and vary the schedule family itself rather than only the number of sequential components.

**Accuracy–compute tradeoff under the $\alpha$-schedule family.** Figure 5 shows how test accuracy changes with adaptation-training FLOPs when we vary the schedule parameter $\alpha$ in the CIFAR100 setting. The point of this figure is to separate the effect of *how* compute is distributed across the three sequential components from the effect of total compute alone. If total compute were the only important factor, then schedules with similar FLOPs would behave similarly; instead, the figure shows that schedule choice also matters. In

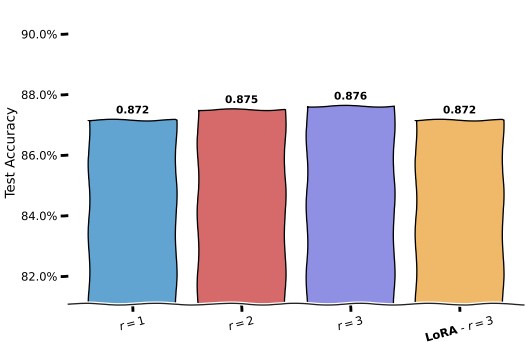

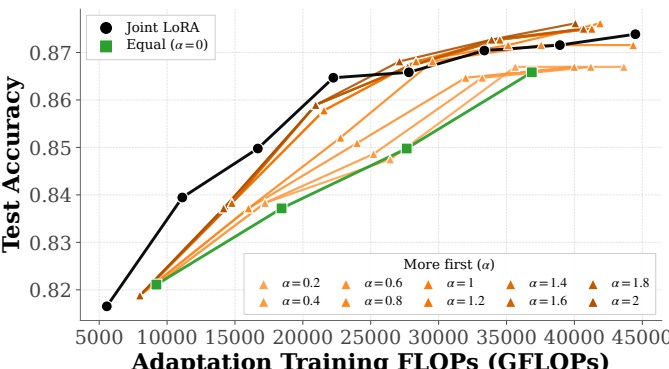

Figure 6: **SST-2 final accuracy after sequentially adding rank-1 components.** The first three bars show the sequential model after one, two, and three rank-1 components have been added; the fourth bar is the jointly trained rank-3 LoRA reference. The first sequential component already reaches the displayed joint-reference accuracy, and the second and third components provide modest additional gains, placing the final sequential model and the joint reference in the same narrow accuracy range for this setup.

Figure 7: **SST-2 accuracy–compute tradeoff under more-first schedules.** Each orange trajectory corresponds to a realized more-first schedule $t(\alpha)$ for sequential rank-3 adaptation; the specific $\alpha$ values are listed in the legend. Points along a trajectory track the accuracy reached as adaptation-training compute accumulates under that schedule. The black curve is the jointly trained rank-3 LoRA reference and the green curve is the equal-schedule baseline. Across much of the plotted range, many more-first schedules lie above the equal schedule, while the strongest more-first runs reach the same narrow top-accuracy range as the joint reference at higher compute.

particular, across much of the plotted range, More first schedules tend to reach a higher accuracy range than Less first schedules at comparable FLOPs, with the Equal schedule typically in between. The gap is more visible in the low-to-moderate compute regime and narrows as compute increases. We interpret this as exploratory qualitative evidence that, in this simple nonlinear PEFT setting as well, allocating more of the budget to earlier components can be beneficial.

### 5.3 Experimental analysis using LoRA on simple language tasks.

**Experimental setting.** We employ a `distilbert-base-uncased` model (Sanh et al., 2019), a smaller and faster variant of BERT, and fine-tune it on the GLUE SST-2 dataset for binary sentiment classification (Yang et al., 2024). Following the same methodology as in the vision tasks, we freeze all base model parameters and inject LoRA adapters into the query (`q_lin`) and value (`v_lin`) projections of each attention layer. We again compare the standard, joint-training LoRA (with $r = 3$) against our sequential rank-1 adaptation, which applies the non-linear optimization from equation (14) for a total target rank of $r = 3$. Because this model and loss are nonlinear, these experiments are not direct instantiations of the regression theory in Sections 3–4; rather, they test whether a similar qualitative scheduling effect appears in a more realistic PEFT setting.

**Adaptation performance.** Figure 6 summarizes the final SST-2 accuracy after sequentially adding one, two, and three rank-1 components, together with the jointly trained rank-3 LoRA reference. The displayed sequential accuracies increase from 0.872 at Rank-1 to 0.875 at Rank-2 and 0.876 at Rank-3, while the joint rank-3 reference reaches 0.872. Notably, the first sequential component already matches the displayed joint-reference accuracy, indicating that a substantial fraction of the adaptation is captured early in this setup. Adding the second and third sequential components then yields modest further gains. For the final rank-3 comparison, the sequential and joint runs are also close in adaptation-training compute, differing by less than 3% in measured FLOPs. Taken together, Figure 6 places sequential rank growth and jointly trained rank-3 LoRA in a very similar final accuracy–compute regime on this SST-2 experiment, while the sequential procedure builds adaptation capacity one component at a time. This indicates that sequential rank growth can remain effective in this nonlinear PEFT setting, without suggesting superiority over joint training.

To study schedule effects more directly, we next fix the total target rank at $r = 3$ and vary the more-first schedule parameter $\alpha$ while tracking accuracy as a function of adaptation-training compute.

**Accuracy–compute tradeoff under more-first schedules.** Figure 7 examines schedule effects more directly by fixing the total target rank at $r = 3$ and sweeping the more-first branch of the common $\alpha$-schedule family. The joint LoRA reference is strongest in the lowest-compute part of the plot, but several more-first schedules close this gap as compute increases. Across much of the plotted range, many more-first trajectories lie above the equal-schedule baseline, showing that how the budget is distributed across the three sequential components matters in addition to the total compute itself. At higher compute, the more-first curves cluster closely, indicating diminishing differences across more-first schedules once earlier components already receive substantial budget. Taken together, Figure 7 shows that, within the more-first branch, schedule choice affects the attainable accuracy–compute tradeoff on SST-2.

Two nuances are important. First, Figure 7 only sweeps the more-first branch of the $\alpha$-schedule family, so it should not be used to draw conclusions about less-first schedules on SST-2. Second, the dependence on $\alpha$ within this branch is not perfectly monotone: several more-first schedules close the gap to the joint reference and improve over the equal-schedule baseline, but the orange trajectories cluster closely at higher compute, indicating diminishing differences once earlier components already receive substantial budget. We therefore interpret the SST-2 results as supporting a nuanced more-first scheduling picture: allocating relatively more compute to earlier components can help in this setting, but Figure 7 does not suggest a single universally optimal $\alpha$.

## 6 Limitations

Our theoretical analysis and experimental results on sequential rank-1 learning have certain limitations. First, the theoretical analysis is constrained to linear low-rank regression. Moreover, the guarantees in Section 4 are parameter-recovery bounds under fixed design; they do not establish out-of-sample prediction risk on a separate test distribution. Accordingly, the LoRA experiments in Sections 5.2–5.3 should be viewed as exploratory qualitative extensions rather than direct validations of the theorems for nonlinear architectures and classification losses. Moreover, while sequential rank-1 updates offer flexible rank determination, they may demand more training iterations than simultaneous rank optimization. Furthermore, our nonlinear PEFT experiments remain limited in scope: the vision study uses a simple feedforward network on standard image-classification tasks, while the language study considers a single DistilBERT-based sentiment-classification task (SST-2). Lastly, it remains an open question to characterize the optimal allocation of training epochs across components.

## 7 Conclusion

By examining error propagation in hierarchical learning frameworks, we demonstrate how the accuracy of sequential components is interconnected, with each subsequent step dependent on the precision of preceding estimations. Synthetic low-rank linear-regression experiments validate the theory, while the LoRA adaptation experiments suggest that related qualitative scheduling effects can persist in simple nonlinear PEFT settings. From a practical point of view, the linear theory and synthetic experiments support prioritizing earlier components, while the nonlinear LoRA results suggest a more nuanced adaptive scheduling picture rather than a simple monotone more-first rule. To this end, our work opens up the following future directions:
—*Adaptive Procedures*: We can develop adaptive procedures that adjust the number of gradient steps $t_k$ based on the estimated approximation error; this connects with learning schedules literature.
—*Component Reoptimization*: Periodically refine earlier components after extracting new ones.
—*Orthogonality Constraints*: Enforce orthogonality between components to reduce interference.
—*Hybrid Approaches*: Combine sequential rank-1 updates with occasional full-rank steps.

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

# Appendix

## Appendix Contents

## A  Proof of Theorem 1

The proof focuses on bounding two crucial quantities: $\left\|\mathbf{b}_k^\star \mathbf{a}_k^{\star\top}\mathbf{X} - \overline{\mathbf{b}}_k\overline{\mathbf{a}}_k^\top\mathbf{X}\right\|_F$ and $\left\|\mathbf{Y}_k - \mathbf{Y}_k^\star\right\|_F$. For convenience, we refer the reader to Table 1 for the distinction between the exact rank-1 pair, the step-optimal pair, and the finite-iteration output at each stage. The first quantity measures the difference between the true *k-th* rank-1 approximation of $Y$ and the leading rank-1 estimation of $\mathbf{Y}_k$ derived from the *k-th* deflation step, which minimizes the loss specified in (1). The second quantity evaluates the distance between the "ground-truth" deflation matrices, $\mathbf{Y}_k^\star$, and the deflation matrices, $\mathbf{Y}_k$, obtained in practice. Together, these bounds provide a comprehensive understanding of the approximation accuracy and the effectiveness of the deflation process.

A key technical step is to verify the perturbation premise required by Lemma 3. We make this explicit in Appendix B.4, equations 32–33, in the proof of Theorem 8, which implies Theorem 1.

The difference between $\sum_{k=1}^r \mathbf{b}_k\mathbf{a}_k^\top\mathbf{X}$, the sum of rank-1 approximations returned by Algorithm (2), and $\mathbf{Y}$, the training data label matrix, consists of three components:

- **Ground-truth approximation error**: $\left\|\mathbf{Y} - \sum_{k=1}^r \mathbf{b}_k^\star \mathbf{a}_k^{\star\top}\mathbf{X}\right\|_F$, which measures the deviation between the true product $\mathbf{Y}$ and the sum of the exact top $r$ rank-1 approximations of $\mathbf{Y}$.

- **Propagation error**: $\left\|\sum_{k=1}^r \mathbf{b}_k^\star \mathbf{a}_k^{\star\top}\mathbf{X} - \sum_{k=1}^r \overline{\mathbf{b}}_k\overline{\mathbf{a}}_k^\top\mathbf{X}\right\|_F$, which accumulates the differences between the exact top rank-1 estimates of each "ground-truth" deflated matrix, $\mathbf{Y}_k^\star$, and the corresponding empirical deflated matrix, $\mathbf{Y}_k$.

- **Optimization error**: $\left\| \sum_{k=1}^{r} \overline{\mathbf{b}}_k \overline{\mathbf{a}}_k^\top \mathbf{X} - \sum_{k=1}^{r} \mathbf{b}_k \mathbf{a}_k^\top \mathbf{X} \right\|_F$, which captures the cumulative numerical error from the sub-routine `rank-1` not being solved exactly.

Combining these, we can express the overall difference $\left\| \mathbf{Y} - \sum_{k=1}^{r} \mathbf{b}_k \mathbf{a}_k^\top \mathbf{X} \right\|_F$ as follows:

$$
\begin{aligned}
\left\| \mathbf{Y} - \sum_{k=1}^{r} \mathbf{b}_k \mathbf{a}_k^\top \mathbf{X} \right\|_F &\leq \left\| \mathbf{Y} - \sum_{k=1}^{r} \mathbf{b}_k^\star \mathbf{a}_k^{\star\top} \mathbf{X} \right\|_F + \left\| \sum_{k=1}^{r} \mathbf{b}_k^\star \mathbf{a}_k^{\star\top} \mathbf{X} - \sum_{k=1}^{r} \overline{\mathbf{b}}_k \overline{\mathbf{a}}_k^\top \mathbf{X} \right\|_F \\
&\quad + \underbrace{\left\| \sum_{k=1}^{r} \overline{\mathbf{b}}_k \overline{\mathbf{a}}_k^\top \mathbf{X} - \sum_{k=1}^{r} \mathbf{b}_k \mathbf{a}_k^\top \mathbf{X} \right\|_F}_{\leq \sum_{k=1}^{r} \| \mathbf{\Psi}_k \mathbf{X} \|_F}
\end{aligned}
\tag{15}
$$

The last inequality follows from the definition in (7). The norm $\| \mathbf{\Psi}_k \|_F$ largely depends on the sub-routine `rank-1` and can be controlled as long as $t$, the number of sub-routine iterations, is sufficiently large.

By Lemma 1 and orthogonality of the singular-vector pairs,

$$
\left\| \mathbf{Y} - \sum_{k=1}^{r} b_k^\star a_k^{\star\top} \mathbf{X} \right\|_F = \left\| \sum_{k=r+1}^{p} \sigma_k^\star u_k^\star v_k^{\star\top} \right\|_F = \left( \sum_{k=r+1}^{p} (\sigma_k^\star)^2 \right)^{1/2}.
$$

By the triangle inequality:

$$
\begin{aligned}
\left\| \mathbf{Y} - \sum_{k=1}^{r} \mathbf{b}_k \mathbf{a}_k^\top \mathbf{X} \right\|_F &\leq \left( \sum_{k=r+1}^{p} (\sigma_k^\star)^2 \right)^{1/2} + \left\| \sum_{k=1}^{r} \mathbf{b}_k^\star \mathbf{a}_k^{\star\top} \mathbf{X} - \sum_{k=1}^{r} \overline{\mathbf{b}}_k \overline{\mathbf{a}}_k^\top \mathbf{X} \right\|_F + \sum_{k=1}^{r} \| \mathbf{\Psi}_k \mathbf{X} \|_F \\
&\leq \left( \sum_{k=r+1}^{p} (\sigma_k^\star)^2 \right)^{1/2} + \sum_{k=1}^{r} \left\| \mathbf{b}_k^\star \mathbf{a}_k^{\star\top} \mathbf{X} - \overline{\mathbf{b}}_k \overline{\mathbf{a}}_k^\top \mathbf{X} \right\|_F + \sum_{k=1}^{r} \| \mathbf{\Psi}_k \mathbf{X} \|_F.
\end{aligned}
\tag{16}
$$

Therefore, our analysis will primarily focus on upper-bounding $\left\| \mathbf{b}_k^\star \mathbf{a}_k^{\star\top} \mathbf{X} - \overline{\mathbf{b}}_k \overline{\mathbf{a}}_k^\top \mathbf{X} \right\|_F$. Intuitively, this depends on the difference between $\mathbf{Y}_k$ and $\mathbf{Y}_k^\star$. As a foundational step in our proof, we will first provide a characterization of $\| \mathbf{Y}_k - \mathbf{Y}_k^\star \|_F$.

**Lemma 2.** *Let $\mathbf{Y}_k^\star$'s and $\mathbf{Y}_k$'s be defined as generated by Algorithm 1 and Algorithm 2, respectively. Then we have that for all $k \in [r]$,*

$$
\| \mathbf{Y}_{k+1} - \mathbf{Y}_{k+1}^\star \|_F \leq \| \mathbf{Y}_k - \mathbf{Y}_k^\star \|_F + \| \mathbf{b}_k^\star \mathbf{a}_k^{\star\top} \mathbf{X} - \overline{\mathbf{b}}_k \overline{\mathbf{a}}_k^\top \mathbf{X} \|_F + \| \mathbf{\Psi}_k \mathbf{X} \|_F.
\tag{17}
$$

Here the additive term $\| \mathbf{\Psi}_k \mathbf{X} \|_F$ is exactly the step-$k$ numerical error induced by the inexact rank-1 subroutine from Definition 1. Lemma 2 upper-bounds $\left\| \mathbf{Y}_{k+1} - \mathbf{Y}_{k+1}^\star \right\|_F$ in terms of $\| \mathbf{Y}_k - \mathbf{Y}_k^\star \|_F$, $\left\| \mathbf{b}_k^\star \mathbf{a}_k^{\star\top} \mathbf{X} - \overline{\mathbf{b}}_k \overline{\mathbf{a}}_k^\top \mathbf{X} \right\|_F$, and $\| \mathbf{\Psi}_k \mathbf{X} \|_F$. To establish a recursive characterization of $\| \mathbf{Y}_k - \mathbf{Y}_k^\star \|_F$, we first need to derive an upper bound for $\left\| \overline{\mathbf{b}}_k \overline{\mathbf{a}}_k^\top \mathbf{X} - \mathbf{b}_k^\star \mathbf{a}_k^{\star\top} \mathbf{X} \right\|_F$.

**Lemma 3.** *Let $\mathcal{T}_k := \min \{ \min_{j>k} |\sigma_k^\star - \sigma_{j_k}|, \sigma_k^\star \}$. If $\| \mathbf{Y}_k^\star - \mathbf{Y}_k \|_2 < \min_{j>k} |\sigma_k^\star - \sigma_j^\star|$, then for all $k \in [r]$, we have:*

$$
\left\| \mathbf{b}_k^\star \mathbf{a}_k^{\star\top} \mathbf{X} - \overline{\mathbf{b}}_k \overline{\mathbf{a}}_k^\top \mathbf{X} \right\|_F \leq \left( \frac{3\sigma_k^\star}{\mathcal{T}_k} + 1 \right) \| \mathbf{Y}_k^\star - \mathbf{Y}_k \|_F.
\tag{18}
$$

Plugging 18 into 17 gives a recurrence that depends purely on $\| \mathbf{\Psi}_k \|_F$ and the spectrum of $\mathbf{Y}$:

$$
\begin{aligned}
\left\| \mathbf{Y}_{k+1} - \mathbf{Y}_{k+1}^\star \right\|_F &\leq \| \mathbf{Y}_k - \mathbf{Y}_k^\star \|_F + \left( \frac{3\sigma_k^\star}{\mathcal{T}_k} + 1 \right) \| \mathbf{Y}_k^\star - \mathbf{Y}_k \|_F + \| \mathbf{\Psi}_k \mathbf{X} \|_F \\
&\leq \left( \frac{3\sigma_k^\star}{\mathcal{T}_k} + 2 \right) \| \mathbf{Y}_k^\star - \mathbf{Y}_k \|_F + \| \mathbf{\Psi}_k \mathbf{X} \|_F.
\end{aligned}
\tag{19}
$$

Unrolling this recurrence gives a closed-form upper bound for $\| \mathbf{Y}_k - \mathbf{Y}_k^\star \|_F$. Combining this upper bound with 3 and plugging the result into 16 gives us an upper bound for $\| \mathbf{Y} - \sum_{k=1}^{r} \mathbf{b}_k \mathbf{a}_k^\top \mathbf{X} \|_F$.

# B    Missing proofs from Appendix A and a generalized theorem

## B.1    Proof of Lemma 1

*Proof.* Let $\{(\mathbf{a}_k^\star, \mathbf{b}_k^\star)\}_{k=1}^r$ denote the outputs of Algorithm 1. According to the third line in Algorithm 1, starting from $k = 1$, we have:

$$(\mathbf{a}_1^\star, \mathbf{b}_1^\star) = \underset{\mathbf{a} \in \mathbb{R}^d, \mathbf{b} \in \mathbb{R}^m}{\arg\min} \frac{1}{2} \left\| \mathbf{Y}_1^\star - \mathbf{b}\mathbf{a}^\top \mathbf{X} \right\|_F^2 .$$

Given that $\mathbf{b}\mathbf{a}^\top$ is a rank-1 matrix, the product $\mathbf{b}\mathbf{a}^\top \mathbf{X}$ is also rank-1. Therefore, according to the Eckart-Young-Mirsky theorem, the best rank-1 approximation of $\mathbf{Y}_1^\star = \mathbf{Y} = \sum_{i=1}^p \sigma_i^\star \mathbf{u}_i^\star \mathbf{v}_i^{\star\top}$, based on its Singular Value Decomposition, is $\sigma_1^\star \mathbf{u}_1^\star \mathbf{v}_1^{\star\top}$. Thus, $\mathbf{b}_1^\star \mathbf{a}_1^{\star\top} \mathbf{X} = \sigma_1^\star \mathbf{u}_1^\star \mathbf{v}_1^{\star\top}$. Next, the deflation step in line 4 yields:

$$\mathbf{Y}_2^\star := \mathbf{Y}_1^\star - \mathbf{b}_1^\star \mathbf{a}_1^{\star\top} \mathbf{X} = \mathbf{Y}_1^\star - \sigma_1^\star \mathbf{u}_1^\star \mathbf{v}_1^{\star\top} = \sum_{i=2}^p \sigma_i^\star \mathbf{u}_i^\star \mathbf{v}_i^{\star\top} .$$

Now, assuming by induction that $\mathbf{Y}_k^\star = \sum_{i=k}^p \sigma_i^\star \mathbf{u}_i^\star \mathbf{v}_i^{\star\top}$ and $\mathbf{b}_k^\star \mathbf{a}_k^{\star\top} \mathbf{X} = \sigma_k^\star \mathbf{u}_k^\star \mathbf{v}_k^{\star\top}$, we obtain for $k+1$:

$$\mathbf{Y}_{k+1}^\star = \mathbf{Y}_k^\star - \mathbf{b}_k^\star \mathbf{a}_k^{\star\top} \mathbf{X} = \sum_{i=k}^p \sigma_i^\star \mathbf{u}_i^\star \mathbf{v}_i^{\star\top} - \sigma_k^\star \mathbf{u}_k^\star \mathbf{v}_k^{\star\top} = \sum_{i=k+1}^p \sigma_i^\star \mathbf{u}_i^\star \mathbf{v}_i^{\star\top} .$$

Returning to line 3, we have:

$$(\mathbf{a}_{k+1}^\star, \mathbf{b}_{k+1}^\star) = \underset{\mathbf{a} \in \mathbb{R}^d, \mathbf{b} \in \mathbb{R}^m}{\arg\min} \frac{1}{2} \left\| \mathbf{Y}_{k+1}^\star - \mathbf{b}\mathbf{a}^\top \mathbf{X} \right\|_F^2 .$$

By the same reasoning as for $k = 1$, $\mathbf{b}_{k+1}^\star \mathbf{a}_{k+1}^{\star\top} \mathbf{X}$ is equal to the first rank-1 approximation of $\mathbf{Y}_{k+1}^\star = \sum_{i=k+1}^p \sigma_i^\star \mathbf{u}_i^\star \mathbf{v}_i^{\star\top}$, which is $\sigma_{k+1}^\star \mathbf{u}_{k+1}^\star \mathbf{v}_{k+1}^{\star\top}$. ☐

## B.2    Proof of Lemma 2

*Proof.* As we defined in (7), we'll have:

$$\mathbf{b}_k \mathbf{a}_k^\top = \overline{\mathbf{b}}_k \overline{\mathbf{a}}_k^\top + \mathbf{\Psi}_k$$

Plugging this into (2) we'll get:

$$\mathbf{Y}_{k+1} = \mathbf{Y}_k - (\overline{\mathbf{b}}_k \overline{\mathbf{a}}_k^\top + \mathbf{\Psi}_k)\mathbf{X}$$

Now let $\mathbf{\Delta}_k = \|\mathbf{Y}_k - \mathbf{Y}_k^\star\|_F$

$$\begin{aligned}
\mathbf{\Delta}_{k+1} &= \|\mathbf{Y}_{k+1} - \mathbf{Y}_{k+1}^\star\|_F \\
&= \|\mathbf{Y}_k - (\overline{\mathbf{b}}_k \overline{\mathbf{a}}_k^\top + \mathbf{\Psi}_k)\mathbf{X} - \mathbf{Y}_k^\star + \mathbf{b}_k^\star \mathbf{a}_k^{\star\top} \mathbf{X}\|_F \\
&= \|\mathbf{Y}_k - \mathbf{Y}_k^\star - (\overline{\mathbf{b}}_k \overline{\mathbf{a}}_k^\top - \mathbf{b}_k^\star \mathbf{a}_k^{\star\top})\mathbf{X} - \mathbf{\Psi}_k \mathbf{X}\|_F \\
&\leq \|\mathbf{Y}_k - \mathbf{Y}_k^\star\|_F + \|(\overline{\mathbf{b}}_k \overline{\mathbf{a}}_k^\top - \mathbf{b}_k^\star \mathbf{a}_k^{\star\top})\mathbf{X}\|_F + \|\mathbf{\Psi}_k \mathbf{X}\|_F \\
&\leq \mathbf{\Delta}_k + \|\overline{\mathbf{b}}_k \overline{\mathbf{a}}_k^\top \mathbf{X} - \mathbf{b}_k^\star \mathbf{a}_k^{\star\top} \mathbf{X}\|_F + \|\mathbf{\Psi}_k \mathbf{X}\|_F
\end{aligned}$$

☐

## B.3    Proof of Lemma 3

*Proof.* We start by observing:

$$\begin{aligned}
\left\| \mathbf{b}_k^\star \mathbf{a}_k^{\star\top} \mathbf{X} - \overline{\mathbf{b}}_k \overline{\mathbf{a}}_k^\top \mathbf{X} \right\|_F &= \| \sigma_k^\star \mathbf{u}_k^\star \mathbf{v}_k^{\star\top} - \sigma_{1_k} \mathbf{u}_{1_k} \mathbf{v}_{1_k}^\top \|_F \\
&= \| \sigma_k^\star (\mathbf{u}_k^\star \mathbf{v}_k^{\star\top} - \mathbf{u}_{1_k} \mathbf{v}_{1_k}^\top) + (\sigma_k^\star - \sigma_{1_k}) \mathbf{u}_{1_k} \mathbf{v}_{1_k}^\top \|_F \\
&\leq \| \sigma_k^\star \|_F \cdot \| \mathbf{u}_k^\star \mathbf{v}_k^{\star\top} - \mathbf{u}_{1_k} \mathbf{v}_{1_k}^\top \|_F + \| \sigma_k^\star - \sigma_{1_k} \|_F \cdot \underbrace{\| \mathbf{u}_{1_k} \|_2}_{=1} \cdot \underbrace{\| \mathbf{v}_{1_k}^\top \|_2}_{=1} \\
&\leq | \sigma_k^\star | \cdot \| \mathbf{u}_k^\star \mathbf{v}_k^{\star\top} - \mathbf{u}_{1_k} \mathbf{v}_{1_k}^\top \|_F + | \sigma_k^\star - \sigma_{1_k} |.
\end{aligned}$$

Therefore,

$$\|\mathbf{b}_k^\star \mathbf{a}_k^{\star\top} \mathbf{X} - \overline{\mathbf{b}}_k \overline{\mathbf{a}}_k^\top \mathbf{X}\|_F \leq \sigma_k^\star \cdot \|\mathbf{u}_k^\star \mathbf{v}_k^{\star\top} - \mathbf{u}_{1_k} \mathbf{v}_{1_k}^\top\|_F + |\sigma_k^\star - \sigma_{1_k}|. \tag{20}$$

Now, we express the term $\|\mathbf{u}_k^\star \mathbf{v}_k^{\star\top} - \mathbf{u}_{1_k} \mathbf{v}_{1_k}^\top\|_F$ as follows:

$$\begin{aligned}
\|\mathbf{u}_k^\star \mathbf{v}_k^{\star\top} - \mathbf{u}_{1_k} \mathbf{v}_{1_k}^\top\|_F &= \|\mathbf{u}_k^\star (\mathbf{v}_k^{\star\top} - \mathbf{v}_{1_k}^\top) + (\mathbf{u}_k^\star - \mathbf{u}_{1_k}) \mathbf{v}_{1_k}^\top\|_F \\
&\leq \underbrace{\|\mathbf{u}_k^\star\|_2}_{=1} \cdot \|\mathbf{v}_k^{\star\top} - \mathbf{v}_{1_k}^\top\|_2 + \|\mathbf{u}_k^\star - \mathbf{u}_{1_k}\|_2 \cdot \underbrace{\|\mathbf{v}_{1_k}^\top\|_2}_{=1} \\
&\leq \|\mathbf{v}_k^{\star\top} - \mathbf{v}_{1_k}^\top\|_2 + \|\mathbf{u}_k^\star - \mathbf{u}_{1_k}\|_2
\end{aligned}$$

Substituting this result into inequality equation 20 gives:

$$\|\mathbf{b}_k^\star \mathbf{a}_k^{\star\top} \mathbf{X} - \overline{\mathbf{b}}_k \overline{\mathbf{a}}_k^\top \mathbf{X}\|_F \leq \sigma_k^\star \left( \|\mathbf{v}_k^{\star\top} - \mathbf{v}_{1_k}^\top\|_2 + \|\mathbf{u}_k^\star - \mathbf{u}_{1_k}\|_2 \right) + \underbrace{|\sigma_k^\star - \sigma_{1_k}|}_{\leq \|\mathbf{Y}_k^\star - \mathbf{Y}_k\|_F}. \tag{21}$$

The last inequality follows from Weyl's theorem. To complete the bound, we will now find an upper bound for $\|\mathbf{v}_k^\star - \mathbf{v}_{1_k}\| + \|\mathbf{u}_k^\star - \mathbf{u}_{1_k}\|$. To do so, we will define two angles:

$$\alpha := \angle \{\mathbf{v}_k^\star, \mathbf{v}_{1_k}\} \quad \text{and} \quad \beta := \angle \{\mathbf{u}_k^\star, \mathbf{u}_{1_k}\}$$

We know that

$$\cos\alpha = \frac{\mathbf{v}_k^{\star\top} \mathbf{v}_{1_k}}{\|\mathbf{v}_k^\star\|_2 \|\mathbf{v}_{1_k}\|_2} = \mathbf{v}_k^{\star\top} \mathbf{v}_{1_k} \leq 1$$

since $\|\mathbf{v}_k^\star\|_2 = 1$ and $\|\mathbf{v}_{1_k}\|_2 = 1$. Therefore

$$\sin^2\alpha = 1 - \cos^2\alpha = 1 - (\mathbf{v}_k^{\star\top} \mathbf{v}_{1_k})^2 \Rightarrow (\mathbf{v}_k^{\star\top} \mathbf{v}_{1_k})^2 = 1 - \sin^2\alpha$$

We use the expansion of the square of $\|\mathbf{v}_k^\star - \mathbf{v}_{1_k}\|_2$ to get:

$$\begin{aligned}
\|\mathbf{v}_k^\star - \mathbf{v}_{1_k}\|_2^2 &= \underbrace{\|\mathbf{v}_k^\star\|_2^2}_{=1} + \underbrace{\|\mathbf{v}_{1_k}\|_2^2}_{=1} - 2(\mathbf{v}_k^{\star\top} \mathbf{v}_{1_k}) \\
&= 2 - 2(\mathbf{v}_k^{\star\top} \mathbf{v}_{1_k}) \\
&\leq 2 - 2(\mathbf{v}_k^{\star\top} \mathbf{v}_{1_k})^2 = 2 - 2(1 - \sin^2\alpha) \\
&= 2\sin^2\alpha
\end{aligned}$$

Thus

$$\|\mathbf{v}_k^\star - \mathbf{v}_{1_k}\|_2^2 \leq 2\sin^2\alpha \tag{22}$$

Following the same procedure for $\beta$, we start with:

$$\cos\beta = \frac{\mathbf{u}_k^{\star\top} \mathbf{u}_{1_k}}{\|\mathbf{u}_k^\star\|_2 \|\mathbf{u}_{1_k}\|_2} = \mathbf{u}_k^{\star\top} \mathbf{u}_{1_k} \leq 1$$

since $\|\mathbf{u}_k^\star\|_2 = 1$ and $\|\mathbf{u}_{1_k}\|_2 = 1$. Therefore

$$\sin^2\beta = 1 - \cos^2\beta = 1 - (\mathbf{u}_k^{\star\top} \mathbf{u}_{1_k})^2 \Rightarrow (\mathbf{u}_k^{\star\top} \mathbf{u}_{1_k})^2 = 1 - \sin^2\beta$$

We use the expansion of the square of $\|\mathbf{u}_k^\star - \mathbf{u}_{1_k}\|_2$ to get:

$$\begin{aligned}
\|\mathbf{u}_k^\star - \mathbf{u}_{1_k}\|_2^2 &= \underbrace{\|\mathbf{u}_k^\star\|_2^2}_{=1} + \underbrace{\|\mathbf{u}_{1_k}\|_2^2}_{=1} - 2(\mathbf{u}_k^{\star\top} \mathbf{u}_{1_k}) \\
&= 2 - 2(\mathbf{u}_k^{\star\top} \mathbf{u}_{1_k}) \\
&\leq 2 - 2(\mathbf{u}_k^{\star\top} \mathbf{u}_{1_k})^2 \\
&= 2 - 2(1 - \sin^2\beta) = 2\sin^2\beta
\end{aligned}$$

Thus

$$\|\mathbf{u}_k^\star - \mathbf{u}_{1_k}\|_2^2 \leq 2\sin^2\beta \tag{23}$$

Using the fact that $a + b \leq \sqrt{2a^2 + 2b^2}$, 22, and 23 we'll get:

$$\|\mathbf{v}_k^\star - \mathbf{v}_{1_k}\|_2 + \|\mathbf{u}_k^\star - \mathbf{u}_{1_k}\|_2 \leq \sqrt{2\|\mathbf{v}_k^\star - \mathbf{v}_{1_k}\|_2^2 + 2\|\mathbf{u}_k^\star - \mathbf{u}_{1_k}\|_2^2} \leq 2\sqrt{\sin^2\alpha + \sin^2\beta} \tag{24}$$

To proceed, we observe that under the assumption $\|Y_k^\star - Y_k\|_2 < \min_{j>k} |\sigma_k^\star - \sigma_j^\star|$, Weyl's inequality implies that

$$\min_{j>k} |\sigma_k^\star - \sigma_{j_k}| \geq \min_{j>k} |\sigma_k^\star - \sigma_j^\star| - |\sigma_{j_k} - \sigma_j^\star| > \|Y_k^\star - Y_k\|_2 - \|Y_k^\star - Y_k\|_2 = 0.$$

We know $\sigma_k^\star > 0$. Since we define $\mathcal{T}_k := \min\{\min_{j>k} |\sigma_k^\star - \sigma_{j_k}|, \sigma_k^\star\}$, and both terms are positive, $\mathcal{T}_k$ is also positive. This satisfies the conditions of Wedin's theorem, allowing us to apply it.

Now since $\alpha := \angle\{\mathbf{v}_k^\star, \mathbf{v}_{1_k}\}$ and $\beta := \angle\{\mathbf{u}_k^\star, \mathbf{u}_{1_k}\}$, and because $\mathbf{v}_k^\star$ and $\mathbf{u}_k^\star$ are the first right and left singular vectors of $\mathbf{Y}_k^\star$, while $\mathbf{v}_{1_k}$ and $\mathbf{u}_{1_k}$ are the first right and left singular vectors of $\mathbf{Y}_k$, we apply Wedin's theorem 10 in the rank-1 case with

$$\mathbf{M} = \mathbf{Y}_k^\star, \qquad \widetilde{\mathbf{M}} = \mathbf{Y}_k, \qquad \mathbf{\Delta}_k := \widetilde{\mathbf{M}} - \mathbf{M} = \mathbf{Y}_k - \mathbf{Y}_k^\star,$$

$$\mathbf{U}_1 = \mathbf{u}_k^\star, \qquad \mathbf{V}_1 = \mathbf{v}_k^\star, \qquad \widetilde{\mathbf{U}}_1 = \mathbf{u}_{1_k}, \qquad \widetilde{\mathbf{V}}_1 = \mathbf{v}_{1_k}, \qquad \delta = \mathcal{T}_k.$$

Note that $\mathbf{u}_k^\star \in \mathbb{R}^m$ and $\mathbf{v}_k^\star \in \mathbb{R}^n$, so $\mathbf{u}_k^{\star\top}\mathbf{\Delta}_k \in \mathbb{R}^{1\times n}$ and $\mathbf{\Delta}_k\mathbf{v}_k^\star \in \mathbb{R}^{m\times 1}$ are dimensionally consistent in the rectangular setting. Hence,

$$\begin{aligned}
\sin^2\alpha + \sin^2\beta &\leq \frac{\|\mathbf{u}_k^{\star\top}\mathbf{\Delta}_k\|_F^2 + \|\mathbf{\Delta}_k\mathbf{v}_k^\star\|_F^2}{\mathcal{T}_k^2} \\
&= \frac{\|\mathbf{u}_k^{\star\top}(\mathbf{Y}_k^\star - \mathbf{Y}_k)\|_F^2 + \|(\mathbf{Y}_k^\star - \mathbf{Y}_k)\mathbf{v}_k^\star\|_F^2}{\mathcal{T}_k^2} \\
&\leq \frac{\|\mathbf{u}_k^\star\|_2^2\|\mathbf{Y}_k^\star - \mathbf{Y}_k\|_F^2 + \|\mathbf{Y}_k^\star - \mathbf{Y}_k\|_F^2\|\mathbf{v}_k^\star\|_2^2}{\mathcal{T}_k^2} \\
&= \frac{2\|\mathbf{Y}_k^\star - \mathbf{Y}_k\|_F^2}{\mathcal{T}_k^2}.
\end{aligned}$$

Here, the second equality uses the sign-invariance of the norm, since $\mathbf{Y}_k - \mathbf{Y}_k^\star = -(\mathbf{Y}_k^\star - \mathbf{Y}_k)$, and the last equality uses $\|\mathbf{u}_k^\star\|_2 = \|\mathbf{v}_k^\star\|_2 = 1$. (Equivalently, because these are vector terms, one may replace $\|\cdot\|_F$ by $\|\cdot\|_2$ throughout this display.)

Plugging this in 24, we'll have:

$$\|\mathbf{v}_k^\star - \mathbf{v}_{1_k}\| + \|\mathbf{u}_k^\star - \mathbf{u}_{1_k}\| \leq 2\sqrt{2}\frac{\|\mathbf{Y}_k^\star - \mathbf{Y}_k\|_F}{\mathcal{T}_k} \leq 3\frac{\|\mathbf{Y}_k^\star - \mathbf{Y}_k\|_F}{\mathcal{T}_k}$$

Plugging this to 21 and using Weyl's inequality, we'll get:

$$\|\mathbf{b}_k^\star\mathbf{a}_k^{\star\top}\mathbf{X} - \overline{\mathbf{b}}_k\overline{\mathbf{a}}_k^\top\mathbf{X}\|_F \leq \sigma_k^\star\left(3\frac{\|\mathbf{Y}_k^\star - \mathbf{Y}_k\|_F}{\mathcal{T}_k}\right) + \|\mathbf{Y}_k^\star - \mathbf{Y}_k\|_F = \left(\frac{3\sigma_k^\star}{\mathcal{T}_k} + 1\right)\|\mathbf{Y}_k^\star - \mathbf{Y}_k\| \tag{25}$$

$\square$

## B.4 Proof of a more general form of Theorem 1

In this section, we prove a more general form of Theorem 1, stated below.

**Theorem 8.** Let $\{(\mathbf{a}_k, \mathbf{b}_k)\}_{k=1}^r$ be the output of Algorithm 2. Let $\mathbf{\Psi}_k$ be given as in Definition 1 with $\|\mathbf{\Psi}_k\|_F > 0$. Let $\mathbf{Y} = \sum_{k=1}^{r^\star} \sigma_k^\star\mathbf{u}_k^\star\mathbf{v}_k^{\star\top}$ be the SVD of $\mathbf{Y}$, with $\sigma_1^\star \geq \cdots \geq \sigma_{r^\star}^\star$ and define $\mathbf{Y}^{(r')} =$

$\sum_{k=1}^{r'} \sigma_k^\star \mathbf{u}_k^\star \mathbf{v}_k^{\star\top}$. *Adopt the convention $\sigma_{r^\star+1}^\star := 0$ and define the singular value gap as $\mathcal{T}_k^\star := \sigma_k^\star - \sigma_{k+1}^\star$. Assume $\mathcal{T}_k^\star > 0$ for all $k \in [r]$. Define error bound $E(k)$ as:*

$$E(k) := \sigma_{\max}(\mathbf{X}) \sum_{k'=0}^{k-1} \|\mathbf{\Psi}_{k'}\|_F \prod_{j=k'+1}^{k-1} \left( \frac{6\sigma_j^\star}{\mathcal{T}_j^\star} + 2 \right)$$

*If $\|\mathbf{\Psi}_k\|_F$'s are small enough such that $E(k) < \frac{1}{2}\mathcal{T}_k^\star$ for all $k \in [r]$, then for any $r \le r' \le r^\star$, the output of Algorithm 2 satisfies:*

$$\left\| \mathbf{Y}^{(r')} - \sum_{k=1}^{r} \mathbf{b}_k \mathbf{a}_k^\top \mathbf{X} \right\|_F \le \left( \sum_{k=r+1}^{r'} (\sigma_k^\star)^2 \right)^{1/2} + \sigma_{\max}(\mathbf{X}) \sum_{k=1}^{r} \sum_{k'=0}^{k} \|\mathbf{\Psi}_{k'}\|_F \prod_{j=k'+1}^{k} \left( 2 + \frac{6\sigma_j^\star}{\mathcal{T}_j^\star} \right) \quad (26)$$

Notice that Theorem 8 naturally implies Theorem 1 by taking $r' = r^\star$. We give the proof of Theorem 8 below.

*Proof.* We decompose the approximation error as

$$\left\| \mathbf{Y}^{(r')} - \sum_{k=1}^{r} \mathbf{b}_k \mathbf{a}_k^\top \mathbf{X} \right\|_F \le \left\| \mathbf{Y}^{(r')} - \sum_{k=1}^{r} \mathbf{b}_k^\star \mathbf{a}_k^{\star\top} \mathbf{X} \right\|_F + \left\| \sum_{k=1}^{r} \mathbf{b}_k^\star \mathbf{a}_k^{\star\top} \mathbf{X} - \sum_{k=1}^{r} \overline{\mathbf{b}}_k \overline{\mathbf{a}}_k^\top \mathbf{X} \right\|_F$$
$$+ \left\| \sum_{k=1}^{r} \overline{\mathbf{b}}_k \overline{\mathbf{a}}_k^\top \mathbf{X} - \sum_{k=1}^{r} \mathbf{b}_k \mathbf{a}_k^\top \mathbf{X} \right\|_F \quad (27)$$

By Lemma 1, we must have that

$$\left\| \mathbf{Y}^{(r')} - \sum_{k=1}^{r} \mathbf{b}_k^\star \mathbf{a}_k^{\star\top} \mathbf{X} \right\|_F = \left\| \sum_{k=1}^{r'} \sigma_k^\star \mathbf{u}_k^\star \mathbf{v}_k^\star - \sum_{k=1}^{r} \sigma_k^\star \mathbf{u}_k^\star \mathbf{v}_k^\star \right\|_F = \left\| \sum_{k=r+1}^{r'} \sigma_k^\star \mathbf{u}_k^\star \mathbf{v}_k^{\star\top} \right\|_F = \left( \sum_{k=r+1}^{r'} (\sigma_k^\star)^2 \right)^{1/2}. \quad (28)$$

Moreover, by Definition 1, we have

$$\left\| \sum_{k=1}^{r} \overline{\mathbf{b}}_k \overline{\mathbf{a}}_k^\top \mathbf{X} - \sum_{k=1}^{r} \mathbf{b}_k \mathbf{a}_k^\top \mathbf{X} \right\|_F \le \sum_{k=1}^{r} \|\mathbf{\Psi}_k \mathbf{X}\|_F \le \sigma_{\max}(\mathbf{X}) \sum_{k=1}^{r} \|\mathbf{\Psi}_k\|_F \quad (29)$$

Therefore, it suffices to study the second term in (27). Towards this end, we use the result in Lemma 2 and Lemma 3 to obtain

$$\left\| \mathbf{Y}_{k+1} - \mathbf{Y}_{k+1}^\star \right\|_F \le \left\| \mathbf{Y}_k - \mathbf{Y}_k^\star \right\|_F + \left\| \mathbf{b}_k^\star \mathbf{a}_k^{\star\top} \mathbf{X} - \overline{\mathbf{b}}_k \overline{\mathbf{a}}_k^\top \mathbf{X} \right\|_F + \|\mathbf{\Psi}_k \mathbf{X}\|_F \quad (30)$$

$$\left\| \mathbf{b}_k^\star \mathbf{a}_k^{\star\top} \mathbf{X} - \overline{\mathbf{b}}_k \overline{\mathbf{a}}_k^\top \mathbf{X} \right\|_F \le \left( \frac{3\sigma_k^\star}{\mathcal{T}_k} + 1 \right) \|\mathbf{Y}_k^\star - \mathbf{Y}_k\|_F. \quad (31)$$

Combining (30) and (31), we have that

$$\left\| \mathbf{Y}_{k+1} - \mathbf{Y}_{k+1}^\star \right\|_F \le \left( \frac{3\sigma_k^\star}{\mathcal{T}_k} + 2 \right) \|\mathbf{Y}_k^\star - \mathbf{Y}_k\|_F + \|\mathbf{\Psi}_k \mathbf{X}\|_F$$

Let the sequence $\{Q_k\}_{k=1}^r$ be defined as

$$Q_{k+1} = \alpha_k Q_k + \beta_k; \quad Q_0 = 0; \quad \alpha_k := 2 + \frac{3\sigma_k^\star}{\mathcal{T}_k}; \quad \beta_k := \|\mathbf{\Psi}_k \mathbf{X}\|_F$$

Then by inequality 19 we must have that $\|\mathbf{Y}_k - \mathbf{Y}_k^\star\|_F \le Q_k$ for all $k$. Invoking lemma 5 gives:

$$
\begin{aligned}
\|\mathbf{Y}_k - \mathbf{Y}_k^\star\|_F &\le \sum_{k'=0}^{k-1} \|\mathbf{\Psi}_{k'}\mathbf{X}\|_F \prod_{j=k'+1}^{k-1} \left(2 + \frac{3\sigma_j^\star}{\mathcal{T}_j}\right) \\
&\le \underbrace{\sigma_{\max}(\mathbf{X}) \sum_{k'=0}^{k-1} \|\mathbf{\Psi}_{k'}\|_F \prod_{j=k'+1}^{k-1} \left(2 + \frac{3\sigma_j^\star}{\mathcal{T}_j}\right)}_{:= \widehat{E}(k)}
\end{aligned}
\tag{32}
$$

We define the right-hand side of 32 to be equal to $\widehat{E}(k)$. Later in the proof, after lower-bounding the perturbed gaps by the population gaps, the theorem-level quantity $E(k)$ in the statement of Theorem 8 is seen to satisfy $E(k) \ge \widehat{E}(k)$. Therefore, the hypothesis

$$
E(k) < \frac{1}{2}\mathcal{T}_k^\star
$$

implies

$$
\|Y_k - Y_k^\star\|_2 \le \|Y_k - Y_k^\star\|_F \le \widehat{E}(k) \le E(k) < \mathcal{T}_k^\star = \min_{j>k} |\sigma_k^\star - \sigma_j^\star|.
\tag{33}
$$

Hence the perturbation condition required by Lemma 3 is satisfied. Combining (31) and (32), and notice that $\|\mathbf{\Psi}_k\mathbf{X}\|_F \le \sigma_{\max}(\mathbf{X})\|\mathbf{\Psi}_k\|_F$, we have:

$$
\left\|\mathbf{b}_k^\star \mathbf{a}_k^{\star\top}\mathbf{X} - \overline{\mathbf{b}}_k\overline{\mathbf{a}}_k^\top\mathbf{X}\right\|_F \le \sigma_{\max}(\mathbf{X})\left(\frac{3\sigma_k^\star}{\mathcal{T}_k} + 1\right) \sum_{k'=0}^{k-1} \|\mathbf{\Psi}_{k'}\|_F \prod_{j=k'+1}^{k-1} \left(2 + \frac{3\sigma_j^\star}{\mathcal{T}_j}\right)
\tag{34}
$$

Notice that $1 + \frac{3\sigma_k^\star}{\mathcal{T}_k} \le 2 + \frac{3\sigma_k^\star}{\mathcal{T}_k}$. Therefore, 34 becomes:

$$
\left\|\mathbf{b}_k^\star \mathbf{a}_k^{\star\top}\mathbf{X} - \overline{\mathbf{b}}_k\overline{\mathbf{a}}_k^\top\mathbf{X}\right\|_F \le \sigma_{\max}(\mathbf{X}) \sum_{k'=0}^{k-1} \|\mathbf{\Psi}_{k'}\|_F \prod_{j=k'+1}^{k} \left(2 + \frac{3\sigma_j^\star}{\mathcal{T}_j}\right)
\tag{35}
$$

Combining (28), (29), and (35) gives

$$
\begin{aligned}
\left\|\mathbf{Y}^{(r')} - \sum_{k=1}^{r} \mathbf{b}_k\mathbf{a}_k^\top\mathbf{X}\right\|_F &\le \left(\sum_{k=r+1}^{r'} (\sigma_k^\star)^2\right)^{1/2} + \sigma_{\max}(\mathbf{X}) \sum_{k=1}^{r} \|\mathbf{\Psi}_k\|_F \\
&\quad + \sigma_{\max}(\mathbf{X}) \sum_{k=1}^{r}\sum_{k'=0}^{k-1} \|\mathbf{\Psi}_{k'}\|_F \prod_{j=k'+1}^{k} \left(2 + \frac{3\sigma_j^\star}{\mathcal{T}_j}\right) \\
&= \left(\sum_{k=r+1}^{r'} (\sigma_k^\star)^2\right)^{1/2} + \sigma_{\max}(\mathbf{X}) \sum_{k=1}^{r}\sum_{k'=0}^{k} \|\mathbf{\Psi}_{k'}\|_F \prod_{j=k'+1}^{k} \left(2 + \frac{3\sigma_j^\star}{\mathcal{T}_j}\right)
\end{aligned}
$$

Finally, due to (33) and Weyl's inequality, we have

$$
|\sigma_{jk} - \sigma_j^\star| \le \|\mathbf{Y}_k - \mathbf{Y}_k^\star\|_2 < \tfrac{1}{2}\mathcal{T}_k^\star.
$$

Hence, for every $j > k$,

$$
|\sigma_k^\star - \sigma_{jk}| \ge |\sigma_k^\star - \sigma_j^\star| - |\sigma_j^\star - \sigma_{jk}| \ge \tfrac{1}{2}\mathcal{T}_k^\star,
$$

and therefore

$$
\mathcal{T}_k \ge \tfrac{1}{2}\mathcal{T}_k^\star.
$$

This allows us to define

$$
E(k) = \sigma_{\max}(\mathbf{X}) \sum_{k'=0}^{k-1} \|\mathbf{\Psi}_{k'}\|_F \prod_{j=k'+1}^{k-1} \left(\frac{6\sigma_j^\star}{\mathcal{T}_j^\star} + 2\right)
$$

and obtain that $E(k) \geq \hat{E}(k)$. Thus, enforcing $E(k) < \frac{1}{2}\mathcal{T}_k^\star$ suffices. Moreover, we have

$$\left\|\mathbf{Y}^{(r')} - \sum_{k=1}^{r} \mathbf{b}_k \mathbf{a}_k^\top \mathbf{X}\right\|_F \leq \left(\sum_{k=r+1}^{r'} (\sigma_k^\star)^2\right)^{1/2} + \sigma_{\max}(\mathbf{X}) \sum_{k=1}^{r} \sum_{k'=0}^{k} \|\mathbf{\Psi}_{k'}\|_F \prod_{j=k'+1}^{k} \left(2 + \frac{6\sigma_j^\star}{\mathcal{T}_j^\star}\right)$$

which completes the proof. $\square$

## C    Proof of Theorem 5 and Theorem 6

**Proof overview**. A rough idea of showing this theorem can build upon our characterization of the training error. Let $\hat{\mathbf{b}}_k^\star$'s be output of Algorithm 1 when using $\mathbf{Y}^\star$ as the label. We consider the following orthonormal basis of $\mathbb{R}^m$ extended from $\hat{\mathbf{b}}_k^\star$'s:

$$\hat{\mathbf{b}}_1, \ldots, \hat{\mathbf{b}}_m; \quad \hat{\mathbf{b}}_i = \hat{\mathbf{b}}_k^\star / \|\hat{\mathbf{b}}_k^\star\|_2 \text{ if } k \leq r^\star$$

Let $\hat{\mathbf{B}} \in \mathbb{R}^{m \times r^\star}$ consist of $\hat{\mathbf{b}}_1, \ldots \hat{\mathbf{b}}_{r^\star}$, and let $\hat{\mathbf{B}}^\perp \in \mathbb{R}^{m \times (m-r^\star)}$ consist of $\hat{\mathbf{b}}_{r^\star+1}, \ldots \hat{\mathbf{b}}_m$. Then, we can write $\mathbf{Y}^\star$ as:

$$\mathbf{Y} = \mathbf{W}^\star \mathbf{X} + \hat{\mathbf{B}}\hat{\mathbf{B}}^\top \boldsymbol{\mathcal{E}} + \hat{\mathbf{B}}^\perp \hat{\mathbf{B}}^{\perp\top} \boldsymbol{\mathcal{E}} = \hat{\mathbf{B}}(\mathbf{\Sigma}\hat{\mathbf{A}}^\top \mathbf{X} + \boldsymbol{\mathcal{E}}_1) + \hat{\mathbf{B}}^\perp \boldsymbol{\mathcal{E}}_2,$$

where $\boldsymbol{\mathcal{E}}_1 \in \mathbb{R}^{r^\star \times n}$ and $\boldsymbol{\mathcal{E}}_2 \in \mathbb{R}^{(m-r^\star) \times n}$ are noise matrices with I.I.D. Gaussian entries. Therefore, based on the above decomposition, $\boldsymbol{\mathcal{E}}_1$ can be seen as the unavoidable noise, which adds up to the training error, and $\boldsymbol{\mathcal{E}}_2$ is the error that can be avoided if we solve for only the top $r^\star$ components. Of course, $\hat{\mathbf{B}}(\mathbf{\Sigma}\hat{\mathbf{A}}^\top \mathbf{X} + \boldsymbol{\mathcal{E}}_1)$ is not the truncated top-$r^\star$ SVD of $\mathbf{Y}$ since $\mathbf{\Sigma}\hat{\mathbf{A}}^\top \mathbf{X} + \boldsymbol{\mathcal{E}}_1$ does not have orthogonal rows. However, when $\boldsymbol{\mathcal{E}}_1$ is small, this term approximates the truncated SVD well enough. Based on this intuition, we have the following lemma:

**Lemma 4.** *Let $\mathbf{Y}^\star$ to have the SVD $\mathbf{Y}^\star = \sum_{k=1}^{r} \hat{\sigma}_k \hat{\mathbf{u}}\hat{\mathbf{v}}^\star$, and let $\mathbf{Y} = \mathbf{Y}^\star + \boldsymbol{\mathcal{E}}$ to have SVD $\mathbf{Y} = \sum_{k=1}^{m} \sigma_k^\star \mathbf{u}_k^\star \mathbf{v}_k^\star$. Let $\mathbf{Y}^{\star(\hat{m})}$ and $\mathbf{Y}^{(\hat{m})}$ be the truncated rank-$\hat{m}$ SVD of $\mathbf{Y}^\star$ and $\mathbf{Y}$, respectively. Then with probability at least $1 - \delta$ we have that*

$$\left\|\mathbf{Y}^{\star(\hat{m})} - \mathbf{Y}^{(\hat{m})}\right\|_F \leq O\left(\varepsilon\sqrt{n\log 1/\delta}\left(\hat{m} + \sqrt{\frac{\min\{r, \hat{m}\}}{\mathcal{T}_{\min}^\star}}\right)\right)$$

The proof of Lemma 4 is given in Appendix C.2. With the help of Lemma 4, the proof of Theorem 6 involves choosing a reference label $\mathbf{Y}^{(r)}$ that involves only the relevant noise that will be fitted by Algorithm 2. We then control the parameter-recovery error by estimating the difference between $\mathbf{W}^\star \mathbf{X}$ and $\mathbf{Y}^{\star(r)}$, and the difference between $\mathbf{Y}^{\star(r)}$ and $\mathbf{Y}^{(r)}$ using Lemma 4.

### C.1    More details in the proof of Theorem 5

By the triangle inequality, Definition 1, and Lemma 3, we have

$$\|\mathbf{b}_k^\star \mathbf{a}_k^{\star\top} \mathbf{X} - \mathbf{b}_k \mathbf{a}_k^\top \mathbf{X}\|_F \leq \|\mathbf{b}_k^\star \mathbf{a}_k^{\star\top} \mathbf{X} - \overline{\mathbf{b}}_k \overline{\mathbf{a}}_k^\top \mathbf{X}\|_F + \|\overline{\mathbf{b}}_k \overline{\mathbf{a}}_k^\top \mathbf{X} - \mathbf{b}_k \mathbf{a}_k^\top \mathbf{X}\|_F.$$

Using Lemma 3 for the first term and Definition 1 for the second term gives

$$\|\mathbf{b}_k^\star \mathbf{a}_k^{\star\top} \mathbf{X} - \mathbf{b}_k \mathbf{a}_k^\top \mathbf{X}\|_F \leq \left(1 + \frac{3\sigma_k^\star}{\mathcal{T}_k}\right) \|\mathbf{Y}_k^\star - \mathbf{Y}_k\|_F + \|\mathbf{\Psi}_k \mathbf{X}\|_F.$$

Now applying (32) and using $\|\mathbf{\Psi}_k \mathbf{X}\|_F \leq \sigma_{\max}(\mathbf{X})\|\mathbf{\Psi}_k\|_F$, we obtain

$$\|\mathbf{b}_k^\star \mathbf{a}_k^{\star\top} \mathbf{X} - \mathbf{b}_k \mathbf{a}_k^\top \mathbf{X}\|_F \leq \sigma_{\max}(\mathbf{X}) \left(1 + \frac{3\sigma_k^\star}{\mathcal{T}_k}\right) \sum_{k'=0}^{k-1} \|\mathbf{\Psi}_{k'}\|_F \prod_{j=k'+1}^{k-1} \left(2 + \frac{3\sigma_j^\star}{\mathcal{T}_j}\right) + \sigma_{\max}(\mathbf{X})\|\mathbf{\Psi}_k\|_F.$$

Using $1 + \frac{3\sigma_k^\star}{\mathcal{T}_k} \leq 2 + \frac{3\sigma_k^\star}{\mathcal{T}_k}$, this implies

$$\|\mathbf{b}_k^\star \mathbf{a}_k^{\star\top} \mathbf{X} - \mathbf{b}_k \mathbf{a}_k^\top \mathbf{X}\|_F \leq \sigma_{\max}(\mathbf{X}) \sum_{k'=0}^{k-1} \|\mathbf{\Psi}_{k'}\|_F \prod_{j=k'+1}^{k} \left(2 + \frac{3\sigma_j^\star}{\mathcal{T}_j}\right) + \sigma_{\max}(\mathbf{X})\|\mathbf{\Psi}_k\|_F.$$

Adopting the empty-product convention $\prod_{j=k+1}^{k}(\cdot) = 1$, we can absorb the last term into the sum and write

$$\|\mathbf{b}_k^\star \mathbf{a}_k^{\star\top}\mathbf{X} - \mathbf{b}_k\mathbf{a}_k^\top\mathbf{X}\|_F \leq \sigma_{\max}(\mathbf{X})\sum_{k'=0}^{k}\|\mathbf{\Psi}_{k'}\|_F\prod_{j=k'+1}^{k}\left(2 + \frac{3\sigma_j^\star}{\mathcal{T}_j}\right).$$

Finally, using the lower bound $\mathcal{T}_j \geq \frac{1}{2}\mathcal{T}_j^\star$ established in Appendix B.4, we obtain

$$\|\mathbf{b}_k^\star \mathbf{a}_k^{\star\top}\mathbf{X} - \mathbf{b}_k\mathbf{a}_k^\top\mathbf{X}\|_F \leq \sigma_{\max}(\mathbf{X})\sum_{k'=0}^{k}\|\mathbf{\Psi}_{k'}\|_F\prod_{j=k'+1}^{k}\left(2 + \frac{6\sigma_j^\star}{\mathcal{T}_j^\star}\right).$$

Since $\sigma_{\min}(\mathbf{X}) > 0$, we can then have

$$\left\|\mathbf{b}_k^\star \mathbf{a}_k^{\star\top}\mathbf{X} - \mathbf{b}_k\mathbf{a}_k^\top\mathbf{X}\right\|_F \geq \sigma_{\min}(\mathbf{X})\left\|\mathbf{b}_k^\star \mathbf{a}_k^{\star\top} - \mathbf{b}_k\mathbf{a}_k^\top\right\|_F$$

$$\Rightarrow \left\|\mathbf{b}_k^\star \mathbf{a}_k^{\star\top} - \mathbf{b}_k\mathbf{a}_k^\top\right\|_F \leq \frac{1}{\sigma_{\min}(\mathbf{X})}\left\|\mathbf{b}_k^\star \mathbf{a}_k^{\star\top}\mathbf{X} - \mathbf{b}_k\mathbf{a}_k^\top\mathbf{X}\right\|_F$$

This implies that

$$\left\|\mathbf{b}_k^\star \mathbf{a}_k^{\star\top} - \mathbf{b}_k\mathbf{a}_k^\top\right\|_F \leq \kappa(\mathbf{X})\sum_{k'=0}^{k}\|\mathbf{\Psi}_{k'}\|_F\prod_{j=k'+1}^{k}\left(2 + \frac{6\sigma_j^\star}{\mathcal{T}_j^\star}\right)$$

which proves the first statement. To prove the second statement, we directly use Theorem 1 to get that

$$\left\|\mathbf{W}^\star - \sum_{k=1}^{r}\mathbf{b}_k\mathbf{a}_k^\top\right\|_F \leq \frac{1}{\sigma_{\min}(\mathbf{X})}\left\|\mathbf{W}^\star\mathbf{X} - \sum_{k=1}^{r}\mathbf{b}_k\mathbf{a}_k^\top\mathbf{X}\right\|_F$$

$$= \frac{1}{\sigma_{\min}(\mathbf{X})}\left\|\mathbf{Y} - \sum_{k=1}^{r}\mathbf{b}_k\mathbf{a}_k^\top\mathbf{X}\right\|_F$$

$$\leq \frac{\left(\sum_{k=r+1}^{r^\star}(\sigma_k^\star)^2\right)^{1/2}}{\sigma_{\min}(\mathbf{X})} + \kappa(\mathbf{X})\sum_{k=1}^{r}\sum_{k'=1}^{k}\|\mathbf{\Psi}_{k'}\|_F\prod_{j=k'+1}^{k}\left(2 + \frac{6\sigma_j^\star}{\mathcal{T}_j^\star}\right)$$

## C.2   Proof of Lemma 4

*Proof.* By Lemma 6, we have that with probability $1 - \delta$

$$\sigma_{\max}(\mathcal{E}) \leq O\left(\varepsilon\left(\sqrt{n} + \sqrt{\log\frac{1}{\delta}}\right)\right)$$

To start, by Weyl's inequality, we have that

$$|\hat{\sigma}_k - \sigma_k^\star| \leq \sigma_{\max}(\mathcal{E}) \leq O\left(\varepsilon\left(\sqrt{n} + \sqrt{\log\frac{1}{\delta}}\right)\right)$$

Therefore, taking

$$\varepsilon \leq c_{\mathrm{ord}}\frac{\mathcal{T}_{\min}^\star}{\sqrt{n} + \sqrt{\log(1/\delta)}}$$

for a sufficiently small absolute constant $c_{\mathrm{ord}} > 0$ ensures that

$$\min\left\{\min_{j\neq k}|\sigma_j - \sigma_k^\star|,\ \sigma_k^\star\right\} \geq \frac{1}{2}\mathcal{T}_{\min}^\star.$$

This is exactly the ordering condition stated in Theorem 6.  Thus, by Wedin's Theorem, we have that

$$\hat{\mathbf{u}}_k^\top\mathbf{u}_k^\star + \hat{\mathbf{v}}_k^\top\mathbf{v}_k^\star \leq 2 - \frac{2}{\mathcal{T}_k^\star}\left(\left\|\mathcal{E}^\top\mathbf{u}_k^\star\right\|_2^2 + \|\mathcal{E}\mathbf{v}_k^\star\|_2^2\right)$$

We will consider two cases. First, when $\hat{m} \leq r$, we have

$$
\begin{aligned}
\left\| \mathbf{Y}^{\star(\hat{m})} - \mathbf{Y}^{(\hat{m})} \right\|_F &= \left\| \sum_{k=1}^{\hat{m}} \left( \sigma_k^\star \mathbf{u}^\star \mathbf{v}^\star - \hat{\sigma}_k \hat{\mathbf{u}}_k \hat{\mathbf{v}}_k \right) \right\|_F \\
&\leq \left\| \sum_{k=1}^{\hat{m}} \left( \hat{\sigma}_k - \sigma_k^\star \right) \mathbf{u}_k \mathbf{v}_k \right\|_F + \left\| \hat{\mathbf{U}}_{\hat{m}} \mathbf{\Sigma}_{\hat{m}}^\star \hat{\mathbf{V}}_{\hat{m}}^\top - \mathbf{U}_{\hat{m}}^\star \mathbf{\Sigma}_{\hat{m}}^\star \hat{\mathbf{V}}_{\hat{m}}^{\top\star} \right\|_F \\
&\leq \sum_{r=1}^{\hat{m}} |\hat{\sigma}_k - \sigma_k^\star| + \left\| \left( \hat{\mathbf{U}}_{\hat{m}} - \mathbf{U}_{\hat{m}}^\star \right) \mathbf{\Sigma}_{\hat{m}}^\star \right\|_F + \left\| \left( \hat{\mathbf{V}}_{\hat{m}} - \mathbf{V}_{\hat{m}}^\star \right) \mathbf{\Sigma}_{\hat{m}}^\star \right\|_F \\
&\leq \sum_{r=1}^{\hat{m}} |\hat{\sigma}_k - \sigma_k^\star| + \sigma_1^\star \left( \left\| \hat{\mathbf{U}}_{\hat{m}} - \mathbf{U}_{\hat{m}}^\star \right\|_F + \left\| \hat{\mathbf{V}}_{\hat{m}} - \mathbf{V}_{\hat{m}}^\star \right\|_F \right) \\
&\leq \sum_{r=1}^{\hat{m}} |\hat{\sigma}_k - \sigma_k^\star| + 2\sigma_1^\star \left( \left\| \hat{\mathbf{U}}_{\hat{m}} - \mathbf{U}_{\hat{m}}^\star \right\|_F^2 + \left\| \hat{\mathbf{V}}_{\hat{m}} - \mathbf{V}_{\hat{m}}^\star \right\|_F^2 \right)^{\frac{1}{2}}
\end{aligned}
$$

Notice that

$$
\left\| \hat{\mathbf{U}}_{\hat{m}} - \mathbf{U}_{\hat{m}}^\star \right\|_F^2 = 2\hat{m} - 2 \left\langle \hat{\mathbf{U}}_{\hat{m}}, \mathbf{U}_{\hat{m}}^\star \right\rangle = 2\hat{m} - 2 \sum_{k=1}^{\hat{m}} \hat{\mathbf{u}}_k^\top \mathbf{u}_k^\star
$$

$$
\left\| \hat{\mathbf{V}}_{\hat{m}} - \mathbf{V}_{\hat{m}}^\star \right\|_F^2 = 2\hat{m} - 2 \left\langle \hat{\mathbf{V}}_{\hat{m}}, \mathbf{V}_{\hat{m}}^\star \right\rangle = 2\hat{m} - 2 \sum_{k=1}^{\hat{m}} \hat{\mathbf{v}}_k^\top \mathbf{v}_k^\star
$$

Therefore

$$
\begin{aligned}
\left\| \hat{\mathbf{U}}_{\hat{m}} - \mathbf{U}_{\hat{m}}^\star \right\|_F^2 + \left\| \hat{\mathbf{V}}_{\hat{m}} - \mathbf{V}_{\hat{m}}^\star \right\|_F^2 &\leq 4k - 2 \sum_{k=1}^{\hat{m}} \left( \hat{\mathbf{u}}_k^\top \mathbf{u}_k^\star + \hat{\mathbf{v}}_k^\top \mathbf{v}_k^\star \right) \\
&\leq \frac{4}{\mathcal{T}_{\min}^\star} \sum_{k=1}^{\hat{m}} \left( \left\| \mathbf{\mathcal{E}}^\top \mathbf{u}_k^\star \right\|_2^2 + \| \mathbf{\mathcal{E}} \mathbf{v}_k^\star \|_2^2 \right) \\
&= \frac{4}{\mathcal{T}_{\min}^\star} \left( \left\| \mathbf{\mathcal{E}}^\top \mathbf{U}_{\hat{m}}^\star \right\|_F^2 + \| \mathbf{\mathcal{E}} \mathbf{V}_{\hat{m}}^\star \|_F^2 \right)
\end{aligned}
$$

Since $\mathbf{\mathcal{E}} \in \mathbb{R}^{m \times n}$ contains I.I.D. Gaussian entries from $\mathcal{N}(0, \varepsilon^2)$, we must have that $\mathbf{U}_k^\star \mathbf{\mathcal{E}} \in \mathbb{R}^{\hat{m} \times n}$ and $\mathbf{\mathcal{E}} \mathbf{V}_k^\star \in \mathbb{R}^{m \times \hat{m}}$ contains I.I.D. Gaussian entries from $\mathcal{N}(0, \varepsilon^2)$. By Lemma 6, we have that with probability at least $1 - \delta$, it holds that

$$
\left\| \mathbf{U}_{\hat{m}}^{\star\top} \mathbf{\mathcal{E}} \right\|_F^2 + \| \mathbf{\mathcal{E}} \mathbf{V}_{\hat{m}}^\star \|_F^2 \leq O \left( \varepsilon^2 (m + n) \hat{m} \log 1/\delta \right)
$$

Thus, we have

$$
\left\| \hat{\mathbf{U}}_{\hat{m}} - \mathbf{U}_{\hat{m}}^\star \right\|_F^2 + \left\| \hat{\mathbf{V}}_{\hat{m}} - \mathbf{V}_{\hat{m}}^\star \right\|_F^2 \leq O \left( \frac{\varepsilon^2}{\mathcal{T}_{\min}^\star} \hat{m}(m + n) \log 1/\delta \right)
$$

Combining the results above, we have

$$
\left\| \mathbf{Y}^{\star(\hat{m})} - \mathbf{Y}^{(\hat{m})} \right\|_F \leq O \left( \varepsilon \hat{m} \left( \sqrt{n} + \sqrt{\log 1/\delta} \right) \right) + O \left( \frac{\varepsilon}{\sqrt{\mathcal{T}_{\min}^\star}} \sqrt{r(m + n) \log 1/\delta} \right)
$$

Next, we consider the case $\hat{m} \geq r$. In this case, we have

$$
\begin{aligned}
\left\| \mathbf{Y}^{\star(\hat{m})} - \mathbf{Y}^{(\hat{m})} \right\|_F &\leq \left\| \mathbf{Y}^\star - \mathbf{Y}^{(r)} \right\|_F + \left\| \sum_{k=r+1}^{\hat{m}} \sigma_k \mathbf{u}_k \mathbf{v}_k^\top \right\|_F \\
&\leq O \left( \varepsilon r \left( \sqrt{n} + \sqrt{\log 1/\delta} \right) \right) + O \left( \frac{\varepsilon}{\sqrt{\mathcal{T}_{\min}^\star}} \sqrt{r(m + n) \log 1/\delta} \right) + \sum_{k=r+1}^{\hat{m}} \sigma_k
\end{aligned}
$$

Notice that by Weyl's inequality, for all $k \geq r$

$$\sigma_k \leq |\sigma_k - 0| \leq \sigma_{\max}(\boldsymbol{\mathcal{E}}) \leq O\left(\varepsilon\left(\sqrt{n} + \sqrt{\log\frac{1}{\delta}}\right)\right)$$

This gives

$$\left\|\mathbf{Y}^{\star(\hat{m})} - \mathbf{Y}^{(\hat{m})}\right\|_F \leq O\left(\varepsilon\hat{m}\left(\sqrt{n} + \sqrt{\log 1/\delta}\right)\right) + O\left(\frac{\varepsilon}{\sqrt{\mathcal{T}_{\min}^{\star}}}\sqrt{r(m+n)\log 1/\delta}\right)$$

Combining the two cases, and using $m \leq n$, we have that

$$\left\|\mathbf{Y}^{\star(\hat{m})} - \mathbf{Y}^{(\hat{m})}\right\|_F \leq O\left(\varepsilon\sqrt{n\log 1/\delta}\left(\hat{m} + \sqrt{\frac{\min\{r,\hat{m}\}}{\mathcal{T}_{\min}^{\star}}}\right)\right)$$

$\square$

## C.3 Proof of Theorem 6

Given the SVD of $\mathbf{Y}$ and $\mathbf{Y}^{\star}$ as $\mathbf{Y} = \sum_{k=1}^{p}\sigma_k^{\star}\mathbf{u}_k^{\star}\mathbf{v}_k^{\star}$ and $\mathbf{Y}^{\star} = \sum_{k=1}^{r^{\star}}\hat{\sigma}_k\hat{\mathbf{u}}_k\hat{\mathbf{v}}_k^{\top}$, we define

$$\mathbf{Y}^{(r)} = \sum_{k=1}^{r}\sigma_k^{\star}\mathbf{u}_k^{\star}\mathbf{v}_k^{\star}; \quad \mathbf{Y}^{\star(r)} = \sum_{k=1}^{\min\{r,r^{\star}\}}\hat{\sigma}_k\hat{\mathbf{u}}_k\hat{\mathbf{v}}_k^{\top}$$

Then we can decompose the error $\left\|\mathbf{W}^{\star}\mathbf{X} - \sum_{k=1}^{r}\mathbf{b}_k\mathbf{a}_k^{\top}\mathbf{X}\right\|_F$ as

$$\left\|\mathbf{W}^{\star}\mathbf{X} - \sum_{k=1}^{r}\mathbf{b}_k\mathbf{a}_k^{\top}\mathbf{X}\right\|_F \leq \left\|\mathbf{W}^{\star}\mathbf{X} - \mathbf{Y}^{\star(r)}\right\|_F + \left\|\mathbf{Y}^{\star(r)} - \mathbf{Y}^{(r)}\right\|_F + \left\|\mathbf{Y}^{(r)} - \sum_{k=1}^{r}\mathbf{b}_k\mathbf{a}_k^{\top}\mathbf{X}\right\|_F$$

We will analyze each of the three terms individually. To start, for the first term, we notice that $\mathbf{Y}^{\star(r)}$ is precisely the truncated SVD of $\mathbf{W}^{\star}\mathbf{X}$ when $r < r^{\star}$. Therefore

$$\left\|\mathbf{W}^{\star}\mathbf{X} - \mathbf{Y}^{\star(r)}\right\|_F = \left(\sum_{k=r+1}^{r^{\star}}\sigma_k(\mathbf{W}^{\star}\mathbf{X})^2\right)^{1/2} \leq \sigma_{\max}(\mathbf{X})\left(\sum_{k=r+1}^{r^{\star}}\sigma_k(\mathbf{W}^{\star})^2\right)^{1/2}.$$

For the second term, by Lemma 4, we have that with probability at least $1 - \gamma$

$$\left\|\mathbf{Y}^{\star(r)} - \mathbf{Y}^{(r)}\right\|_F \leq O\left(\varepsilon\sqrt{n\log 1/\gamma}\left(r + \sqrt{\frac{\min\{r^{\star},r\}}{\mathcal{T}_{\min}^{\star}}}\right)\right)$$

Lastly, by Theorem 8, we have that

$$\left\|\mathbf{Y}^{(r)} - \sum_{k=1}^{r}\mathbf{b}_k\mathbf{a}_k^{\top}\mathbf{X}\right\|_F \leq \sigma_{\max}(\mathbf{X})\sum_{k=1}^{r}\sum_{k'=0}^{k}\|\boldsymbol{\Psi}_{k'}\|_F\prod_{j=k'+1}^{k}\left(2 + \frac{6\sigma_j^{\star}}{\mathcal{T}_j^{\star}}\right)$$

Combining the above equations, we have that

$$\left\|\mathbf{W}^{\star}\mathbf{X} - \sum_{k=1}^{r}\mathbf{b}_k\mathbf{a}_k^{\top}\mathbf{X}\right\|_F \leq \sigma_{\max}(\mathbf{X})\left(\sum_{k=r+1}^{r^{\star}}\sigma_k(\mathbf{W}^{\star})^2\right)^{1/2} + \sigma_{\max}(\mathbf{X})\sum_{k=1}^{r}\sum_{k'=0}^{k}\|\boldsymbol{\Psi}_{k'}\|_F\prod_{j=k'+1}^{k}\left(2 + \frac{6\sigma_j^{\star}}{\mathcal{T}_j^{\star}}\right)$$

$$+ O\left(\varepsilon\sqrt{n\log 1/\gamma}\left(r + \sqrt{\frac{\min\{r^{\star},r\}}{\mathcal{T}_{\min}^{\star}}}\right)\right)$$

This gives that

$$\left\| \mathbf{W}^\star - \sum_{k=1}^r \mathbf{b}_k \mathbf{a}_k^\top \right\|_F \le \kappa(\mathbf{X}) \left( \left( \sum_{k=r+1}^{r^\star} \sigma_k(\mathbf{W}^\star)^2 \right)^{1/2} + \sum_{k=1}^r \sum_{k'=0}^k \|\mathbf{\Psi}_{k'}\|_F \prod_{j=k'+1}^k \left( 2 + \frac{6\sigma_j^\star}{\mathcal{T}_j^\star} \right) \right)$$
$$+ O \left( \frac{\varepsilon \sqrt{n \log 1/\gamma}}{\sigma_{\min}(\mathbf{X})} \left( r + \sqrt{\frac{\min\{r^\star, r\}}{\mathcal{T}_{\min}^\star}} \right) \right)$$

which completes the proof.

## D    Supporting theorems and lemmas

**Lemma 5.** *Consider a sequence of quantities $\{Q_k\}_{k=1}^\infty$ satisfying*

$$Q_{k+1} = \mathbf{a}_k Q_k + \mathbf{b}_k$$

*with some $\mathbf{a}_k, \mathbf{b}_k \ge 0$ for all $k \in \mathbb{Z}^+$. Set $b_0 = Q_1$. Then we have that*

$$Q_k = \sum_{k'=0}^{k-1} b_{k'} \prod_{j=k'+1}^{k-1} a_j$$

*Proof.* We shall prove by induction. For the base case, let $k = 1$. In this case, we have that

$$Q_1 = \sum_{k'=0}^0 b_{k'} \prod_{j=k'+1}^0 a_j = b_0 = Q_1$$

For the inductive case, assume that the property holds for $k$. Then we have that

$$Q_{k+1} = \mathbf{a}_k Q_k + \mathbf{b}_k = \mathbf{a}_k \cdot \sum_{k'=0}^{k-1} b_{k'} \prod_{j=k'+1}^{k-1} a_j + \mathbf{b}_k = \sum_{k'=0}^k b_{k'} \prod_{j=k'+1}^k a_j$$

This proves the inductive step and finishes the proof. $\square$

**Lemma 6.** *Let $\mathbf{M} \in \mathbb{R}^{m \times n}$ be a matrix containing I.I.D. Gaussian entries from $\mathcal{N}(0, 1)$. Then we have that with probability at least $1 - \delta$, the following holds*

- $\sigma_{\max}(\mathbf{M}) \le O \left( \sqrt{m} + \sqrt{n} + \sqrt{\log 1/\delta} \right)$

- $\|\mathbf{M}\|_F \le O \left( \sqrt{mn \log 1/\delta} \right)$

*Proof.* By standard results of Gaussian random matrices and vectors, we have that

- $\mathbb{P} \left( \sigma_{\max}(\mathbf{M}) \le O \left( \sqrt{m} + \sqrt{n} + t_1 \right) \right) \ge 1 - \exp \left( -t_1^2 \right)$

- $\mathbb{P} \left( \|\mathbf{M}\|_F \le t_2 \right) \ge 1 - 2 \exp \left( -\frac{t_2^2}{2mn} \right)$

Take $t_1 = \sqrt{\log \frac{2}{\delta}}$ and $t_2 = \sqrt{2mn \log \frac{4}{\delta}}$ finishes the proof. $\square$

**Theorem 9** (Eckart-Young-Mirsky Theorem)**.** *Let $\mathbf{A} \in \mathbb{R}^{m \times n}$ be a matrix with singular value decomposition $\mathbf{A} = \mathbf{U} \mathbf{\Sigma} \mathbf{V}^\top$, where $\mathbf{U}$ and $\mathbf{V}$ are orthogonal matrices and $\mathbf{\Sigma}$ is a diagonal matrix with singular values $\sigma_1 \ge \sigma_2 \ge \cdots \ge \sigma_{\min(m,n)} \ge 0$. For any integer $k \le \min(m, n)$, let $\mathbf{A}_k = \mathbf{U}_k \mathbf{\Sigma}_k \mathbf{V}_k^\top$ be the best rank-$k$*

approximation of $\mathbf{A}$, where $\mathbf{U}_k$ and $\mathbf{V}_k$ consist of the first $k$ columns of $\mathbf{U}$ and $\mathbf{V}$, and $\mathbf{\Sigma}_k$ is the diagonal matrix of the largest $k$ singular values.

Then $\mathbf{A}_k$ minimizes the approximation error in both the Frobenius norm and the spectral norm:

$$\mathbf{A}_k = \underset{\mathbf{B}, rank(\mathbf{B})=k}{\arg\min} \|\mathbf{A} - \mathbf{B}\|_F$$

**Theorem 10** (Wedin's theorem; (Wedin, 1972)). *Let* $\mathbf{M}$, $\tilde{\mathbf{M}} \in \mathbb{R}^{m \times n}$ *be two matrices with rank-r SVDs:*

$$\mathbf{M} = \begin{bmatrix} \mathbf{U}_1 & \mathbf{U}_2 \end{bmatrix} \begin{bmatrix} \mathbf{\Sigma}_1 & 0 \\ 0 & \mathbf{\Sigma}_2 \end{bmatrix} \begin{bmatrix} \mathbf{V_1}^\top \\ \mathbf{V_2}^\top \end{bmatrix}, \quad and \quad \tilde{\mathbf{M}} = \mathbf{M} + \mathbf{\Delta} = \begin{bmatrix} \tilde{\mathbf{U}}_1 & \tilde{\mathbf{U}}_2 \end{bmatrix} \begin{bmatrix} \tilde{\mathbf{\Sigma}}_1 & 0 \\ 0 & \tilde{\mathbf{\Sigma}}_2 \end{bmatrix} \begin{bmatrix} \tilde{\mathbf{V}}_1^\top \\ \tilde{\mathbf{V}}_2^\top \end{bmatrix}.$$

*If* $\delta = \min \{\min_{1 \le i \le r, r+1 \le j \le n} |\sigma_i - \tilde{\sigma}_j|, \min_{1 \le i \le r} \sigma_i\} > 0$, *then:*

$$\left\|\sin\theta(\tilde{\mathbf{U}}_1, \mathbf{U}_1)\right\|_F^2 + \left\|\sin\theta(\tilde{\mathbf{V}}_1, \mathbf{V}_1)\right\|_F^2 \le \frac{\left\|\mathbf{U}_1^\top \mathbf{\Delta}\right\|_F^2 + \|\mathbf{\Delta} \mathbf{V}_1\|_F^2}{\delta^2}$$

In the rank-1 case, where $\boldsymbol{U}_1 = \boldsymbol{u}$, $\boldsymbol{V}_1 = \boldsymbol{v}$, $\tilde{\boldsymbol{U}}_1 = \tilde{\boldsymbol{u}}$, and $\tilde{\boldsymbol{V}}_1 = \tilde{\boldsymbol{v}}$, Theorem 10 reduces to

$$\sin^2 \angle(\tilde{\boldsymbol{u}}, \boldsymbol{u}) + \sin^2 \angle(\tilde{\boldsymbol{v}}, \boldsymbol{v}) \le \frac{\|\boldsymbol{u}^\top \mathbf{\Delta}\|_2^2 + \|\mathbf{\Delta} \boldsymbol{v}\|_2^2}{\delta^2}.$$

This is the form used in the proof of Lemma 3.

**Theorem 11** (Weyl's theorem for singular values; (Weyl, 1912)). *Let* $\mathbf{M}$ *and* $\mathbf{\Delta}$ *be* $m \times n$ *matrices. If* $\tilde{\mathbf{M}} = \mathbf{M} + \mathbf{\Delta}$, *then the singular values* $\sigma_i$ *of* $\mathbf{M}$ *and the singular values* $\tilde{\sigma}_i$ *of* $\tilde{\mathbf{M}}$ *satisfy the following inequality for all* $i = 1, 2, \ldots, \min(m, n)$:

$$|\tilde{\sigma}_i - \sigma_i| \le \|\mathbf{\Delta}\|_2,$$

# E  Experimental details and reproducibility.

All experiments use the common schedule family from section 5 to compare equal, more-first, and less-first allocations of the training budget across components via the $\alpha$-scheduler. Unless noted otherwise, synthetic linear experiments use rank $r = 20$, while non-linear LoRA experiments use $r = 3$. Joint LoRA trains all rank-one components simultaneously, whereas sequential LoRA adds one component at a time while freezing previously learned components. We use seed 42 by default and seeds $43 \to 46$ for more trials. Performance is reported on a held-out test split at the end of training.

- **Synthetic experiments.** We use full-batch gradient descent on rank-one factors with learning rates $\eta_A = \eta_B = 0.003$. We set $m = 500$, $d = 1000$, $n = 2500$, and $r = 20$. The design matrix $\mathbf{X}$ is sampled from a standard Gaussian and column-normalized, and the target matrix has an exponential singular-value profile. The total training budget is 500 iterations, with at least two iterations per component. We run five trials with different random seeds and report means and standard deviations where appropriate. For each component, training stops when its allocated budget is exhausted or when the absolute change in training error drops below $10^{-10}$.

- **MNIST.** We use a three-layer MLP $784 \to 512 \to 512 \to 10$ with ReLU activations. The base model is trained on classes 0–4, and LoRA adaptation is evaluated on classes 5–9. Inputs are normalized. We use batch size 64, Adam with learning rate $10^{-3}$, and seed 42. The base model is trained for 10 epochs. In the ablations, joint LoRA budgets are $2, 4, \ldots, 14$ epochs, and sequential budgets are $6, 12, \ldots, 40$ epochs, with $\alpha \in \{0, 0.2, \ldots, 2.0\}$ and the corresponding negative values. For the single-budget results, we use 10 epochs for joint LoRA and choose the corresponding sequential budget to match adaptation-training compute.

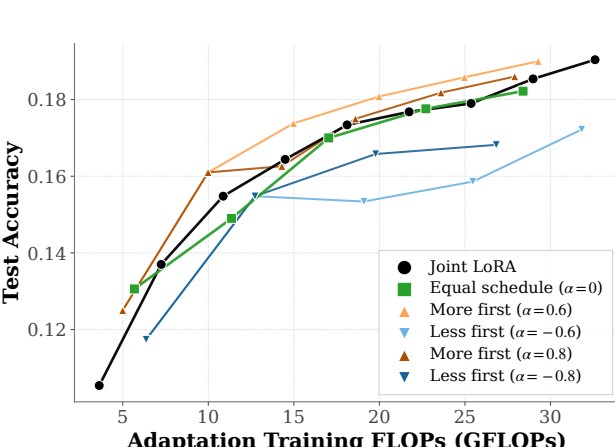

Figure 8: Compute-matched accuracy comparison for the CIFAR100 vision experiment from Section 5.2. The black curve gives jointly trained LoRA as the adaptation budget varies, while the colored curves show sequential LoRA under realized schedules from the common $\alpha$-family. Each point corresponds to one training budget under the same model and evaluation protocol, isolating how schedule choice affects accuracy at a given adaptation compute budget. Across much of the range, more-first schedules achieve higher accuracy than less-first schedules, with the equal schedule typically in between; the strongest more-first runs reach the same upper accuracy range as the joint LoRA reference. This figure complements the main-text schedule sweep by making the comparison to the joint baseline explicit on the FLOPs scale.

- **CIFAR-10.** We follow the MNIST setup, except that the MLP input dimension is 3072, inputs are normalized using CIFAR-10 channel statistics, and adaptation is performed from classes 0–4 to 5–9. All other training details remain the same.

- **CIFAR-100.** We follow the CIFAR-10 setup, except that the output dimension is 100, the base model is trained on classes 0–49, and adaptation is evaluated on classes 50–99. All other training details remain the same.

- **SST-2.** We use `distilbert-base-uncased` as the base model, with LoRA inserted into the query and value projections of all six transformer blocks at rank $r = 3$. We train on the SST-2 training split and evaluate on the GLUE validation split. Inputs are tokenized to length 128. Training uses the HuggingFace `Trainer` with AdamW, learning rate $2 \times 10^{-5}$, weight decay 0.01, batch size 32, and mixed precision. The joint and sequential epoch budgets match those used in the vision ablations.

## F   Additional compute analysis for the nonlinear PEFT experiments

To make the compute comparisons in Section 5.2 explicit, this appendix serves two purposes. First, Figure 9 reports the FLOP accounting used for the nonlinear PEFT experiments. Second, Figure 8 compares sequential LoRA with jointly trained LoRA at matched adaptation-training compute budgets in the CIFAR100 setting.

**Per-sample FLOP accounting.** Figure 9 separates the forward and backward contributions to adaptation-training FLOPs. The forward pass is shared across methods, while the backward/update cost differs because sequential LoRA optimizes one rank-1 component at a time rather than updating all rank directions jointly. For the per-sample adaptation FLOP accounting considered here, sequential LoRA uses fewer FLOPs than joint LoRA once $r \geq 2$, and the relative reduction increases with the target rank over the plotted range. This makes the cost model behind the compute-aware comparisons in Section 5.2 explicit.

**Compute-matched accuracy on CIFAR100.** Using the FLOP accounting above, Figure 8 compares jointly trained LoRA and sequential LoRA at matched adaptation-training compute in the CIFAR100 vision experiment. This figure makes the fairness criterion explicit by comparing methods at matched adaptation compute. If total adaptation compute were the only relevant factor, then runs with similar FLOPs would achieve similar accuracy regardless of how the budget is distributed across the sequential components. Instead, the figure shows that schedule choice still matters after normalizing by compute. In particular, more-first schedules tend to outperform less-first schedules over much of the plotted range, with the equal schedule typically in between. At the same time, the strongest more-first runs reach the same upper accuracy

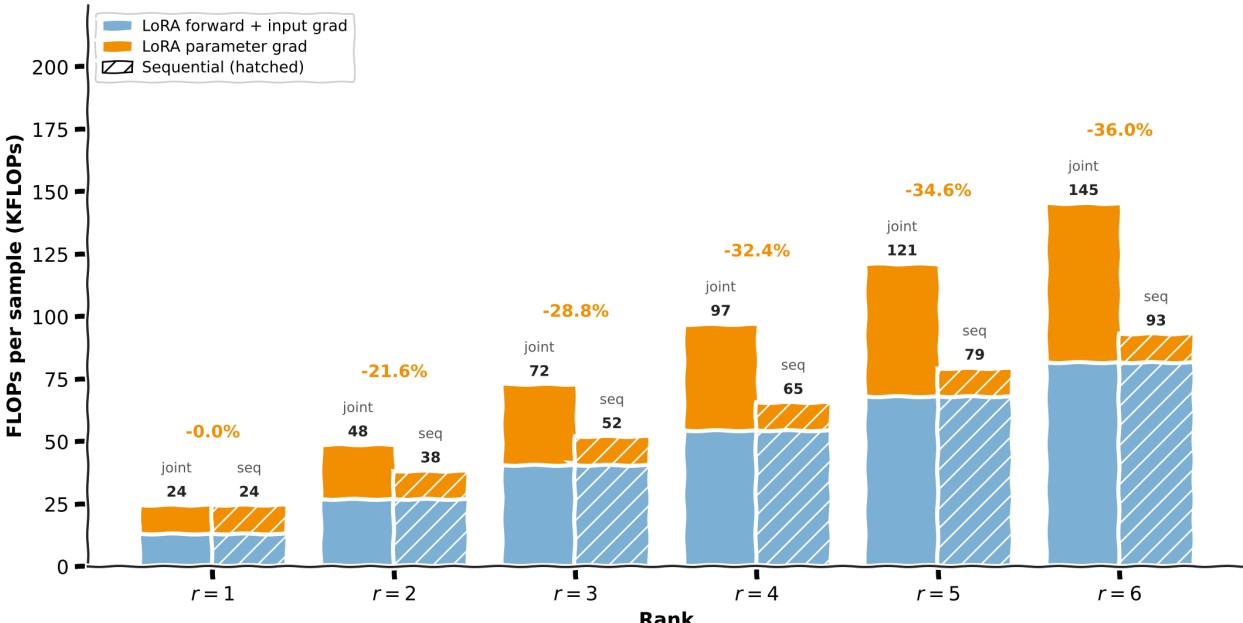

Figure 9: **Per-sample FLOPs of joint versus sequential LoRA as a function of total rank $r$.** For each target rank $r \in \{1, \ldots, 6\}$, we compare a jointly trained rank-$r$ LoRA update (*joint*) with a sequential rank-1 construction reaching the same total rank (*seq*). The stacked bars decompose the total per-sample cost (in KFLOPs): blue denotes the LoRA forward pass plus input-gradient computation, orange denotes the LoRA-parameter-gradient computation, and hatched bars indicate the sequential variant. The numbers above the bars report the total FLOPs per sample, while the orange percentages report the relative reduction of the sequential total with respect to the joint total. Under this FLOP accounting, the two methods coincide at $r = 1$, while for $r \geq 2$ sequential LoRA uses fewer FLOPs per sample than the joint update. Over the plotted range, this reduction increases with the target rank, from 21.6% at $r = 2$ to 36.0% at $r = 6$.

range as the joint LoRA reference. We interpret this only as exploratory qualitative evidence in the nonlinear PEFT setting, not as a theorem-level validation or a superiority claim over jointly trained LoRA.

# G   Experimental analysis on linear matrix regression.

We present experiments that validate our theory on error propagation in sequential rank-1 learning. Our experiments aim to demonstrate how the distribution of computational resources across rank-1 components affects the overall approximation quality, particularly focusing on how errors in early components propagate to later stages of the sequential learning process.

**Concrete instantiation of the $\alpha$-schedule family for Figure 1.** Section 5 defines a common one-parameter schedule family that maps $(r, T, m, \alpha)$ to a realized integer schedule $t(\alpha) = (t_1(\alpha), \ldots, t_r(\alpha))$. To make this concrete, we record the exact schedules used in Figure 1.

For Figure 1, we use
$$r = 20, \qquad T = 500, \qquad m = 2, \qquad \alpha \in \{-1, 0, 1\}.$$

Applying the allocation-and-rounding rule from Section 5 yields

$$t(-1) = (2, 4, 7, 9, 12, 14, 17, 19, 21, 24, 26, 29, 31, 33, 36, 38, 41, 43, 46, 48),$$

$$t(0) = (25, 25, 25, 25, 25, 25, 25, 25, 25, 25, 25, 25, 25, 25, 25, 25, 25, 25, 25, 25),$$

$$t(1) = (48, 46, 43, 41, 38, 36, 33, 31, 29, 26, 24, 21, 19, 17, 14, 12, 9, 7, 4, 2).$$

Thus, the less-first and more-first schedules are exact time reversals of one another, and the first-component allocations shown in Figure 1 are $t_1 = 2, 25, 48$.

**Experimental setting.** In this appendix, we consider the low-rank linear regression problem of finding $\mathbf{W}^\star \in \mathbb{R}^{m \times d}$ with rank $\leq r$ such that $\mathbf{Y} = \mathbf{W}^\star \mathbf{X} + \mathcal{E}$ where $\mathcal{E}$ is the noise term. This corresponds to finding a low-rank approximation of $\mathbf{W}^\star$. In these synthetic experiments, $\mathbf{X}$ is sampled from a standard Gaussian distribution only to generate controlled data, and this is not a requirement of the theory. We investigate the following settings:

1. *Singular Value Profiles:* We vary the singular value distribution of $\mathbf{W}^\star$ to analyze how the spectrum of ground truth influences error propagation.

2. *Noise Variations:* We introduce different types and levels of noise to assess the robustness of sequential rank-1 learning to perturbations.

3. *Iteration allocation strategies:* We evaluate three different iteration allocation strategies:
   (a) **Equal:** Same number of optimization iterations to each rank-1 component.
   (b) **More First:** More iterations allocated to the earlier components and fewer to later ones.
   (c) **Less First:** Fewer iterations allocated to the earlier components and more to later ones.

To ensure statistical robustness, all experiments are repeated across 5 independent trials. We report the mean performance across these trials, and visualize variability using shaded bands that represent the standard deviation.

We consider matrix dimensions $\mathbf{W}^\star \in \mathbb{R}^{500 \times 1000}$; we observed that experiments varying the dimensions of $\mathbf{W}^\star$ do not introduce any additional value to the main messages of this section. We generate $\mathbf{W}^\star$ with different singular value profiles, as in:

- **Uniform:** $\sigma_i = 10$ for all $i = 1, \ldots, r^\star$;

- **Exponential decay:** $\sigma_i = 100 \cdot \left(\frac{1}{100}\right)^{\frac{i-1}{r^\star - 1}}$ for $i = 1, \ldots, r^\star$;

- **Power-law decay:** $\sigma_i = \frac{100}{i^2}$ for $i = 1, \ldots, r^\star$;

where $r^\star$ is the true rank of $\mathbf{W}^\star$. Without loss of generality, we fix the rank $r^\star$ to 20 as we did not observe unexpected behaviors in the performance of the algorithms for different rank values.

We also consider several noise scenarios to evaluate robustness: *i*) Noiseless; *ii*) Gaussian where $\mathcal{E}$ has i.i.d. entries from $\mathcal{N}(0, \kappa)$ with $\kappa \in \{0.01, 0.05, 0.1\}$, and *iii*) Sparse where $\mathcal{E}$ is a sparse matrix (in our case 5% of entries are non-zeros) with non-zero entries from $\mathcal{N}(0, \kappa)$ with $\kappa \in \{1, 10\}$. In these synthetic experiments, $\mathbf{X}$ is sampled entrywise from a standard Gaussian distribution, $N(0, 1)$.

**Effect of singular value profile.** We study whether the singular value profile of $\mathbf{W}^\star$ has impact on error propagation through the singular gaps $\mathcal{T}_k^\star = \sigma_k^\star - \sigma_{k+1}^\star$ appearing in our error bound. Figure 10 shows the singular value decay patterns of both $\mathbf{W}^\star$ and the resulting $\mathbf{Y}$ under different spectral profiles. Figure 11 illustrates the training and reconstruction errors under these three profiles.

To ensure a fair comparison across different spectral profiles, we normalize the singular values of $\mathbf{W}^\star$ such that all generated matrices have the same Frobenius norm. This avoids artificially inflating or deflating error magnitudes due to differences in matrix scale rather than the structure of singular value decay.

*Observations:* The power-law decay profile shows the best performance, followed by the exponential decay, with the uniform profile performing worst. This matches the theoretical insight that large singular value gaps reduce the compounding of downstream error. Notably, power-law decay starts steep at the head—its first few singular values are significantly larger—creating large gaps for early components. In contrast, exponential decay is smoother initially and decays more evenly. Uniform singular values exhibit no decay, leading to minimal or zero gaps throughout.

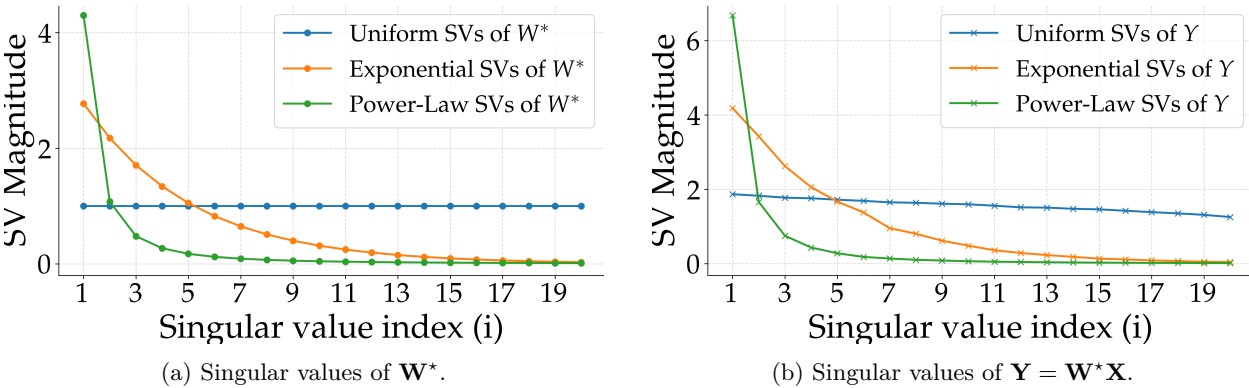

(a) Singular values of $\mathbf{W}^\star$.

(b) Singular values of $\mathbf{Y} = \mathbf{W}^\star \mathbf{X}$.

Figure 10: Comparison of singular value decay under different profiles. *Left:* $\mathbf{W}^\star$. *Right:* $\mathbf{Y} = \mathbf{W}^\star \mathbf{X}$.

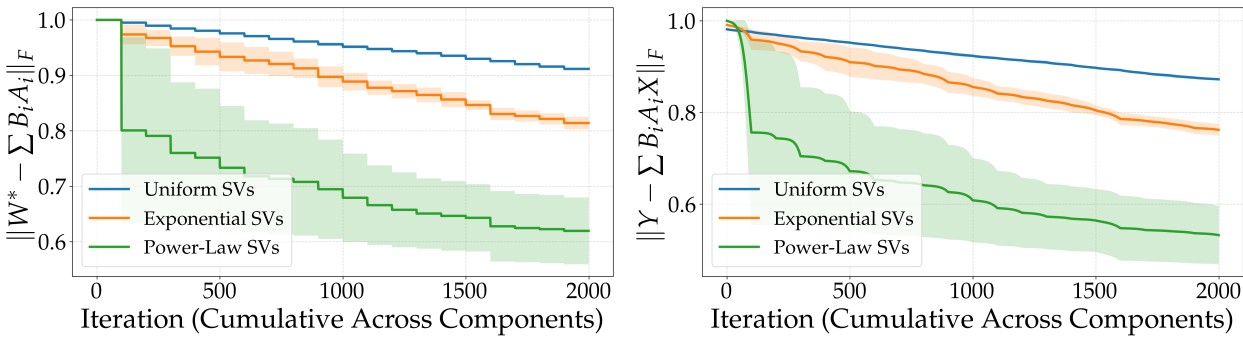

Figure 11: Effect of singular value profile on sequential learning performance. *Left*: $\mathbf{W}^\star$ reconstruction error. *Right*: Objective's training error.

**Impact of noise level.** Our theoretical analysis extends to noisy settings through Theorem 6, which characterizes how additive noise impacts parameter-recovery performance under fixed design. Figure 12 illustrates the effect of increasing noise levels $\kappa$ on both the training and reconstruction error under Gaussian and sparse noise levels.

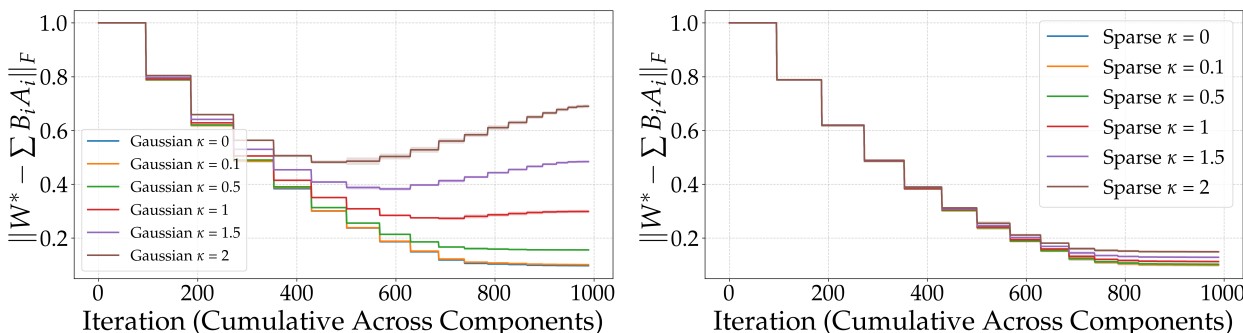

Figure 12: Impact of noise level. *Left:* Gaussian noise. *Right:* Sparse noise.

*Observations:* As expected, increasing the noise level $\kappa$ leads to higher reconstruction error in both Gaussian and sparse settings. Higher noise levels tend to corrupt the smaller singular values of $\mathbf{Y}$, making it difficult to distinguish low-rank structure from noise. This can lead to overfitting in later components of the sequential learner, as the algorithm begins to capture noise rather than signal.

**Effect of iteration allocation strategies in noisy settings.** To investigate mitigation strategies, we first evaluate how different iteration allocation strategies perform under noisy conditions. Figure 13 shows that the "more-first" strategy consistently outperforms others across varying noise levels by concentrating effort where it matters most—early in the sequence.

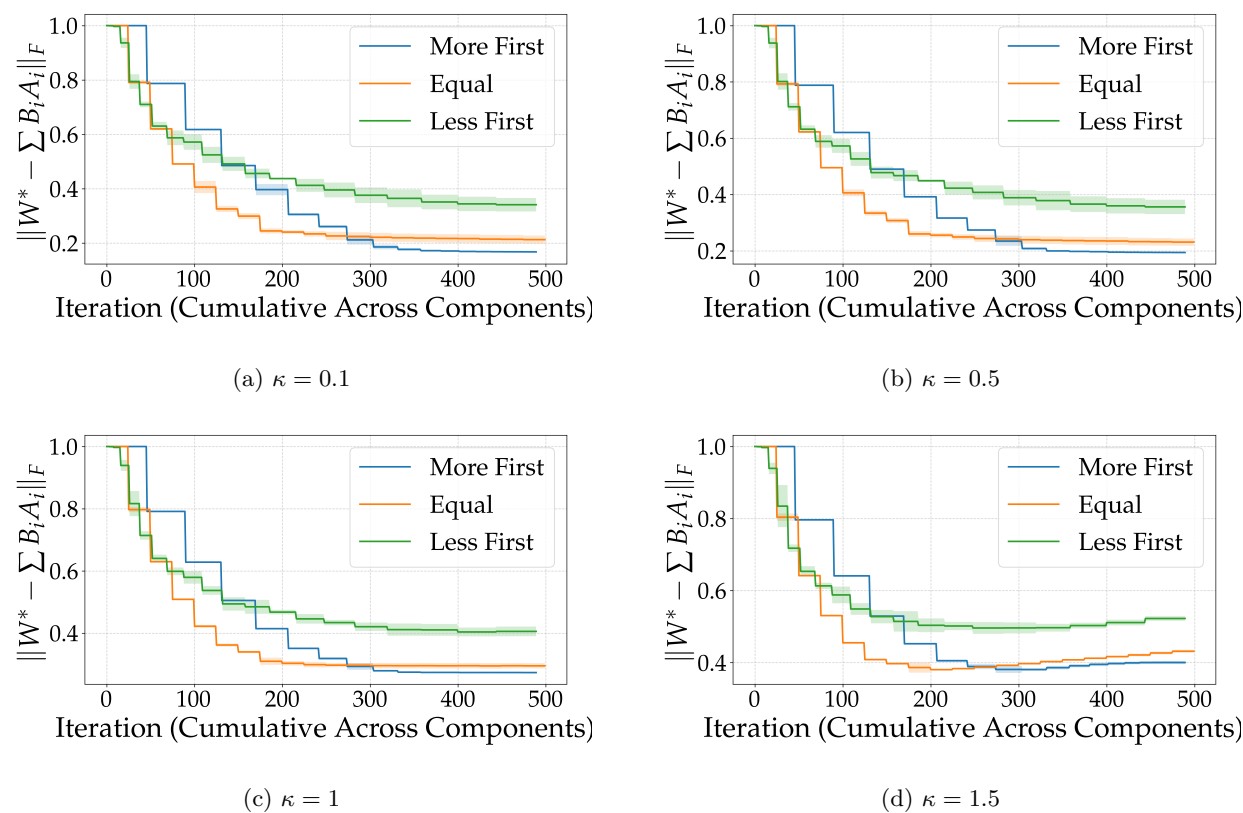

(a) $\kappa = 0.1$         (b) $\kappa = 0.5$

(c) $\kappa = 1$         (d) $\kappa = 1.5$

Figure 13: Comparison of iteration allocation strategies under different noise levels. The "more-first" strategy achieves better reconstruction error across all $\kappa$ values.

*Observation:* Even in noisy settings, the *more-first* strategy consistently outperforms *equal*, which in turn outperforms *less-first*, across all noise levels $\kappa$. This highlights the importance of prioritizing early iterations to mitigate error amplification under noise.

**Effect of singular value profiles in noisy settings.** We further examine how spectral decay influences robustness under noise. Using the *more-first* allocation strategy, Figure 14 shows that power-law decay consistently achieves lower reconstruction error compared to exponential and uniform profiles across all noise levels $\kappa$.

*Observation:* Spectral decay plays a critical role in robustness. Power-law decay, with its large leading singular values and wider gaps, allows early components to capture most of the signal, mitigating downstream error propagation. In contrast, uniform profiles lack this protective structure, making them more vulnerable to noise.

*Implications for Practical Use:* These results suggest that sequential learners can be more robust in the presence of noise by combining two strategies: allocating more iterations to early components and leveraging spectral decay. By front-loading optimization effort where it is impactful—at the beginning of the sequence—and favoring matrices with decaying singular values (especially power-law decay), models maintain lower reconstruction error despite increasing noise levels.

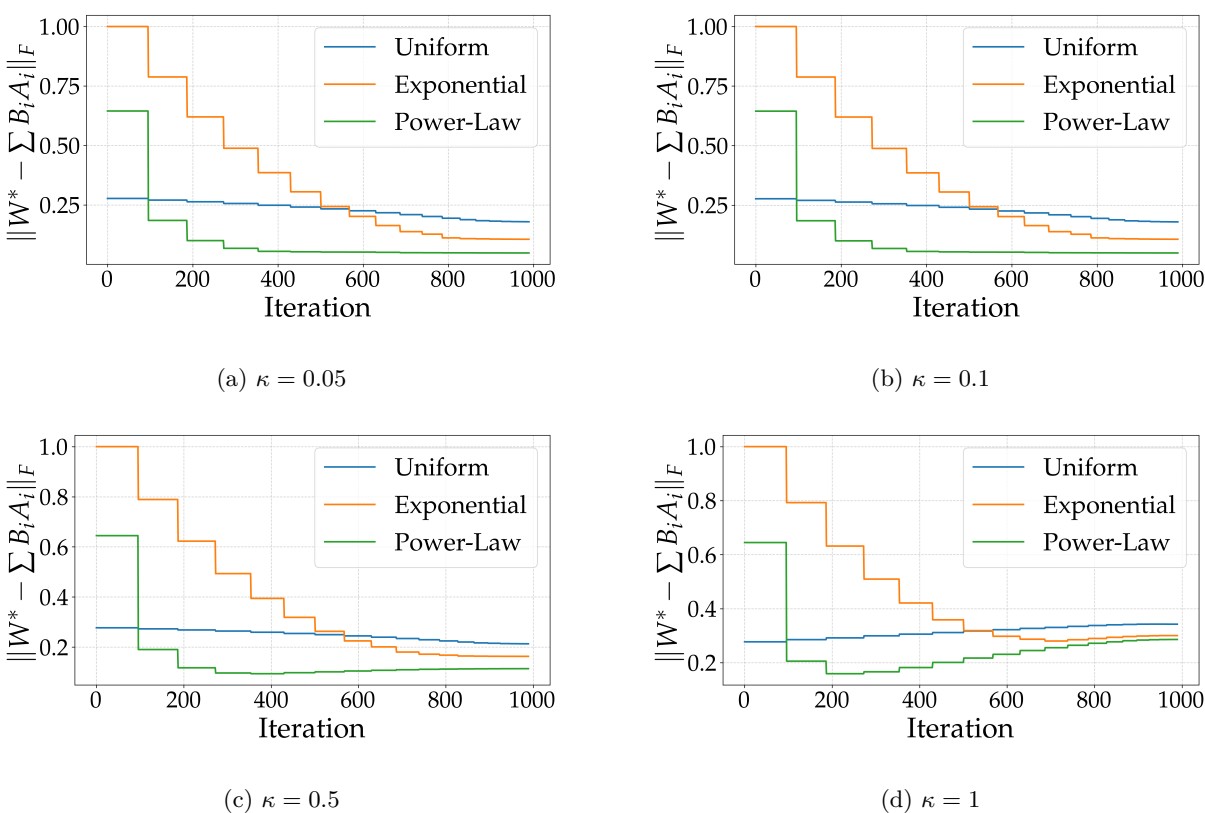

Figure 14: Effect of singular value profiles under noise. Power-law decay consistently achieves lower reconstruction error, followed by exponential and then uniform profiles, highlighting the benefit of spectral decay even in noisy settings.

**Computational efficiency analysis.** Beyond approximation quality, we also analyze the computational efficiency of different iteration allocation strategies. Specifically, we investigate how quickly each strategy reduces the reconstruction error to a desired threshold. Figure 15 illustrates, for a range of target error thresholds, the number of iterations required by each allocation strategy to reach that threshold.

*Observations:* The more-first iterations strategy consistently reaches target reconstruction thresholds faster than the equal or less-first strategies. This aligns with our intuition that prioritizing the early components—those with the greatest influence on downstream error propagation—leads to quicker convergence. In contrast, less-first allocation delays learning the principal directions, requiring more total iterations to reach the same accuracy. This suggests that our theoretical insights can lead to more computationally efficient algorithms for low-rank approximation.

**Dependence of the evaluated bound.** We empirically evaluate the two main quantities that drive Theorem 1: the numerical error terms $\boldsymbol{\Psi}_k$ and the measured spectral gaps $\mathcal{T}_k^\star$. Figure 16 isolates the role of $\boldsymbol{\Psi}_k$ by varying the iteration-allocation schedule while keeping the data matrix, target matrix, spectrum, and total iteration budget fixed. It plots the RHS directly as a function of the measured cumulative $\boldsymbol{\Psi}_k$ term. Figure 17 isolates the spectral dependence by plotting the measured gaps $\mathcal{T}_k^\star = \sigma_k^\star - \sigma_{k+1}^\star$ together with the evaluated theorem RHS across different spectral profiles.

*Observations:* The evaluated bound follows both measured dependencies predicted by the theorem. More-first schedules (larger $\alpha$) reduce the measured cumulative optimization error $\sum_k \|\boldsymbol{\Psi}_k\|_F$, and the evaluated RHS decreases accordingly. Across spectral profiles, smaller measured gaps lead to larger evaluated RHS values, reflecting the spectral product terms in the theorem. Together, these experiments show that the evaluated bound is controlled by the same empirical quantities that appear in the theory: the optimization

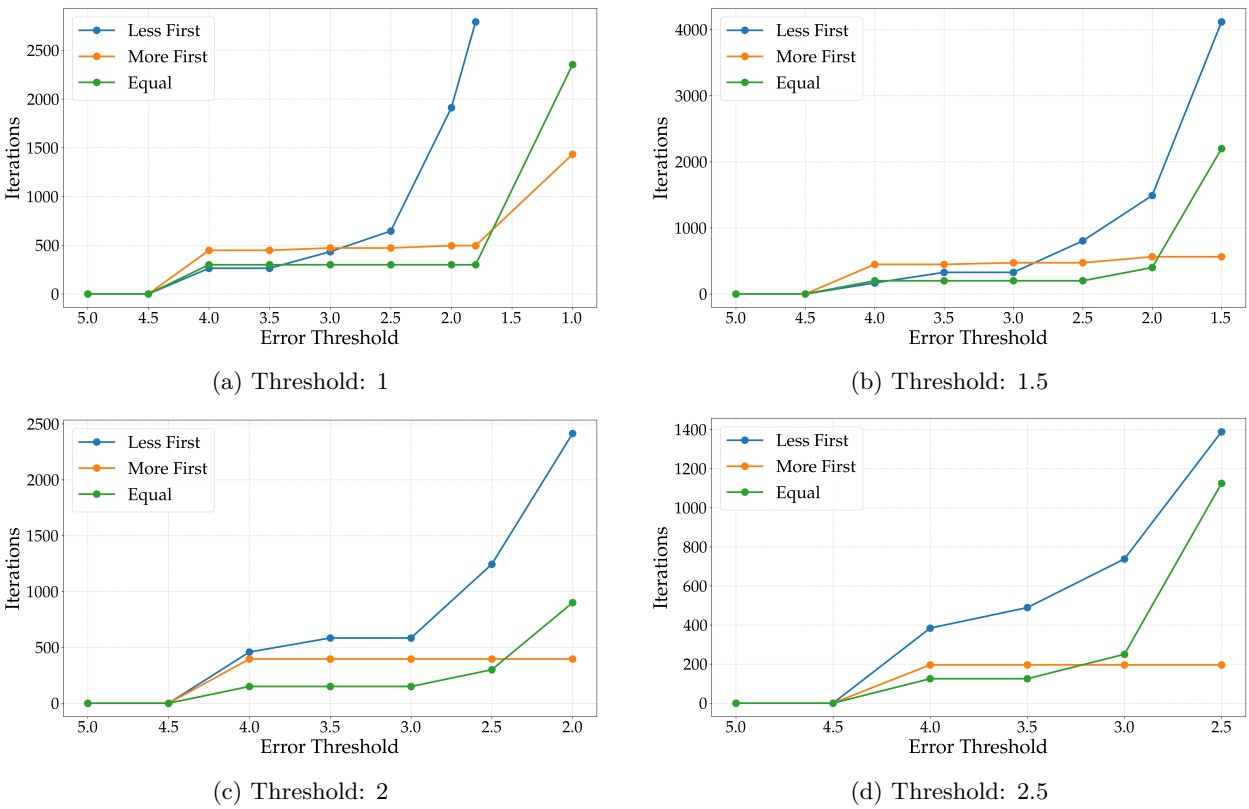

(a) Threshold: 1

(b) Threshold: 1.5

(c) Threshold: 2

(d) Threshold: 2.5

Figure 15: Number of iterations required to reach reconstruction error thresholds for different allocation strategies. Each subplot corresponds to a fixed error threshold. The "more-first" strategy consistently reaches the thresholds faster, especially for tighter reconstruction targets. In subplot (a), the "less-first" strategy fails to reach the threshold even after 10,000 iterations.

error $\mathbf{\Psi}_k$ sets the schedule-dependent part of the RHS, while the spectral gaps determine how strongly these errors contribute to the final bound.

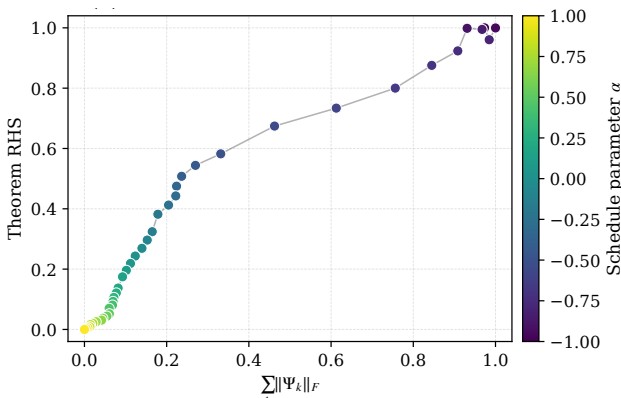

Figure 16: **Effect of the numerical error on the evaluated bound in Theorem 1.** We sweep the schedule parameter $\alpha$ while keeping $\mathbf{X}$, $\mathbf{W}^\star$, the singular spectrum, and the total iteration budget fixed. Larger $\alpha$ allocates more optimization steps to earlier rank-one components. We plot the evaluated theorem right-hand side against $\sum_k \|\mathbf{\Psi}_k\|_F$, with both axes normalized for visualization and color indicating $\alpha$. More-first schedules ($\alpha > 0$) reduce the measured numerical error, and that the evaluated bound decreases accordingly. Thus, with the spectrum and total compute fixed, the bound changes in the direction predicted by its explicit dependence on numerical error.

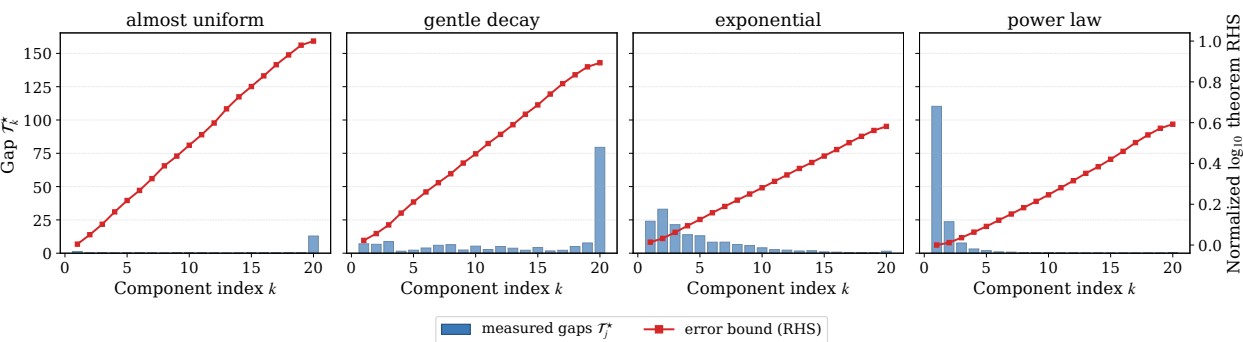

Figure 17: **Effect of spectral gaps on the evaluated bound in Theorem 1.** Each panel corresponds to one singular-value profile. Blue bars show the measured spectral gaps $\mathcal{T}_k^\star = \sigma_k^\star - \sigma_{k+1}^\star$ on the left axis, indexed by the component number $k$. The red curve shows the evaluated theorem right-hand side on the right axis, computed using the measured $\|\mathbf{\Psi}_{k'}\|_F$ values and the measured spectrum of $\mathbf{Y}$. The bound is plotted on a shared normalized $\log_{10}$ scale across all profiles and stages, so the different spectral profiles are directly comparable. Smaller measured gaps (e.g. uniform and gentle decay) increase the product terms appearing in the theorem and lead to a larger evaluated bound.

