# OpenReview forum: "One Rank at a Time: Cascading Error Dynamics in Sequential Learning"
_TMLR — Accepted by TMLR_

### Review · Reviewer_AXJc · 2026-02-28

**Summary Of Contributions:**

This paper investigates a fundamental research question, which is ‘how errors propagate during rank-1 sequential learning’. The authors first clearly define the error formulation based on exact and inexact learned parameters in the linear setting. They derive upper bounds on the output error that reveal key factors influencing error accumulation, and furthermore, they study the generalization of sequential rank-1 update under both noiseless and noisy settings. These theoretical results give a clear view of how accumulated errors affect model outputs in sequential low-rank learning. The authors also include empirical studies that qualitatively support the theoretical findings.

Strength: The work is clear and insightful for understanding error propagation in sequential learning.

Weakness: The experimental section lacks clarity in presentation and explanation, making it difficult to fully interpret the empirical results, as mentioned in the ‘requested changes’.

**Audience:**

Yes

**Audience Explanation:**

This work gives a good understanding of how error propagation occurs in sequential learning, which can help study LoRA fine-tuning to explore more efficient and accurate methods.

**Broader Impact Concerns:**

None.

**Claims And Evidence:**

Yes

**Claims Explanation:**

This paper works on theoretically explaining how approximation errors propagate during sequential training. It derives the error upper bound and generalization of sequential rank-1 update, which gives a clear view of what happens in sequential learning.

**Requested Changes:**

1. In the experiment section, from Figure 3 to Figure 5, the caption of each figure currently only explains the meaning of symbols without a brief overall summary to explain what performance is shown through the figure. Can authors add short summaries for each figure?

2. In Figure 4, the meaning of the different colors used for the horizontal bars is unclear. For example, four bars show the same accuracy (0.972) but are colored differently. Please clarify what each color represents.

3. For the mathematical formulation of LoRA in Eq(13) and Eq(14), this part appears disconnected from the surrounding discussion of experimental models and dataset performance. I think moving this non-linear formulation to the end of the linear formulation would be better, for example, moving to the front of ‘Definition 1’.

4. In section 5.2, ‘problem setting’ should be ‘experimental setting’.

---

> ### Author Response · Authors · 2026-04-29
>
> We thank the reviewer for their careful reading and constructive feedback. We have uploaded a revised manuscript, with revisions highlighted in blue. We also refer the reviewer to the General Response for overall clarifications made in the revision. We agreed that the submitted experimental section could be clearer. In the revision, we clarified it by redesigning several figures and rewriting the figure captions for clarity. In the responses below, we address each requested change and point to the exact locations in the revised paper.
>
> ## Responses
>
> **Response to Requested Change #1.**
> We agreed that the original experiment captions were too terse. In the revised manuscript, we rewrote the figure captions so that each caption stated the main empirical takeaway in addition to explaining the symbols. Because the empirical section was reorganized in the revision, the corresponding material now appears across Figs. 3--7 in Secs. 5.2--5.3. In particular, the revised captions now summarize the dataset-level adaptation results, the CIFAR100 accuracy--compute tradeoffs, and the SST-2 results, so that each figure is more interpretable without relying on the surrounding prose. We also made minor supporting edits for consistency with this revision.
>
> **Response to Requested Change #2.**
> We agreed that the submitted horizontal-bar visualization was visually ambiguous. We apologize for the confusion. In the original plot, the colors were only used to visually separate adjacent bars and did not encode an additional variable. In the revised manuscript, we removed that ambiguity by redesigning the schedule visualizations as accuracy--compute plots with explicit legends and captions (Sec. 5.2, Fig. 5; Sec. 5.3, Fig. 7), so that the schedule family and the corresponding \(\alpha\) values are directly identified in the figure itself. We also made minor supporting edits for consistency with this revision.
>
> **Response to Requested Change #3.**
> Thank you for this helpful suggestion. Our original intent was to keep a theory-first structure: Sections 3 and 4 developed the linear setting, while Section 5 introduced LoRA as a nonlinear experimental extension. However, we agreed that the original placement of Eqs. (13) and (14) made the transition into the LoRA experiments less smooth and made the connection to the sequential framework less explicit than it should have been. In the revised manuscript, we moved these equations to the beginning of Section 5.2, immediately after the introductory paragraph and before the experimental-setting details, and we added a brief transition sentence explaining that they are the nonlinear analogue of the sequential rank-1 updates studied in the linear setting. We appreciate the idea of introducing them earlier, but we preferred not to place them before Definition 1. Definition 1 and the surrounding discussion belong to the linear low-rank regression setup used in the theory; moving the nonlinear LoRA objective there would have blurred the distinction between the linear theoretical development and the later nonlinear experimental extension. We believe this revised placement addressed the flow issue while keeping the theory section clean and self-contained. We also made a minor supporting edit in the opening of Section 5 for consistency.
>
> **Response to Requested Change #4.**
> We thank the reviewer for catching this wording issue. In the revised manuscript, we renamed the subsection heading `Problem setting' to 'Experimental setting' in Section 5.2 and made the same change in Section 5.3 for consistency.
>
> We again thank the reviewer for the feedback. We are happy to discuss any further questions or concerns.

---

### Review · Reviewer_PDsx · 2026-03-02

**Summary Of Contributions:**

Summary:

This paper studies error propagation in sequential low-rank learning via repeated rank-1 deflation. The authors provide theoretical bounds on the recovery error under inexact rank-1 updates, characterizing how numerical errors accumulate across iterations. The analysis is based on perturbation arguments and spectral gap conditions. While the topic is interesting and potentially relevant to sequential low-rank methods (including LoRA-style approaches), I have several concerns regarding the assumptions and interpretability of the main theoretical results.

My overall recommendation is weak reject, mainly due to issues in the clarity and strength of the theoretical guarantees.

**Audience:**

Yes

**Audience Explanation:**

This topic is interesting and potentially relevant to sequential low-rank methods (including LoRA-style approaches)

**Claims And Evidence:**

No

**Claims Explanation:**

I have several concerns regarding the assumptions and interpretability of the main theoretical results. Please see requested changes below.

**Requested Changes:**

Theorem 1:

Spectral gap assumption.
The theorem implicitly requires the minimal spectral gap to be nonzero. This should be clearly stated as an explicit assumption, as a vanishing spectral gap is not uncommon in practice. The current formulation may not apply in such regimes.

Assumption in Lemma 3 not verified.
In Lemma 3, it is assumed that the approximation error of Y_k is smaller than the spectral gap at sigma_k^*. This assumption is not verified through the proof. The authors should either consider a specific rank-1 algorithm and use its convergence rate to validate such an assumption, or explicitly state this assumption in Theorem 1. As written, Theorem 1 relies on Lemma 3 but does not clearly propagate its assumptions.

Exponential terms in the RHS of (8).
The RHS of (8) contains exponentially growing terms due to the product factors. Could the authors comment on how fast the numerical error Phi_k has to decay in order to guarantee a nontrivial upper bound, i.e., that the RHS of (8) is smaller than O(1)? This is a crucial question because it identifies the benign regime where the result of Theorem 1 makes sense. Without such clarification, the bound may be vacuous in realistic settings.

These concerns seem to be valid also for Theorem 5, Theorem 6, and Theorem 7 (in the Appendix), which rely on similar structural arguments.

Theorem 6:

Although the authors impose a smallness assumption on the noise epsilon, it does not seem sufficient to guarantee that the second term (the orange term) in the RHS of (12) is smaller than O(1). Could the authors clarify what additional conditions on the noise (or other parameters) are needed so that the RHS in (12) is meaningful, i.e., bounded by O(1)? As written, the bound may still grow with rank or other quantities.

Assumption on Data Distribution:

The paper assumes that the data matrix X is sampled entrywise from a normal distribution with zero mean and unit variance, followed by a row-wise normalization process. However, this assumption does not appear to be used anywhere in the theoretical analysis. If the results are fully deterministic and depend only on spectral quantities (e.g., sigma_max(X), sigma_min(X)), then the Gaussian sampling assumption seems unnecessary and potentially confusing.

Typos:

Above (16): "ttThe" -> "The".
Remark 3: Did you mean to require T_k^* > 0?

Notation Issues:

Section 3: nsv1_L(Y_k) and nsv1_R(Y_k) were never defined. Did you mean sv_L(Y_k) and sv_R(Y_k), or simply the top left/right singular vectors?

Theorem 1: The inner minimum in T_k^* appears redundant since singular values are assumed ordered. It seems that T_k^* = sigma_k^* - sigma_{k+1}^*. Similarly, the assumption on E(k) can also be simplified.

---

> ### Author Response · Authors · 2026-04-29
>
> We thank the reviewer for their careful reading and constructive feedback. We have uploaded a revised manuscript, with revisions highlighted in blue. We also refer the reviewer to the General Response for overall clarifications made in the revision. We appreciated the reviewer’s broader concern and clarified the theorem assumptions and interpretation in the revision. In the responses below, we address each requested change and point to the exact locations in the revised paper.
>
> ## Responses
>
> **Response to Requested Change #1.**
>
> We agreed that the nonzero spectral-gap condition should have been stated explicitly in the theorem statement, rather than only appearing implicitly through the definition of $T_k^\\star$.
> In the revision, we added the assumption $T_k^\\star>0$ for all $k\\in[r]$ as an explicit hypothesis of Theorem 1. We also revised Remark 3 to make clear that the case $T_k^\\star = 0$ corresponds to repeated singular values, for which the individual rank-1 component is not uniquely identifiable. In that regime, our current component-wise analysis does not apply as stated; a meaningful treatment would require a subspace-level perturbation analysis over the repeated-singular-value block. We also made the same theorem-level assumption explicit in Theorems 5 and 6 in the main text, and in Theorem 8 in Appendix B.4 for consistency.
>
> **Response to Requested Change #2.**
>
> Thank you for pointing this out. The premise of Lemma 3 was already implied by the hypothesis of Theorem 1, but we agreed that this implication was not stated clearly enough in the previous version. In the revision, we made this implication explicit both in the discussion after Theorem 1 and in Appendix B.4, in the proof of the more general Theorem 8 (which implied Theorem 1), especially equations (32)--(33) in Appendix B.4. In Appendix B.4, the proof of the more general Theorem 8 (which implied Theorem 1) showed in equation (32) that
> $$\\|Y_k-Y_k^\\star\\|_F\\le\\widehat{E}(k).$$
> At the end of the proof of Theorem 8 in Appendix B.4, after establishing $T_k\\ge\\tfrac{1}{2}T_k^\\star$, the argument passed from the sharper quantity $\\widehat{E}(k)$ to the theorem-level bound $E(k)$ and obtained $E(k)\\ge\\widehat{E}(k)$. Therefore, the hypothesis of Theorem 1,
> $$E(k)<\\tfrac{1}{2}T_k^\\star,$$
> implied
> $$\\|Y_k-Y_k^\\star\\|_2\\le\\|Y_k-Y_k^\\star\\|_F\\le\\widehat{E}(k)\\le E(k)<T_k^\\star=\\min\\_{j>k}|\\sigma_k^\\star-\\sigma_j^\\star|,$$
> which was exactly the condition required in Lemma 3; this was the content of equation (33). Therefore, the condition in Lemma 3 was not an additional assumption beyond what Theorem 1 already stated. Since Theorem 1 was deliberately stated in terms of the numerical errors $\\|\\Psi_k\\|_F$, rather than a particular rank-1 subroutine, we did not need to introduce an additional algorithm-specific convergence assumption here. We acknowledged, however, that this logical chain was not spelled out clearly enough in the previous version. In the revision, we made this implication explicit in the theorem discussion and in the appendix proof. We also made minor supporting edits for consistency.
>
> **Response to Requested Change #3.**
>
> Thank you for raising this important point. We agreed that the original Remark 2 was only qualitative, and in the revised manuscript we made the benign regime explicit in Remark 2. In particular, for
> $$1\\le\\ell\\le k\\le r,\\qquad P\_{\\ell,k}:=\\prod\_{j=\\ell+1}^{k}\\left(2+\\frac{6\\sigma_j^\\star}{T_j^\\star}\\right),$$
> we interpreted $P\_{\\ell,k}$ as the downstream amplification of the stage-$\\ell$ numerical error. Hence a sufficient condition for the numerical-error contribution in (8) to be $O(1)$ is
>
> $$\\sum_{\\ell=1}^{r} \\|\\Psi_\\ell\\|_F \\sum\_{k=\\ell}^{r}P\_{\\ell,k}=O(\\frac{1}{\\sigma\_{\\max}(X)}).$$
>
> A simple per-stage sufficient condition is
>
> $$\\|\\Psi_\\ell\\|_F\\le\\frac{c}{r\\,\\sigma\_{\\max}(X)}\\left(\\sum\_{k=\\ell}^{r}P\_{\\ell,k}\\right)^{-1},\\qquad \\ell\\in[r],$$
>
> under which the numerical-error contribution in (8) remains $O(1)$. This makes clear that each stage error must decay like the inverse of its total downstream amplification, so earlier components must be solved more accurately than later ones, especially when the spectral gaps are small. We also noted there that the same product-based amplification structure appears in Theorems 5, 6, and 8, with $\\kappa(X)$ replacing $\\sigma\_{\\max}(X)$ in Theorems 5 and 6. While Remark 4 stated that the overall upper bound was not claimed to be tight, the revised Remark 2 still made the theorem more interpretable by extracting an explicit sufficient benign regime from the current bound and by clarifying why earlier components had to be solved more accurately than later ones for the numerical-error contribution in (8) to remain $O(1)$.

---

> > ### Author Response · Authors · 2026-04-29
> > **Response to Reviewer PDsx (continued)**
> >
> > **Response to Requested Change #4.**
> >
> > Thank you. We agree. The noise-scale condition stated in Theorem 6 was used to preserve the ordering of the relevant singular components after perturbation, but it did not by itself force the second term in equation (12) to be $O(1)$. We added Remark 7 after Theorem 6 to clarify this point, and we added a forward pointer in the discussion of Theorem 6. Indeed, the second term in equation (12) scales as
> > $$O\\left(\\frac{\\varepsilon\\sqrt{n\\log(1/\\gamma)}}{\\sigma\_{\\min}(X)}\\left(r+\\sqrt{\\frac{\\min\\{r^\\star,r\\}}{T\_{\\min}^\\star}}\\right)\\right).$$
> > Therefore, a sufficient additional condition for this noise contribution to be $O(1)$ is
> > $$\\varepsilon=O\\left(\\frac{\\sigma\_{\\min}(X)}{\\sqrt{n\\log(1/\\gamma)}\\left(r+\\sqrt{\\min\\{r^\\star,r\\}/T\_{\\min}^\\star}\\right)}\\right).$$
> > This was stronger than the ordering condition stated in Theorem 6. Theorem 6 was stated under the weaker ordering condition so as to cover the broader regime in which the noise contribution in equation (12) need not be $O(1)$ and should instead be interpreted through its explicit scaling. This made explicit that the second term in equation (12) could still grow with $r$ and with smaller spectral gaps unless $\\varepsilon$ was correspondingly smaller. Accordingly, Remark 7 now separates these two roles: the current noise-scale condition preserves component ordering, while the stronger condition above is sufficient to make the second term in equation (12) constant-order. Without this stronger condition, the second term should be interpreted as an explicit scaling law rather than as an $O(1)$ guarantee. Of course, for the full right-hand side of equation (12) to be $O(1)$, the first term must also remain controlled. We also made minor supporting edits for consistency.
> >
> > **Response to Requested Change #5.**
> >
> > We agree that the Gaussian sampling statement for $X$ was unnecessary for the theoretical analysis, and we removed it from the theoretical setup. In the revised manuscript, we clarified this distinction explicitly in two places. First, in the Problem setup in Section 2, we now state the theory for an arbitrary fixed design matrix $X$ and note that the bounds depend on $X$ only through spectral quantities such as $\\sigma\_{\\max}(X)$ and, where needed, $\\sigma\_{\\min}(X)>0$. Second, in Appendix G, we moved the Gaussian design assumption to the synthetic experimental setup and stated explicitly that it is used only to generate controlled data, not as a requirement of the theory.
> >
> > **Response to Requested Change #6.**
> >
> > Thank you for the careful proofreading. We corrected these typos and notation issues in the revision. Specifically, we corrected the typo above equation (16), revised Remark 3 to require $T_k^\\star > 0$, and replaced the undefined notation for the top singular vectors by the already defined $svL(Y_k)$ and $svR(Y_k)$ in Section 3. We also simplified the theorem-level gap notation in Theorems 1, 5, 6, and 8 by writing
> > $$T_k^\\star=\\sigma_k^\\star-\\sigma\_{k+1}^\\star,$$
> > under the convention $\\sigma\_{p+1}^\\star:=0$ (or $\\sigma\_{r^\\star+1}^\\star:=0$ when the rank-$r^\\star$ SVD is used), and simplified the corresponding smallness conditions on $E(k)$ accordingly. We kept the perturbation quantity $T_k$ in Lemma 3 unchanged, since there it compared the population singular value $\\sigma_k^\\star$ with the singular values of the perturbed matrix $Y_k$, rather than only with the next population singular value. We also made minor supporting edits for consistency.
> >
> > We again thank the reviewer for the feedback. We are happy to discuss any further questions or concerns.

---

### Review · Reviewer_UYWh · 2026-04-14

**Summary Of Contributions:**

This work analyzes how gradient-based rank-1 updates in sequential learning accumulate error in regression tasks. In particular, it characterizes the accumulated error through the gap between the rank-1 matrix product $ab^{\top}$ obtained by the gradient-based update and the SVD-based optimal solution, respectively. Since this update can be viewed as the simplest form of LoRA, a widely used method for fine-tuning large-scale models, this analysis is meaningful.

**Additional Comments:**

N/A

**Audience:**

Yes

**Audience Explanation:**

I feel that this work presents interesting results on the error analysis of rank-1 updates for sequential learning. Although the work itself does not directly focus on LoRA error analysis, it could still attract researchers who are interested in analyzing errors in sequential learning with LoRA-based methods. Therefore, I believe this work would be of interest to part of the TMLR audience.

**Claims And Evidence:**

Yes

**Claims Explanation:**

Most of the theoretical claims are clear, and the empirical results support the claims of the paper.

But, I have some questions about some experiments:

* For the LoRA experiments in Sections 5.2, the models are trained with a classification loss. Does your analysis still imply the same conclusions, even though it is derived for a regression task? I find it difficult to understand the intention and purpose of these experiments because of this mismatch.

* In addition, in Section 5.3, the BERT variant used for the language tasks includes non-linear terms beyond the model assumptions used in the analysis. This mismatch raises questions about the implications of the empirical results, as they seem less relevant to the theoretical analysis.

**Requested Changes:**

* In Theorem 2, it is necessary to make the assumption on the noise variance $\epsilon$ explicit. Although the authors mention in the last part of Section 4 that $\epsilon$ should be set so that the ordering of the rank-1 components changes after adding noise, this should be clearly stated, for example, as an assumption on $\epsilon$ or as a remark


* In Figure 1, it is not clear at a glance by when more iterations are considered and by when fewer iterations are considered. Please clarify this in the caption of Figure 1, for example, by explicitly explaining the different iterations as shown on the x-axis or by using empty markers in the plot.

* In Figure 1, it would be much better to add the progress of $\sum_{k'} \lvert \Psi_{k'} \rvert_F$ from the error bound in Theorems 1 and 5, because this would more directly illustrate the relationship among the generalization error, the parameter error, and the accumulated error $\sum_{k'} \lvert \Psi_{k'} \rvert_F$ caused by the gradient-based optimization.

---

> ### Author Response · Authors · 2026-04-29
>
> We thank the reviewer for their careful reading and constructive feedback. We have uploaded a revised manuscript, with revisions highlighted in blue. We also refer the reviewer to the General Response for overall clarifications made in the revision. In the responses below, we address each requested change and point to the exact location in the revised paper.
>
> ## Responses
>
> **Question 1 (Section 5.2).**
>
> Thank you for this helpful question. We agree that the original exposition did not separate these roles clearly enough. The analysis in Sections 3--4 was derived for sequential rank-1 linear regression with squared loss, so it did not directly imply the same quantitative guarantees for the classification-loss LoRA experiments in Section 5.2.
>
> Our intent in Section 5.2 was more modest: the direct validation of the theory was the synthetic linear setting in Section 5.1 (and Appendix G), while Section 5.2 asked whether the same *qualitative* scheduling phenomenon suggested by the linear theory, especially the importance of learning early components accurately, also appeared in a simple PEFT setting. In the revised manuscript, we made this distinction explicit in the opening of Section 5 and in the first paragraph of Section 5.2, where we stated that these experiments use classification loss and nonlinear networks and therefore are not direct instantiations of the regression theory. We also framed Section 5.2 as exploratory qualitative evidence rather than as a direct validation of the regression theorems, and stated the same limitation explicitly in Section 6.
>
> **Question 2 (Section 5.3).**
>
> Thank you for pointing this out. We agreed that Section 5.3 needed to be framed more carefully. Our analysis in Sections 3--4 was derived for linear low-rank regression and did not cover the nonlinear DistilBERT architecture or the classification loss used in the SST-2 LoRA experiment. Our theory therefore did not cover this setting. Our intent in Section 5.3 was therefore more modest. The direct validation of the theory was provided by the synthetic linear setting in Section 5.1 and Appendix G, while the language-task experiment in Section 5.3 was included only to examine whether the same *qualitative* scheduling effect suggested by the linear analysis also appeared in a more realistic PEFT setting. We revised Section 5.3 to make this distinction explicit and to present the SST-2 experiment as exploratory qualitative evidence rather than as a direct validation of the linear theory. We also stated this limitation explicitly in Section 6 and adjusted the conclusion accordingly in Section 7. We also made minor supporting edits for consistency.
>
> **Response to Requested Change #1.**
>
> Thank you. We agree that the role of the small-noise assumption on the noise scale $\\epsilon$ should be stated more explicitly. In our noisy-label theorem (Theorem 6), a small-noise assumption was already present in the original submission, but its purpose was not explained clearly enough. In the revised manuscript, we made this explicit in Theorem 6 and Remark 7: Theorem 6 now states the ordering condition on $\\epsilon$ directly, and the first paragraph of Remark 7 explains that this condition keeps the additive perturbation small relative to the minimum singular-value gap, so that the relevant rank-1 components are not reordered after noise is added. In addition, Appendix C.2 (proof of Lemma 4, which is invoked in the proof of Theorem 6) now shows how the same condition yields the required singular-value separation. Remark 7 further distinguishes this theorem-level ordering condition from the stronger condition needed if one wants the additive-noise term in (12) itself to remain $O(1)$.
>
> **Response to Requested Change #2.**
>
> Thank you for this helpful suggestion. We revised Figure 1 to make the iteration allocation across components easier to read directly from the plot. In particular, we revised Figure 1 so that the iteration allocation can be read directly from the plot: the x-axis now states that iterations are cumulative across successive rank-1 components, so each interval corresponds to one component, with longer intervals indicating more iterations and shorter intervals indicating fewer. We also added open-circle markers at component-switch points, vertical dashed lines with annotated $t_1$ values to highlight the budget allocated to the first component for demonstration. We revised Section 5 to specify the common schedule family, and Appendix G now records the exact realized per-component schedules used in Figure 1. We also made minor supporting edits for consistency.

---

> > ### Author Response · Authors · 2026-04-29
> > **Response to Reviewer UYWh (continued)**
> >
> > **Response to Requested Change #3.**
> >
> > We agree that making this quantity visible strengthens the connection to Theorems 1 and 5. As suggested by the reviewer, we revised Figure 1 to add a third panel that tracked a cumulative numerical-error proxy built from the $\\|\\Psi_k\\|_F$ quantities appearing in Theorems 1 and 5. Specifically, in the synthetic linear setting, the right panel plotted
> > $$S(t)=\\sum\_{j<k}\\|\\Psi_j\\|_F+\\|\\Psi_k^{(t)}\\|_F,$$
> > where $k$ denotes the component currently being optimized at cumulative iteration $t$. Together with the reconstruction and training-error panels, this made the relation between optimization-induced numerical error, the reconstruction error of $W^\\star$, and the training error more direct under the same fixed-budget schedules (Figure 1).
> >
> > We again thank the reviewer for the feedback. We are happy to discuss any further questions or concerns.

---

### Review · Reviewer_tfKm · 2026-04-21

**Summary Of Contributions:**

## Summary of contributions

This paper studies sequential low-rank learning through a linear regression model in which a rank-$r$ solution is built one rank-1 component at a time via deflation. The paper claims three main contributions:
1. an error-propagation bound showing how numerical errors in each rank-1 subproblem compound through the sequence (Theorem 1),
2. noiseless and noisy-label recovery bounds for the recovered low-rank model (Theorems 5–6), and
3. synthetic plus toy LoRA experiments suggesting that earlier components deserve more optimization budget.

The practical message is intuitive: errors made early in the sequential deflation chain contaminate later residuals, so early components should matter more.

## Key strengths

- The paper asks a relevant question. Sequential low-rank updates are natural in unknown-rank, streaming, and modular-adaptation settings.
- The framing is refreshingly honest that the goal is explanatory rather than to claim superiority over joint training.
- The synthetic appendix contains useful ablations over singular-value profile, noise level, and schedule choice. These ablations are directionally aligned with the intended intuition.

## Key weaknesses

- The core theory is not trustworthy in its current form. There are multiple concrete mathematical and statement-level issues in the proof chain.
- The "generalization" framing is overstated: the theorems are fixed-design recovery / estimation bounds, not standard out-of-sample generalization guarantees.
- The paper overclaims the practical transfer of the linear theory to non-linear LoRA. Its own non-linear experiments do not support a simple monotone front-loading rule.
- The empirical comparison to joint LoRA is not compute-clean enough to support strong practical conclusions.
- The practical LoRA evaluation is too narrow to justify a broad PEFT or "sequential learning" takeaway.
- The manuscript needs significant tightening in notation, theorem statements, and proofreading.

**Additional Comments:**

Please refer to requested changes.

**Audience:**

No

**Audience Explanation:**

The paper studies an interesting question, and some members of the TMLR audience (especially those working on low-rank approximation, sequential optimization, or the theory of deflation-style methods) may find the mathematical framing and the singular-gap intuition worthwhile. I also appreciate that the paper is honest about its goal being explanatory rather than claiming superiority over joint training, and the synthetic experiments do provide some qualitative support for the linear theory.However, my enthusiasm is limited because the paper does not convincingly establish the applicability of its theory to real-world non-linear settings. The main practical motivation is tied to modern methods such as LoRA, yet the empirical results in these non-linear settings do not consistently support the paper’s central front-loading message. As a result, the relevance of the theory to everyday modern ML practice is not established. So while the math is interesting and the core intuition is appealing, the paper currently feels more like a theoretically suggestive analysis of a stylized linear setting than a result with demonstrated practical relevance to the broader TMLR audience.

**Claims And Evidence:**

No

**Claims Explanation:**

The biggest issue is the soundness of the proof chain.

First, Appendix A / Equation 16 mishandles the Frobenius norm of the truncated SVD tail. The manuscript treats the tail as a simple sum of singular values, whereas the Frobenius truncation residual should involve the square root of the sum of squared singular values. Even if the authors intended an upper bound, the proof as written uses an equality where only a loose inequality could hold.

Second, Lemma 2 introduces $\delta_k$ without defining it, even though the surrounding derivation clearly uses $\Psi_k$. This is not just cosmetic because the recurrence is central to the whole argument.

Third, Lemma 3's use of Wedin-style perturbation is suspicious relative to the paper's own theorem statement in Appendix D. The manuscript's quoted Wedin theorem is formulated using terms like $U_1^T \Delta$ and $\Delta V_1$, but the actual proof uses expressions of the form $v_k^{\ast T}(Y_k^\ast - Y_k)$  and  $(Y_k^\ast- Y_k)u_k^\ast$, which do not line up cleanly and appear dimensionally inconsistent in the rectangular setting.

Fourth, the theorem statements are internally inconsistent. Theorems 1, 5, 6, and 7 do not consistently use the same gap quantities, with $T_k^\ast$ and $T_j^\ast$ appearing interchangeably across the main text and appendix. Theorem 6 also appears to have a wrong truncation term, using a repeated $\sigma_r(W^\ast)$ summand where $\sigma_k(W^\ast)$ is the natural quantity.

These are not presentation-only issues. They affect the main technical claim.

I also do not think the paper should market Theorems 5–6 as "generalization" results without qualification. The results bound quantities such as $||W^\ast - \sum_k b_k a_k^T||_F$ under a fixed design matrix $X$; they do not establish out-of-sample risk on a test distribution.

The empirical section is also not convincing enough to rescue the paper. The synthetic experiments are directionally aligned with the theory's intuition, but they do not validate the actual scaling form of the theorem: they do not measure $\Psi_k$, the relevant spectral gaps, or the dependence predicted by the bound.

More importantly, the non-linear LoRA section weakens rather than strengthens the practical story. The paper's theory suggests that early errors should be the most dangerous, which invites a monotone front-loading interpretation. But the paper's own results do not cleanly support that in the non-linear setting. In Figure 4 (MNIST), the back-loaded schedule $1 \rightarrow 1 \rightarrow 10$ reaches 0.967, slightly outperforming the much heavier $10 \rightarrow 10 \rightarrow 3$ schedule at 0.966 despite using far fewer total epochs. In Figure 5 (SST-2), the paper itself notes that $1 \rightarrow 1 \rightarrow 5$ slightly outperforms $3 \rightarrow 3 \rightarrow 3$. So the non-linear evidence does not justify a clean "allocate more compute earlier" rule; at best, it supports a softer adaptive principle.

The comparison to joint LoRA is also not rigorous enough. Figure 2 does not clearly specify the exact training budget behind each sequential bar. Figure 3 uses total epochs and marker size rather than reporting FLOPs or wall-clock time. As a result, the reader cannot tell whether sequential $r = 1 + 1 + 1$ and joint $r = 3$ are actually being compared on equal footing. This is especially important because the paper uses these figures to argue practical relevance.

Finally, the empirical scope is too narrow. The vision experiments use a simple 3-layer MLP, explicitly described as intentionally simplistic, and the language side is limited to DistilBERT on SST-2. That is too small a surface to support broad claims about LoRA or sequential learning in modern models.

**Requested Changes:**

### Critical changes

1. **Repair the theorem/proof chain completely.** Fix the Frobenius-tail handling in Equation 16, define $\delta_k$ or replace it consistently with $\Psi_k$, repair the Lemma 3 perturbation argument, and reconcile the $T_k^\ast$ versus $T_j^\ast$ inconsistencies across Theorems 1/5/6/7 and the appendix. Also correct Theorem 6's truncation term if it is indeed a typo.

2. **Reframe "generalization" as recovery / estimation, or prove actual out-of-sample bounds.** The present terminology oversells what the theorems establish.

3. **Tone down the practical non-linear conclusion.** The LoRA section does not support a simple monotone front-loading message. The paper should either formalize a more nuanced adaptive scheduling principle or explicitly state that the linear theorem does not directly predict non-linear LoRA behavior.

4. **Provide compute-clean sequential-vs-joint comparisons.** Replot Figure 2 or add new experiments under a fixed total training budget, and report real compute metrics such as FLOPs or wall-clock time rather than only epochs.

5. **Clarify the exact schedule definitions.** The paper should explicitly state the iteration splits behind "more-first", "equal", and "less-first". Schedule allocation is one of the central practical claims, but the setup is described only qualitatively in Section 5.1.

6. **Strengthen the schedule ablations.** Section 5.1 is too coarse. Instead of only three broad categories, I would like to see a finer-grained sweep over schedules so the reader can judge where front-loading helps, where it saturates, and where it fails.

7. **Sharpen the scope claim.** Either present the work clearly as a linear-theory paper with illustrative non-linear experiments, or provide a real theoretical bridge to non-linear settings.

### Changes that would further strengthen the paper

1. **Tighten notation and proofreading throughout.** In particular, the exact / step-optimal / finite-iteration rank-1 objects remain confusing in dense derivations.
2. **Remove redundant statements** such as explicitly writing that $||\Psi_k||_F \ge 0$ in Equation 7.
3. **Provide full experimental details**: optimizer, learning rate, batch size, schedule values, stopping criteria, seeds, and code release status.
4. **Show uncertainty directly in the LoRA figures** rather than only mentioning a maximum standard deviation in the text.
5. **Add at least one broader sequential-learning testbed** if the authors want the framing to extend beyond low-rank linear regression and toy LoRA. In particular, I encourage the authors to test whether similar scheduling principles appear in gradient boosting methods such as XGBoost, which also follow a sequential learning paradigm. I do not view this as required for acceptance, but it would substantially strengthen the broader “sequential learning” framing and help demonstrate whether the paper’s intuition has utility beyond the specific low-rank regression / LoRA setting.

---

> ### Author Response · Authors · 2026-04-29
>
> We thank the reviewer for the careful and technically substantive feedback. We agree that the previous draft needed a cleaner theorem/proof presentation and a clearer separation between the linear theory and the exploratory nonlinear LoRA experiments. In the revision, we corrected the main theorem/proof-chain issues and compute-matched LoRA comparisons explicit, and tightened the notation and empirical presentation. We have uploaded a revised manuscript, with revisions highlighted in blue. We also refer the reviewer to the General Response for overall clarifications made in the revision. In the responses below, we address each requested change and point to the exact locations in the revised paper.
>
> ## Responses
>
> **Response to Critical Change 1.**
>
> We thank the reviewer for the detailed technical reading or our proofs. The issues raised here fell into two categories: (i) genuine formula/proof corrections, namely the Frobenius-tail expression and Lemma 3 perturbation argument, and (ii) notation/statement inconsistencies, namely the stray $\\delta_k$ symbols, the use of $T_k^\\star$ instead of the stage-wise $T_j^\\star$ inside the amplification products, and the truncation typo in Theorem 6. Following the reviewer's feedback, we corrected the Frobenius-tail handling in Appendix A and propagated that correction to Theorems 1, 5, 6, and Appendix Theorem 8. We standardized the numerical-error notation to $\\Psi_k$ throughout, and rewrote the Lemma 3 perturbation step using the rank-1 form of Theorem 10. Finally, we synchronized the stage-wise gap notation across the main text and appendix, while also correcting the truncation term in Theorem 6. Please see Theorem 1, Theorems 5, 6, and 8, Appendix A, Appendices B.2--B.4, and Appendix D. These edits corrected the statements and synchronized them with the recursive perturbation argument. However, we would like to note that these corrections did not change the underlying sequential-deflation recursion or the paper’s qualitative technical conclusion that earlier numerical errors have larger downstream influence. We also made minor supporting edits for consistency with this revision.
>
> **Response to Critical Change 2.**
>
> We agree that the term ``generalization'' was too strong for Theorems 5--6. In the revision, we reframed these results as fixed-design parameter-recovery statements: the contribution summary in Sec. 1 was updated accordingly, Sec. 4 was retitled to "Parameter recovery under fixed design" and its opening paragraph now explicitly stated that Theorems 5--6 concern recovery under fixed design rather than out-of-sample prediction risk, and the Limitations section repeated this boundary. Please see Sec. 1, Sec. 4, and Limitations.
>
> **Response to Critical Change 3.**
>
> We agree that the nonlinear LoRA section should not be read as a direct theorem-level prediction or as evidence for a simple monotone more-first rule. In the revision, we therefore both narrowed the claim and redesigned the nonlinear figures to make the intended evidence clearer. At the start of Sec. 5, we stated explicitly that the LoRA experiments are exploratory qualitative evidence rather than direct validations of the linear regression theorems, and we introduced a common one-parameter schedule family $t(\\alpha)$ so that schedule effects could be discussed consistently across the nonlinear experiments.
>
> Relative to the submitted version, the nonlinear visualizations were reorganized rather than merely redrawn. On the vision side, instead of the submitted Fig. 4 path plot, Sec. 5.2 now used revised Figs. 4--5. Revised Fig. 4 gave a compute-aware view within the more-first branch of the $t(\\alpha)$ family, while revised Fig. 5 broadened the comparison to equal and less-first schedules by sweeping $\\alpha$ directly. These updated plots made the qualitative message more precise: in this simple PEFT setting, more-first schedules often occupied a higher accuracy range than less-first schedules at comparable adaptation compute, with the equal schedule typically in between, but this was not presented as a theorem-level prediction or superiority claim.
>
> On the language side, the submitted Fig. 5 was split into revised Figs. 6--7 so that final sequential-vs.-joint performance and schedule effects could be read separately. The revised discussion around Fig. 7 then stated the two intended nuances explicitly: several more-first schedules improved over the equal schedule and closed the gap to the joint reference, but the dependence on $\\alpha$ within the more-first branch was not perfectly monotone and did not indicate a single universally optimal schedule.
>
> Please see the opening of Sec. 5, Secs. 5.2--5.3 (Figs. 4--7), and Limitations.

---

> > ### Author Response · Authors · 2026-04-29
> > **Response to Reviewer tfKm (continued)**
> >
> > **Response to Critical Change 4.**
> >
> > We agree that the comparison to jointly trained LoRA should be made compute-clean. Our previous analysis indeed did not account for the per-rank computational cost, nor for the fact that updating earlier ranks is less expensive than updating later ones (see Fig. 9). In the revision, we reported adaptation-training FLOPs rather than relying only on epochs, clarified in the main text that the single-budget comparison in Fig. 3 was already compute-close, and added an explicit compute-matched comparison in Appendix F. In particular, Appendix F now reports both the per-sample FLOP accounting (Fig. 9) and the resulting compute-matched comparison between sequential and jointly trained LoRA across representative schedules (Fig. 4, Fig. 7 and Fig. 8). We interpreted these results conservatively: sequential LoRA stayed in a similar accuracy range to joint LoRA under matched compute, without making a superiority claim.
> >
> > **Response to Critical Change 5.**
> >
> > We appreciated this suggestion and made the schedule definitions explicit in the revision. In Sec. 5 ("Common schedule family"), we introduced a single schedule family $t(\\alpha)$ and stated the exact allocation-and-rounding rule that maps $(r,T,m,\\alpha)$ to realized integer per-component budgets, with $\\alpha>0$ denoting more-first schedules, $\\alpha=0$ equal allocation, and $\\alpha<0$ less-first allocation. We then recorded the exact realized schedules used in Figure 1 in Appendix G. We also made minor supporting edits for consistency with this revision.
> >
> > **Response to Critical Change 6.**
> >
> > We agree that the original three-way schedule comparison was too coarse. In the revision, we therefore went beyond the three representative schedules by introducing a common one-parameter schedule family $t(\\alpha)$ and sweeping $\\alpha$ over a finer grid while keeping the same synthetic setting and total budget fixed. The resulting Fig. 2 now lets the reader see where more-first schedules helped and where the gains saturated, rather than relying only on the three anchor cases from Fig. 1. Please see the "Common schedule family" paragraph in Sec. 5 and Fig. 2 in Sec. 5.1.
> >
> > **Response to Critical Change 7.**
> >
> > We appreciate this comment and revised the framing accordingly. In the revision, we presented the paper explicitly as a theory paper for sequential low-rank linear regression, with the nonlinear LoRA experiments serving only as illustrative/exploratory extensions rather than a theoretical bridge to modern PEFT more broadly. In particular, the opening of Sec. 5 now stated that Secs. 5.2--5.3 provide exploratory qualitative evidence rather than direct theorem validations or superiority claims. Sec. 5.2 clarified that the vision experiments use intentionally simple feedforward PEFT testbeds and are not meant as realistic deployment scenarios, while Sec. 5.3 framed the DistilBERT/SST-2 study as a single nonlinear case study rather than a broad empirical bridge. This narrower framing now appeared in the contribution summary, the opening of Sec. 5, Secs. 5.2--5.3, and Limitations.
> >
> > **Additional clarification on theorem-aligned synthetic validation.**
> >
> > In Appendix G of the revised manuscript, we directly measured the schedule-dependent numerical-error terms $\\Psi_k$, the stage-wise gaps $T_k^\\star$, and the evaluated right-hand side of the bound in Theorem 1. Specifically, Figs. 16--17 showed that the measured dependencies tracked the same quantities that appeared in the Theorem 1.
> >
> > **Response to Strengthening Change 1.**
> >
> > We appreciate this suggestion. In the revision, we clarified the distinction between the exact, step-optimal, and finite-iteration rank-1 objects in Definition 1 and Sec. 3, standardized the notation used in the main proof chain, and pointed readers more explicitly to Table 1 before the denser appendix derivations. Please see Table 1.
> >
> > **Response to Strengthening Change 2.**
> >
> > We agreed that the explicit nonnegativity statement in Eq. (7) was redundant. In the revision, we removed that clause from Definition 1 and tightened the surrounding notation accordingly. Please see Definition 1 / Eq. (7).
> >
> > **Response to Strengthening Change 3.**
> >
> > We added full experimental and reproducibility details in Appendix E. Additionally, we clarified the compute calculation used in the PEFT comparisons in Appendix F. We will release the code publicly upon acceptance.
> >
> > **Response to Strengthening Change 4.**
> >
> > Thank you for this helpful suggestion. In the revision, we now included in the LoRA figure (Fig. 3), the mean performance and standard deviation across 5 runs.

---

> > > ### Author Response · Authors · 2026-04-29
> > > **Response to Reviewer tfKm (continued)**
> > >
> > > **Response to Strengthening Change 5.**
> > >
> > > We appreciated this suggestion and agreed that a broader sequential-learning testbed would be valuable. Given the rebuttal-time constraints, we did not add a new non-LoRA sequential benchmark in the revision. Instead, we sharpened the scope claim so that the paper was presented as a linear-theory paper with exploratory nonlinear PEFT experiments, and we left broader sequential learners as future work.
> > >
> > > We thank the reviewer once again for the helpful feedback and would be glad to discuss any additional questions or concerns.

---

### Author Response · Authors · 2026-04-29
**General Response**

We thank all reviewers for their thoughtful feedback. We are glad that the reviewers found our work clear (AXJc, UYWh), insightful (AXJc), and meaningful (UYWh). They also noted that it studies an interesting problem (PDsx, tfKm), provides a good understanding of error propagation (AXJc) and conveys an intuitive practical message (tfKm). In addition, the reviewers appreciated the connection to LoRA and noted the potential of our analysis to inform improving finetuning methods (AXJc, PDsx, UYWh, tfKm).

We have revised the manuscript according to the reviewers comments. In the revised version, we marked all changes in blue. Below, we address the main concerns raised by the reviewers.

**Clarity and ablation on the budget-allocation schedule.**

In the revised manuscript, we formalize the iteration-allocation schedule as a family parameterized by the hyperparameter $\\alpha$ in Sec. 5, and provide a concrete example of the resulting allocation for $\\alpha=1.0$ in Appendix G. We also included an ablation over $\\alpha$ in Figure 2. The results show that allocating more iterations to earlier components (i.e., $\\alpha > 0$) consistently outperforms equal iteration allocation across components ($\\alpha = 0$), while allocating fewer iterations to earlier components (i.e., $\\alpha < 0$) degrades performance. These findings are consistent with the intuition from our theoretical analysis in the linear setting.

**Non-Linear analysis and comparison with Joint LoRA.**
Our paper focuses its theoretical analysis on the linear setting, where we show that errors in earlier ranks propagate more strongly and suggest allocating more training iterations to the early components in sequential low-rank training. As reviewer tfKm noted, our experimental extension to the nonlinear setting, as well as the comparison with joint LoRA, was intended to be exploratory rather than definitive. In the revised manuscript, we show that under a fixed compute budget (total FLOPs) in the nonlinear setting with sequential low-rank update, allocating more iterations to earlier components yields an advantage over equal compute allocation across components (See Fig. 5 and Fig. 7). When compared with joint LoRA fine-tuning, sequential low-rank finetuning with the suggested schedule performs comparably overall and in some cases slightly outperforms joint LoRA for certain choices of the budget-allocation schedule (see Fig. 4, 7, and 8).

As reviewer tfKm also pointed out, sequential low-rank updates are potentially relevant in settings such as unknown-rank, streaming, and modular adaptation, where sequential optimization is natural and joint LoRA training does not directly apply. At the same time, we acknowledge that our nonlinear experiments are not large-scale and therefore do not fully test the generality of this approach. Their purpose was to examine whether the theoretical insights carry over qualitatively to more practical nonlinear settings, while leaving broader empirical validation to future work. To make this distinction clear, we revised Sections 5.2 and 5.3 to present the nonlinear experiments as exploratory qualitative evidence rather than as a direct validation of the linear theory. We also state this limitation explicitly in Sec. 6 and adjusted the conclusion accordingly in Section 7.

**Theoretical clarifications and proof-chain revisions.**
Across the reviews, the most common theoretical concern was to make the assumptions, notation, and proof chain fully explicit and internally consistent. In the revised manuscript, we corrected inaccuracies and synchronized the theorem statements with the appendix proofs. In particular, we fixed the Frobenius-tail expressions, standardized the numerical-error notation and the stage-wise spectral-gap terms, and clarified the perturbation step used in the rank-1 argument. We also made the key assumptions explicit by stating the positive spectral-gap requirement, clarifying the smallness condition under which the perturbation argument is valid, and spelling out the noise-ordering assumption together with a stronger sufficient regime under which the additive-noise term remains $O(1)$. Overall, these revisions make the theoretical claims more accurate and easier to verify, while preserving the main message of the analysis.

We again thank all reviewers for their helpful feedback and remain available to address any remaining concerns.

---

### Decision · Action_Editor_Spxa · 2026-05-16

**Recommendation:** Accept as is

**Audience:**

Yes

**Audience Explanation:**

All the reviewers agree the potential interests within the community.

**Claims And Evidence:**

Yes

**Claims Explanation:**

The practical message is intuitive: errors made early in the sequential deflation chain contaminate later residuals, so early components should matter more.

All the reviewers believe that no major technical concerns. During the review process, One reviewer mentioned the key technical inconsistencies. Later the authors reframed the “generalization” results appropriately as fixed-design parameter-recovery guarantees, and made the nonlinear LoRA section by presenting it as exploratory qualitative evidence rather than direct validation of the linear theory. Therefore, I do not see outstanding concerns.